# Beyond Raw Detection Scores: Markov-Informed Calibration for Boosting Machine-Generated Text Detection

**Chenwang Wu**[1]   **Yiu-ming Cheung**[1]*  **Shuhai Zhang**[2]   **Bo Han**[1]   **Defu Lian**[3]

[1]Department of Computer Science, Hong Kong Baptist University, Hong Kong, China
[2]School of Software Engineering, South China University of Technology, Guangzhou, China
[3]School of Computer Science, University of Science and Technology of China, Hefei, China
{cscwwu, ymc, bhanml}@comp.hkbu.edu.hk, shuhaizhangshz@gmail.com,
liandefu@ustc.edu.cn

## Abstract

While machine-generated texts (MGTs) offer great convenience, they also pose risks such as disinformation and phishing, highlighting the need for reliable detection. Metric-based methods, which extract statistically distinguishable features of MGTs, are often more practical than complex model-based methods that are prone to overfitting. Given their diverse designs, we first place representative metric-based methods within a unified framework, enabling a clear assessment of their advantages and limitations. Our analysis identifies a core challenge across these methods: the token-level detection score is easily biased by the inherent randomness of the MGTs generation process. To address this, we theoretically and empirically reveal two relationships of context detection scores that may aid calibration: Neighbor Similarity and Initial Instability. We then propose a Markov-informed score calibration strategy that models these relationships using Markov random fields, and implements it as a lightweight component via a mean-field approximation, allowing our method to be seamlessly integrated into existing detectors. Extensive experiments in various real-world scenarios, such as cross-LLM and paraphrasing attacks, demonstrate significant gains over baselines with negligible computational overhead. The code is available at https://github.com/tmlr-group/MRF_Calibration.

## 1 Introduction

In recent years, generative AI, represented by large language models (LLMs) (Achiam et al., 2023; Radford et al., 2019), has advanced rapidly, and the machine-generated texts (MGTs) they produce often match human writing in fluency, coherence, and diversity. While this technological breakthrough offers immense opportunities, it has also triggered widespread societal concerns, such as the spread of disinformation (Vykopal et al., 2024), the violation of intellectual property rights (Yu et al., 2023b), and phishing attacks (Hong, 2012). Therefore, the research and development of MGT detection technologies hold significant theoretical and practical value in uncovering the distinct patterns of generated text and ensuring a trustworthy AI environment.

An effective detection method is to identify LLM's watermarks (Hou et al., 2024), but this requires injecting watermarks into the LLM, which is often impractical due to high access permissions. Therefore, passive detection methods, including model-based and metric-based methods, have garnered significant attention. Model-based methods, which use a set of human- and machine-generated texts to train a binary classifier, such as OpenAI detector (Solaiman et al., 2019), ChatGPT detector (Guo et al., 2023), SeqXGPT (Wang et al., 2023), and CoCo (Liu et al., 2022). However, such models are often too complex, leading to overfitting to the training data. Instead, metric-based methods exploit the inherent statistical biases of LLM to discriminate MGTs, which is model-agnostic and has better generalization properties. These methods use metrics such as log-likelihood, log-rank,

---

*Corresponding author: Yiu-ming Cheung (ymc@comp.hkbu.edu.hk).

and entropy. Furthermore, methods such as DetectGPT (Mitchell et al., 2023), DNA-GPT (Yang et al., 2024), and SimLLM (Nguyen-Son et al., 2024) detect MGTs by comparing the differences between a given text and a perturbed, regenerated, or continued text from an alternative model.

Obviously, metric-based methods exhibit diverse designs. Therefore, this paper first systematically examines several representative approaches, including Log-Likelihood (Solaiman et al., 2019), Log-Rank (Mitchell et al., 2023), Entropy (Gehrmann et al., 2019), DetectGPT (Mitchell et al., 2023), Fast-DetectGPT (Bao et al., 2024), and DNA-GPT (Yang et al., 2024), and situates them within a unified framework (**Section 2**). Our analysis reveals that they share a threshold-based detection criterion, with some commonalities, such as the inclusion of auxiliary data (e.g., perturbed texts). This offers a theoretical basis for understanding their core mechanisms, strengths, and limitations.

Based on the unified framework, we summarize the core challenge of metric-based methods (**Section 2**): the imprecision of token-level detection scores. Specifically, since these methods make decisions based on a threshold, their effectiveness is directly tied to score precision. However, randomness introduced by the LLM sampling mechanism can violate their underlying assumptions, leading to biased scores with low discrimination, as shown in "w/o refine" in Fig. 1 (more results can be found in Appendix E.1). Moreover, they tend to derive an overall detection score by naively aggregating token-level scores, failing to correct the underlying imprecision. Therefore, calibrating the token-level detection score is essential for improving overall detection performance.

Given that detection scores are tied to tokens and the LLM generation process induces dependencies among tokens (Achiam et al., 2023), context tokens' detection scores may have relationships that are easy to overlook. Revealing and modeling these relationships may help calibrate these scores. Accordingly, we theoretically and empirically reveal two relationships among detection scores of context tokens (**Section 3**): **Neighbor Similarity**, where adjacent tokens exhibit similar detection scores, and **Initial Instability**, where the detection scores of initial tokens are unstable.

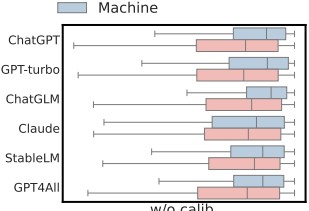 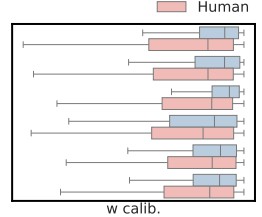

Figure 1: Distribution of token scores obtained by the DetectGPT method without and with score calibration in the Essay dataset. The proposed calibration method enhances the discriminative nature of the token scores.

Finally, building on these two relationships, we propose a Markov-informed score calibration method to enhance MGT detection (**Section 4**). Our method models the identified relationships through Markov random fields and, via a mean-field approximation, implements it as a lightweight iterative neural network. As shown by the more discriminative detection scores in the "w refine" method in Fig. 1, the proposed method boosts the discriminative nature of the scores. Notably, our method can be seamlessly stacked on top of existing detectors without architectural changes, providing flexibility. Compared with complex model-based approaches, our method introduces only a negligible 2×2 parameterization, making its computational delay negligible and less prone to overfitting. Extensive experiments demonstrate our method's enhanced effectiveness. Our contributions can be summarized as follows:

- We view existing metric-based detection methods through a unified lens, which facilitates precise comparison and enables potential improvements.

- We theoretically and empirically demonstrate that token-level detection scores exhibit neighbor similarity and initial instability, offering avenues for improved detection.

- We propose a Markov-informed score calibration method and, via a mean-field approximation, implement it as a lightweight component that can be seamlessly integrated into existing detectors to further unlock their potential.

- We conduct extensive experiments across three datasets to demonstrate superior performance in diverse scenarios, including cross-LLM, cross-domain, mixed-text, and paraphrase attacks.

Table 1: Comparing existing metric-based methods from a unified view. Here, $s$ is the text to be detected containing $N$ tokens, $s'$ is the perturbed text generated by DetectGPT, $\tilde{s}$ and $\hat{s}$ are the regenerated texts of Fast-DetectGPT and DNA-GPT, respectively. Function $\mu(\cdot)$ and $\sigma(\cdot)$ represent the mean and standard deviation of the given set, respectively.

| Method | Data | Token-level Score | Score Aggregation | Detection |
|---|---|---|---|---|
| Log-likelihood | $s$ | $\log p(s_t\|s_{<t})$ | $\frac{1}{N-1}\sum_{t=2}^{N}\log p(s_t\|s_{<t})$ | $score > \epsilon$ |
| Log-Rank | $s$ | $\text{rank}(p(s_t\|s_{<t}))$ | $\frac{1}{N-1}\sum_{t=2}^{N}\text{rank}(p(s_t\|s_{<t}))$ | $score > \epsilon$ |
| Entropy | $s$ | $\sum_{v\in V} p(v\|s_{<t})\log p(v\|s_{<t})$ | $-\frac{1}{N-1}\sum_{t=2}^{N}\sum_{v\in V} p(v\|s_{<t})\log p(v\|s_{<t})$ | $score > \epsilon$ |
| DetectGPT | $\{s, s_1', s_2', ..., s_n'\}$ | $p(s_t\|s_{<t})$ | $\frac{\frac{1}{N-1}\sum_{t=2}^{N}p(s_t\|s_{<t})-\mu(\{\frac{1}{N-1}\sum_{t=2}^{N}p(s_{i,t}'\|s_{i,<t}')\}_i)}{\sigma(\{\frac{1}{N-1}\sum_{t=2}^{N}p(s_{i,t}'\|s_{i,<t}')\}_i)}$ | $score > \epsilon$ |
| Fast-DetectGPT | $\{s, \tilde{s}_1, ..., \tilde{s}_n\}$ | $p(s_t\|s_{<t})$ | $\frac{\frac{1}{N-1}\sum_{t=2}^{N}p(s_t\|s_{<t})-\mu(\{\frac{1}{N-1}\sum_{t=2}^{N}p(\tilde{s}_{i,t}\|s,\tilde{s}_{i,<t})\}_i)}{\sigma(\{\frac{1}{N-1}\sum_{t=2}^{N}p(\tilde{s}_{i,t}\|s,\tilde{s}_{i,<t})\}_i)}$ | $score > \epsilon$ |
| DNA-GPT | $\{s, \hat{s}_1, ..., \hat{s}_n\}$ | $p(s_t\|s_{<t})$ | $\frac{1}{N-1}\sum_{t=2}^{N}\log p(s_t\|s_{1:t-1}) - \frac{1}{n}\sum_{i=1}^{n}\frac{1}{N_t-1}\sum_{t=2}^{N_t}\log p(\hat{s}_{i,t}\|\hat{s}_{i,1:t-1})$ | $score > \epsilon$ |

## 2 A UNIFIED PERSPECTIVE ON METRIC-BASED DETECTION

Although model-based methods have shown competitive potential in specific domains, they are often too complex, leading to a tendency to overfit their training data. This limitation requires them to re-train or fine-tune for newly released LLMs, which hinders their generalizability. In contrast, metric-based methods extract discriminative features from MGT, and their model-agnostic nature provides superior generalization potential. Given the diverse implementations of representative metric-based methods such as Log-Likelihood (Solaiman et al., 2019), Log-Rank (Gehrmann et al., 2019), Entropy (Gehrmann et al., 2019), DetectGPT (Mitchell et al., 2023), Fast-DetectGPT (Bao et al., 2024), and DNA-GPT (Yang et al., 2024), we first provide a systematic examination of them from a unified perspective. This facilitates a deeper understanding of their mechanisms and allows for a fair comparison of their strengths and weaknesses. As illustrated in Table 1, we compare these methods across data, score aggregation, and detection dimensions. Note that their diverse core metric designs are not discussed here. This is because we aim to design a general enhancement framework decoupled from specific detectors, requiring us to start from possible commonalities rather than diverse designs.

- **Data**. Log-likelihood, Log-Rank, and Entropy are computationally efficient as they rely solely on the original input text $s$. However, the detection error based on a single text may be large, because the randomness inherent in the LLM sampling mechanism may cause the MGT to deviate from these methods' underlying assumptions, e.g., Log-Rank assumes that the generated tokens have high rankings. In contrast, DetectGPT, Fast-DetectGPT, and DNA-GPT incorporate multiple perturbed (i.e., $s'$) or regenerated (i.e., $\tilde{s}$ and $\hat{s}$) samples, which mitigates the errors caused by randomness. However, this comes at the cost of increased computational overhead compared to single-text-based methods.

- **Score Aggregation**. Although these methods appear to calculate scores differently, they all tend to directly aggregate token scores to obtain the final text score, typically through summation. As discussed, the randomness introduced by the LLM generation process may cause token-level scores to be biased. Therefore, the direct aggregation of these potentially imprecise token scores may result in an inaccurate final detection score.

- **Detection**. These methods employ threshold-based detection mechanisms, whose effectiveness relies heavily on the accuracy of their calculated scores. Including uncalibrated, high-noise scores in threshold-based decision-making may lead to poor performance.

In summary, to enhance detection, existing methods incorporate more textual information (e.g., regenerated texts in DetectGPT and Fast-DetectGPT) and use different score calculation strategies (e.g., the Likelihood difference between the detected and regenerated text in DNAGPT). However, as we discussed, they fail to address the underlying token-level errors caused by inherent randomness, limiting their detection potential. Considering that detection scores are tied to tokens and the LLMs' generative mechanism induces dependencies among tokens, revealing and modeling the relationships between tokens may help correct score errors and thus improve detection effectiveness.

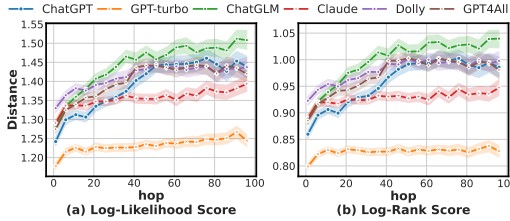 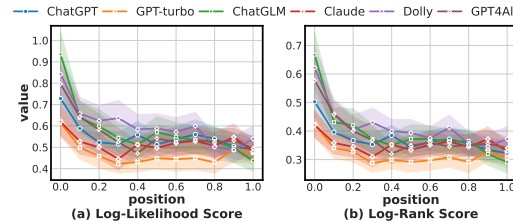

Figure 2: The detection score distances (Mean Absolute Difference) of neighbors at different hops in the Essay dataset. Log-likelihood and Log-Rank score are used.

Figure 3: The detection score distances (Mean Absolute Difference) of 1-hop neighbors at different normalized relative positions in Essay. Log-likelihood and Log-Rank score are used.

## 3 RELATION BETWEEN CONTEXTUAL TOKEN-LEVEL DETECTION SCORES

To understand the relationship between context tokens' detection scores, following existing work (Liu et al., 2023), we consider the token generation process of a simplified model: a single-layer transformer model with single-head attention:

$$x_{t+1} = \mathcal{F}(a_t), \quad \text{where} \quad a_t = \text{softmax}\left(1/t \cdot x_t W_Q W_K^\top X_{t-1}^\top\right) X_{t-1} W_V W_O. \tag{1}$$

$x_t \in \mathbb{R}^{1 \times d}$ is the embedding of token $s_t$, and $d$ denotes the embedding dimension. The matrix $X_{t-1} \in \mathbb{R}^{(t-1) \times d}$ is stacked by the embeddings $x_1, \ldots, x_{t-1}$, where the $j$-th row is $x_j$. $W_Q, W_K, W_V \in \mathbb{R}^{d \times d}$ and $W_O \in \mathbb{R}^{d \times d}$ are the attention weights. Following the attention block, an MLP block, denoted as $\mathcal{F} : \mathbb{R}^{1 \times d} \to \mathbb{R}^{1 \times d}$, is applied, and it is a two-layer network with skip connections:

$$\mathcal{F}(x) = x + W_2 \, \text{relu}\,(W_1 x).$$

As shown in Table 1, the detection score of token $s_t$ is usually the function of $s_t$ (i.e., $x_i$), and $x_i$ is related to the attention scores $\alpha_{t-1} = \text{softmax}\left(1/t \cdot x_t W_Q W_K^\top X_{t-1}^\top\right)$ in Eq. (1). The following theorem will reveal the relationship between attention scores, which in turn help us understand the relationship between detection scores of context tokens.

**Theorem 1.** *Let $\lambda_K, \lambda_Q, \lambda_V, \lambda_O$ be the largest singular values of parameters $W_K, W_Q, W_V, W_O$, respectively, and let $W = W_V W_O W_Q W_K^\top$. For the transformer defined in Eq. (1), assuming normalized inputs ($\|x_t\|_2 = 1$ for all $t$) and constants $c, \epsilon > 0$, consider $a_t x_{t+1}^\top \geq (1 - \delta) \|a_t\|_2$ with $\delta \leq \left(\frac{c\epsilon}{\lambda_Q \lambda_K \lambda_V \lambda_O}\right)^2$. If $x_\ell$ satisfies $x_\ell W x_\ell^\top \geq c$ and $x_\ell W x_\ell \geq \epsilon^{-1} \max_{j \in [\ell], j \neq \ell} x_j W x_\ell^\top$, then*

$$\alpha_{t+1,l} \leq \frac{\exp\left(C_l \cdot \alpha_{t,l} + \eta\right)}{\exp\left(C_l \cdot \alpha_{t,l} + \eta\right) + \sum_{j \neq l} \exp\left(C_j \cdot \alpha_{t,j} - \eta\right)},$$

$$\alpha_{t+1,l} \geq \frac{\exp\left(C_l \cdot \alpha_{t,l} - \eta\right)}{\exp\left(C_l \cdot \alpha_{t,l} - \eta\right) + \sum_{j \neq l} \exp\left(C_j \cdot \alpha_{t,j} + \eta\right)},$$

*where*

$$C_j = \frac{x_j W x_j^T}{t \, |a_t|_2}, \text{and } \eta = \frac{(1 + \sqrt{2})\epsilon x_j W x_j^T}{(t + 1) \, |a_t|_2}.$$

The proof can be found in Appendix C. This theorem establishes the upper and lower bounds for the attention score at step $t + 1$, which are determined by the attention scores at step $t$. In Transformer, the function mapping $a_t \to \log p(x_{t+1})$ is continuous, constraints on the dynamics of $a_t$ naturally propagate to the detection scores. Therefore, Theorem 1 can reveal two relationships in token-level detection scores:

- **Neighbor Similarity**, where the detection scores of adjacent tokens in a sequence exhibit statistically lower variance compared to tokens that are far apart. This is because Theorem 1 creates a positive feedback loop analogous to simulated annealing (Kirkpatrick et al., 1983), where high scores at the current step lead to high scores at the next, and vice versa. That is, the attention (also token-level detection scores) cannot shift fast between steps.

- **Initial Instability**, where the detection scores of initial tokens in a sequence are statistically unstable compared to the subsequent tokens (i.e., fluctuate greatly). This is because these bounds are closely tied to the current step $t$. When $t$ is small (early position), $\eta$ and $C$ are large, allowing for dramatic fluctuations in $a_t$. That is, the attention score (also token-level detection scores) is unstable in the initial generation process.

Given that analyzing the dynamics of full-scale LLMs is mathematically intractable, we follow a practical setup Liu et al. (2023) and start with a simplified model, which provides theoretical motivation for the neighbor similarity and initial instability phenomena. We then empirically verify these two findings.

First, to empirically validate the neighbor similarity property, we evaluated the distance (mean absolute difference) in detection scores across k hops (i.e., $\frac{1}{|S|} \frac{1}{N-K} \sum_{s \in S} \sum_{t=0}^{N-K} |score(s_t) - score(s_{t+k})|$, where $S$ is the text set, and $score(s_t)$ is provided in Table 1). As illustrated in Fig. 2 (more results can be found in Appendix E.2), there is a clear positive correlation between the detection score distance and the hop, and adjacent tokens have the highest detection score similarity; thereby providing empirical evidence for our theoretical finding.

Second, to validate the initial instability property, we analyzed the distance in detection score between adjacent tokens at different percentage positions (i.e., $\frac{1}{S} \sum_{s \in S} |score(s_t) - score(s_{t+1})|$). Fig. 3 illustrates that the score difference is substantially larger for tokens at the beginning of the text and progressively decreases, eventually stabilizing. More results can be found in Appendix E.2. Considering our established finding on neighbor similarity (i.e., adjacent scores should be highly similar), a high detection score difference at the sequence beginning indicates a significant instability in detection scores of initial tokens.

## 4 MARKOV-INFORMED DETECTION SCORE CALIBRATION

Based on the revealed relationships, this section uses an MRF to capture them (Section 4.1) and adopts the mean field approximation to model the MRF model as a lightweight component stacked on existing detectors to calibrate detection scores (Section 4.2), thereby enhancing detection.

### 4.1 MARKOV RANDOM FIELD FOR MGT DETECTION

We capture these two types of relationships by modeling the joint probability distribution of text's token detection scores through pairwise Markov Random Fields (pMRF). Specifically, for each token $s_t$ in text $s$, we assign a binary random variable $y_{s_t}$, where $y_{s_t} = 0$ and $y_{s_t} = 1$ indicate a human- or machine-generated token [1], respectively, as measured by the detection score of the token. Let $y_s$ denote the label set for all tokens in text $s$, the pMRF over these tokens can be formalized as a Gibbs distribution: $P(y_s) = \frac{1}{Z} \exp(-E(s, y_s))$, where $Z$ is a normalizing constant and $E(s, y_s)$ is the energy function. Our objective is to maximize the posterior probability of the token labels $y_s$ by minimizing the global energy function $E(s, y_s)$. The energy function typically consists of two components: the unary potential $\Psi_U$ and the pairwise potential $\Psi_P$:

$$E(s, y_s) = \sum_{t=1}^{N} \Psi_U(s_t, y_{s_t}) + \sum_{t=1}^{N} \sum_{s_j \in \mathcal{N}(s_t)} \Psi_P(y_{s_t}, y_{s_j}),$$

where $\mathcal{N}(s_t)$ denotes the adjacent tokens of token $s_t$, i.e., $\mathcal{N}(s_t) = \{s_{t-1}, s_{t+1}\}$ (if existing).

**Unary potential** $\Psi_U(s_t, y_{s_t})$ quantifies the cost of assigning label $y_{s_t}$ to token $s_t$. We let $\Psi_U(s_t, y_{s_t}) = -\log p(s_t)$, where $p(s_t)$ is the output probability from the original detector. For detectors without probability output, it is measured by the 0-1 normalized score of token $s_t$.

**Pairwise potential** $\Psi_P(y_{s_t}, y_{s_j})$ models the similarity in detection scores between adjacent tokens. A penalty is applied if two adjacent tokens are assigned different labels; otherwise, a reward is given. This enforces label smoothness and captures the neighbor similarity property:

$$\Psi_P(y_{s_t}, y_{s_j}) = w \cdot (2 \cdot I(y_{s_t} \neq y_{s_j}) - 1), \tag{2}$$

---

[1]Note that token labels are not absolute but depend on the context in which they appear. For example, "the" can be a human token or a machine token depending on the text.

where $I(\cdot)$ is the indicator function, and the reward and penalty factor $w \geq 0$. This implies an energy penalty of $w$ when adjacent tokens have different labels; otherwise, the reward is $-w$.

To model the initial instability property, we introduce a positional weighting function $\beta(t)$ in the binary potential. This function assigns lower weights to binary potentials at earlier positions, thereby mitigating the amplification of energy errors caused by unstable initial neighbor tokens. In this paper, we define the positional weighting function $\beta(t)$ as a Sigmoid function to ensure a smooth transition of weights, and the revised binary potential is then given by:

$$\Psi_P\left(y_{s_i}, y_{s_j}\right) = \beta(j) \cdot w \cdot (2 \cdot I(y_{s_i} \neq y_{s_j}) - 1), \ \ \text{with} \ \ \beta(t) = \frac{1}{1 + exp(-(t - t_0))}, \quad (3)$$

where $t_0$ is the weighted center, effectively suppressing the pairwise potential of tokens before $t_0$.

## 4.2 MEAN FIELD APPROXIMATE IN MGT DETECTION

Given the MRF model, this subsection details how to model it as a lightweight component stacked on the original detector through mean field approximation theory, thereby enhancing detection.

In the MRF posterior probability $P(y_s) = \frac{1}{Z} \exp(-E(s, y_s))$, the partition function $Z = \sum_{y_s} exp(-E(s, y_s))$, which is obtained by adding over all possible combinations of $y_s$. For a text with $N$ tokens, there are $2^N$ combinations, making exact computation of $P(y_s|s)$ infeasible. Inspired by existing work (Deng et al., 2022), we employ mean-field theory for approximate inference. Its core idea is to use a simpler, factorized distribution $Q\left(y_s\right) = \prod_{t=1}^N Q_{s_t}\left(y_{s_t}\right)$ to approximate the true joint distribution $P(y_s)$, achieved by minimizing the KL divergence between these two distributions:

$$\begin{aligned} D(Q\|P) =& \mathbb{E}_{y_s \sim Q}\left[\log Q\left(y_s\right)\right] - \mathbb{E}_{y_s \sim Q}\left[\log P\left(y_s\right)\right] \\ =& \sum_{t=1}^N \mathbb{E}_{y_{s_t} \sim Q_{s_t}}\left[\log Q_{s_t}\left(y_{s_t}\right)\right] + \mathbb{E}_{y_s \sim Q}\left[E\left(s, y_s\right)\right] + \log Z. \end{aligned} \quad (4)$$

The first equation is the definition of KL divergence, and the second is obtained by subsituting $P(y_s)$ and $Q(y_s)$. Then, we define a Lagrangian composed of all terms involving $Q_{s_t}\left(y_{s_t}\right)$ in $D(Q\|P)$:

$$L_{s_t}(Q) = Q_{s_t}\left(y_{s_t}\right) \log Q_{s_t}\left(y_{s_t}\right) + \mathbb{E}_{y_s \sim Q}\left[E\left(s, y_s\right)\right] + \lambda(\sum_{y_{s_t}} Q_{s_t}\left(y_{s_t}\right) - 1). \quad (5)$$

Here, the term involving Lagrange multiplier $\lambda$ assures that $Q_{s_i}$ is a proper probability distribution. Now we take derivatives of Eq. (5) with respect to $Q_{s_t}(y_{s_t})$ and set the corresponding derivative to 0, then we get the optimal $Q_{s_t}(y_{s_t})$:

$$Q_{s_t}(y_{s_t}) = \frac{1}{z} exp(\Psi_U(s_t, y_{s_t}) - \sum_{s_j \in \mathcal{N}(s_t)} \mathbb{E}_{y_{s_j} \sim Q_{s_j}}\left[\Psi_P(y_{s_t}, y_{s_j})\right]). \quad (6)$$

We can express the single token calculation in Eq. (6) as a matrix form of multiple token calculations, which involves three steps:

- **Unary potential calculation**. Corresponding to the unary potential $\Psi_U(s_t, y_{s_t})$ of token $s_t$ being $-\log p(s_i)$, we can express the unary potential of text $s$ in matrix form $-\log H$, where the i-th row of $H$ corresponds to token $s_i$'s prior probability and the two columns correspond to identity labels (HGT and MGT), that is, $H = [1 - p(s), p(s)]$, where $p(s) = [p(s_1), ..., p(s_N)]^\top$.

- **Pairwise potential calculation**. For the revised pairwise potential (i.e., Eq. (3)), we can define the weighted adjacency matrix as $A^{corr}$, where $A_{t,t+1}^{corr} = \beta(t + 1)$ for all $t = 0, ..., N - 2$, $A_{t-1,t}^{corr} = \beta(t - 1)$ for all $t = 1, ..., N - 1$, and 0 otherwise. Then the matrix form of the weighted pairwise potential is $A^{corr}Q \begin{bmatrix} -w & w \\ w & -w \end{bmatrix}$. This update strategy enforces that all relationships receive the same reward or penalty. However, the influence of MGT neighbors and HGT neighbors is intuitively different, so this kind of reward and punishment mechanism with the same weight will limit the expressive ability of MRF. To this end, we relax the weights for different relationships and get the pairwise potential as $A^{corr}Q(W_{mrf} \odot \begin{bmatrix} -1 & 1 \\ 1 & -1 \end{bmatrix})$, where $W_{mrf} \in \mathbb{R}_+^{2 \times 2}$.

- **Normalization**. The operator $\frac{1}{z}exp(\cdot)$ in Eq. (6) can be naturally modeled as Softmax function.

In summary, we get the following update rule for tokens' detection scores:

$$Q = \text{softmax}\left(\log H - A^{corr}Q\left(W_{mrf} \odot \begin{bmatrix} -1 & 1 \\ 1 & -1 \end{bmatrix}\right)\right). \tag{7}$$

Notably, Eq. (7) shows that the computation of $Q$ relies on $Q$ itself; hence, iterative computation is required. Initializing the initial $Q$ to $H$, we can get the iterative version as follows:

$$Q^t = \text{softmax}\left(\log Q^{t-1} - A^{corr}Q^{t-1}\left(W_{mrf} \odot \begin{bmatrix} -1 & 1 \\ 1 & -1 \end{bmatrix}\right)\right), \text{ where } Q^0 = H. \tag{8}$$

In addition to using position weight function $\beta(t)$ in pairwise potential to reduce the impact of initial unstable scores, we also use this position weight function on the final calibrated scores $Q^T$ of $T$ iterations to reduce their impact on detection:

$$Q_{final} = [\beta(1), ..., \beta(N)] \odot Q^T. \tag{9}$$

The MRF-informed calibration component is then directly stacked on the original detector to calibrate the token detection scores. Specifically, if the de-

---

**Algorithm 1** Markov-informed Enhancement Framework

1: **Input:** Text $s$ to be detected, the original detector $f_1 \circ f_2$, iteration steps $T$, MRF weights $W$.
2: Construct $A^{corr}$ based on $s$.
3: Get each token $s_i$'s detection score from the detection score calculation module $f_1$, and set $Q^0 = H$.
4: **for** $t = 0$ **to** $T - 1$ **do**
5:     Update $Q$ according to Eq. (8).
6: **end for**
7: Calculate $Q_{final}$ according to Eq. (9).
8: **Return** detection score $f_2(Q_{final})$ of text $s$.

---

tector is formalized as a combination of the detection score calculation module $f_1$ and the detection module $f_2$, i.e., $f(s) = f_1 \circ f_2(s)$, the MRF-informed component of Eq. (9) is defined as $f_{mrf}$, then the enhanced detector is $f_{enh.}(s) = f_1 \circ f_{mrf} \circ f_2(s)$. The complete inference process is shown in Alg. 1. To learn weights $W$, we use supervised training: $\mathcal{L} = -\sum_{s \in \mathcal{D}_{train}}(Y_s \log f_{enh.}(s) + (1 - Y_s) \log(1 - f_{enh.}(s)))$.

**Computational Complexity**. The MRF layer can be computed using sparse-dense matrix multiplication. For a text containing $N$ tokens, the number of iterations is T, resulting in a computational complexity of $\mathcal{O}(NT)$, which is negligible for calculating the detection score. We will also empirically verify the efficiency of our method in Appendix E.15.

## 5 EXPERIMENTS

**Dataset**. We conducted experiments on three widely used public datasets: Essay (Verma et al., 2024), Reuters (Verma et al., 2024), DetectRL (Wu et al., 2024), and TruthfulQA (Lin et al., 2022). The Essay and Reuters datasets collect machine text generated by GPT4All, ChatGPT, ChatGPT-turbo, ChatGLM, Dolly, and Claude. The TruthfulQA dataset collects machine text from GPT4, ChatGPT-turbo, ChatGLM, Dolly, ChatGPT, and StableLM. The DetectRL dataset not only includes pure machine-generated text by Google-PaLM, ChatGPT, and Llama-2-70b, but its unique mixed, paraphrased, and cross-domain texts also allow us to more comprehensively evaluate the model's performance in complex real-world scenarios. For a complete description of the datasets, please refer to Appendix D.1.

**Baselines**. We select the following metric-based methods for comparison and enhancement: Log-Likelihood (Likelihood) (Solaiman et al., 2019), Log-Rank (Gehrmann et al., 2019), Entropy (Gehrmann et al., 2019), DetectGPT (Mitchell et al., 2023), Fast-DetectGPT (FastGPT) (Bao et al., 2024), DNA-GPT (Yang et al., 2024), Repreguard (Chen et al., 2025), Lastde (Xu et al., 2025), FourierGPT (Xu et al., 2024), and Binoculars Hans et al. (2024). The versions equipped with the proposed method are defined with 'M' suffix, e.g., Likelihood-M. Furthermore, although we focus on metric-based methods, we also compare with model-based methods ChatGPT-D (Guo et al., 2023) and MPU (Tian et al., 2024). Their details can be found in Appendix D.2.

**Metrics**. First, as a binary classification problem, we use the area under the receiver operating characteristic curve (AUROC). Second, following (Tufts et al., 2024; Fraser et al., 2025; Hans et al., 2024), we recognize the negative impact of misclassifying human text as machine-generated text. Therefore, another important evaluation metric is the true positive rate (TPR) at a low false positive rate (FPR). Specifically, we measure the TPR at an FPR of 1%, denoting this as TPR@FPR-1%.

Table 2: Performance concerning AUROC (%) on Essay (left) and DetectRL (right).

| Method | Essay (Training Text: GPT4All) | | | | | | | DetectRL (Training Text: Llama-2-70b) | | | |
|---|---|---|---|---|---|---|---|---|---|---|---|
| | GPT4All | ChatGPT | Dolly | ChatGLM | Claude | ChatGPT-turbo | Avg. | Llama-2-70b | ChatGPT | Google-PaLM | Avg. |
| Likelihood | $96.16_{\pm0.30}$ | $98.79_{\pm0.19}$ | $90.90_{\pm1.33}$ | $99.29_{\pm0.25}$ | $92.76_{\pm0.23}$ | $99.13_{\pm0.19}$ | 96.17 | $78.58_{\pm0.41}$ | $66.61_{\pm0.99}$ | $71.42_{\pm0.49}$ | 72.20 |
| **Likelihood-M** | $98.58_{\pm0.14}$ | $99.47_{\pm0.11}$ | $94.59_{\pm0.96}$ | $99.54_{\pm0.18}$ | $94.82_{\pm0.36}$ | $99.72_{\pm0.13}$ | 97.79 | $87.61_{\pm0.48}$ | $73.70_{\pm0.58}$ | $81.21_{\pm1.21}$ | 80.84 |
| Log-Rank | $96.55_{\pm0.31}$ | $98.95_{\pm0.13}$ | $90.08_{\pm1.28}$ | $99.36_{\pm0.13}$ | $92.01_{\pm0.20}$ | $99.24_{\pm0.15}$ | 96.03 | $79.67_{\pm0.46}$ | $65.85_{\pm0.94}$ | $70.66_{\pm0.40}$ | 72.06 |
| **Log-Rank-M** | $98.57_{\pm0.06}$ | $99.41_{\pm0.09}$ | $93.82_{\pm0.91}$ | $99.55_{\pm0.08}$ | $92.91_{\pm0.30}$ | $99.64_{\pm0.10}$ | 97.32 | $90.20_{\pm0.44}$ | $75.32_{\pm0.91}$ | $84.34_{\pm0.61}$ | 83.29 |
| Entropy | $74.19_{\pm1.62}$ | $89.49_{\pm0.34}$ | $73.26_{\pm1.48}$ | $84.11_{\pm0.77}$ | $86.58_{\pm0.66}$ | $95.94_{\pm0.35}$ | 83.93 | $66.21_{\pm0.88}$ | $63.09_{\pm1.05}$ | $60.73_{\pm0.91}$ | 63.34 |
| **Entropy-M** | $83.52_{\pm0.73}$ | $93.28_{\pm0.15}$ | $81.11_{\pm0.91}$ | $91.44_{\pm0.35}$ | $87.96_{\pm0.48}$ | $96.75_{\pm0.17}$ | 89.01 | $69.97_{\pm0.90}$ | $65.26_{\pm1.36}$ | $65.82_{\pm0.96}$ | 67.02 |
| DetectGPT | $50.81_{\pm0.58}$ | $46.40_{\pm0.77}$ | $57.48_{\pm0.84}$ | $50.41_{\pm1.70}$ | $41.54_{\pm0.60}$ | $17.90_{\pm1.25}$ | 44.09 | $52.37_{\pm0.59}$ | $50.22_{\pm0.60}$ | $43.19_{\pm1.41}$ | 48.60 |
| **DetectGPT-M** | $95.37_{\pm2.42}$ | $96.20_{\pm2.48}$ | $85.39_{\pm7.02}$ | $95.97_{\pm2.93}$ | $80.29_{\pm10.43}$ | $98.49_{\pm0.78}$ | 91.95 | $78.81_{\pm3.68}$ | $64.15_{\pm4.48}$ | $75.90_{\pm4.53}$ | 72.95 |
| FastGPT | $64.63_{\pm1.53}$ | $67.68_{\pm1.70}$ | $47.17_{\pm1.53}$ | $71.08_{\pm1.51}$ | $75.31_{\pm0.90}$ | $88.62_{\pm0.67}$ | 69.08 | $67.72_{\pm1.02}$ | $58.50_{\pm1.22}$ | $56.68_{\pm1.21}$ | 60.97 |
| **FastGPT-M** | $87.22_{\pm3.40}$ | $91.56_{\pm3.33}$ | $82.61_{\pm17.98}$ | $95.36_{\pm0.48}$ | $59.29_{\pm1.89}$ | $63.48_{\pm18.13}$ | 79.92 | $66.55_{\pm3.36}$ | $55.19_{\pm3.26}$ | $63.30_{\pm4.37}$ | 61.68 |
| DNA-GPT | $98.08_{\pm0.23}$ | $96.56_{\pm0.28}$ | $92.78_{\pm0.68}$ | $98.00_{\pm0.13}$ | $89.92_{\pm0.29}$ | $96.22_{\pm0.27}$ | 95.26 | $71.91_{\pm0.91}$ | $64.14_{\pm0.91}$ | $67.32_{\pm0.94}$ | 67.79 |
| **DNA-GPT-M** | $99.68_{\pm0.07}$ | $98.88_{\pm0.06}$ | $97.04_{\pm0.49}$ | $99.26_{\pm0.05}$ | $94.73_{\pm0.28}$ | $98.85_{\pm0.09}$ | 98.07 | $74.39_{\pm1.02}$ | $64.36_{\pm1.08}$ | $70.50_{\pm1.03}$ | 69.75 |

Figure 4: The performance improvement of the proposed method on Likelihood and Log-Rank. Values greater than 0 indicate an enhanced effect.

## 5.1 PERFORMANCE COMPARISON

In this section, we evaluate the enhancement effectiveness of the proposed method in various real-world scenarios, including cross-LLM, cross-domain, detecting mixed machine text, and resisting paraphrase attacks. Details of these scenarios are provided in Appendix D.3.

**Performance across Different LLMs**. Table 2 reports AUROC for detectors trained on GPT4All (Essay) and Llama-2-70b (DetectRL) and evaluated on texts from various LLMs. TPR@FPR-1% results are provided in Table 3 in the Appendix. Besides, results of detectors trained on other LLM texts and on the Reuters dataset appear in Tables 4–15 of the Appendix. The proposed method yields significant gains for nearly all baselines. For example, on Essay, it raises Likelihood from 52.4% to 77.86% (+25.46%). Encouragingly, our method also benefits weak detectors: DetectGPT significantly improves from 0.15% to 37.18% (+37.03%), suggesting that the assumptions underlying its scoring are reasonable and that underperformance mainly stemmed from score estimation error. Finally, detection performance on texts from the same LLM is not always superior to cross-LLM. For example, on Essay, performance on ChatGLM is particularly strong and even exceeds the intra-LLM case of GPT4All, possibly because ChatGLM texts are more discriminative.

**Performance across Different Domains**. We evaluate cross-domain performance in four high-risk domains: arXiv, Writing Prompts, XSum, and Yelp Reviews. Results are summarized in Fig. 4; additional enhanced results for more detectors under cross-domain settings are provided in Fig. 21 and Fig. 22 of the Appendix. In most settings, detectors equipped with our strategy show substantially stronger cross-domain generalization. We attribute this to calibrating the detection score with only a few carefully designed parameters (a 2x2 reward–penalty coefficient matrix), which helps prevent overfitting and thereby improves out-of-domain detection.

**Performance against Mixed Texts**. In practice, human–AI collaboration is pervasive, leading to widespread mixed human–machine text. We therefore evaluate the proposed strategy for mixed-text detection. Two training strategies are considered: training on the original text and training on the mixed text. Results concerning AUROC are shown in Fig. 5, with TPR@FPR-1% shown in Fig. 23 of the Appendix. In most settings, the proposed strategy enhances the detector's ability to recognize mixed-text. Moreover, comparing training on original versus mixed text shows that simply training on mixed text does not improve the detection ability of mixed text, highlighting the focus of mixed text detection research.

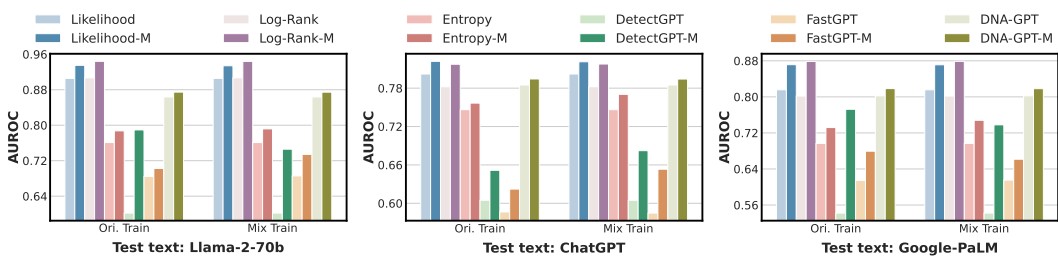

Figure 5: Detection performance concerning AUROC under different LLM mixed texts. All detectors are trained on Llama-2-70b texts.

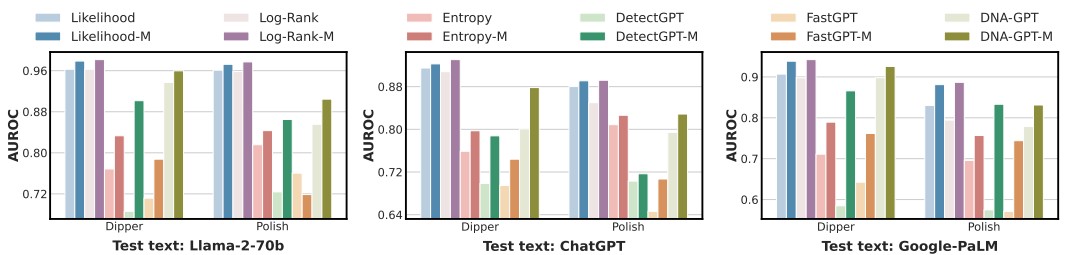

Figure 6: Detection performance concerning AUROC under different paraphrasing texts (Dipper and Polish). All detectors are trained on Llama-2-70b texts.

**Performance against Paraphrasing Attacks**. Prior work (Sadasivan et al., 2023) shows that MGT detection is typically vulnerable to paraphrasing attacks, where an adversary rewrites a passage without altering its semantics to evade detection. We therefore assess the robustness gains of our framework on the Dipper and Polish paraphrase attacks provided by DetectRL. The results in Fig. 6 and Fig. 24 clearly indicate that, even in adversarial settings, our strategy yields encouraging improvements in robustness against paraphrasing attacks.

## 5.2 ABLATION STUDY

We introduce an MRF layer and a positional weighting function to model neighbor similarity and initial instability, respectively. In this section, we verify their effectiveness through ablation studies, denoted as "w/o MRF" and "w/o Pos". Results on the Essay dataset are shown in Fig. 7, with additional results in Fig. 25 and Fig. 26 of the Appendix. We can find that removing either component significantly drops performance, while retaining only one still outperforms the baseline detector, underscoring their designs' rationality. Besides, the contribution of each component varies by detector type. For single-text detectors like Likelihood and Log-Rank, the positional weighting function provides the most improvement by addressing instability in initial scores. However, for methods that aggregate multiple text scores, such as DetectGPT and FastGPT, this instability is already partially mitigated, making the MRF-based score calibration the primary source of gains.

## 5.3 MRF VS. NEURAL NETWORK

This paper proposes a Markov-informed calibration method to model the relationship of detection scores of context tokens. To highlight the rationale of this design, we compare it with methods that directly use neural networks to calibrate scores. A simple three-layer neural network is used here, defined with the "nn" suffix. The comparison results on Likelihood are shown in Fig. 8, and more results can be found in Fig. 31 and Fig. 32 of the Appendix. Although NN-based methods exhibit competitive performance in some settings for intra-LLMs, their generalization ability on cross-LLMs drops significantly. This suggests that it does not truly learn the general ability to correct scores, but rather overfits the training data. In contrast, our strategy shows good generalization.

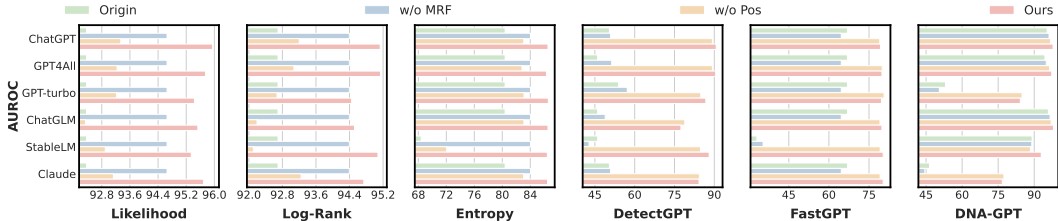

Figure 7: Ablation results on the Essay dataset. The y-axis represents the LLM text on which the detector was trained, and the x-axis represents the average performance across LLMs.

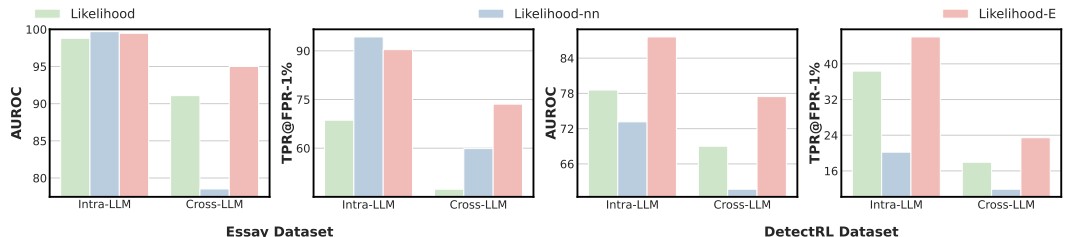

Figure 8: Comparison with NN-based calibration methods. The detector used is Likelihood.

## 6 CONCLUSION

This paper has systematically examined representative metric-based detectors within a unified framework, revealing a core challenge: the inherent randomness of the LLM generation process leads to inaccurate detection scores, and naive aggregation of existing methods fails to fix this. Therefore, we have theoretically and empirically established two key properties of these scores: neighbor similarity and initial instability. Building on these insights, we have proposed a Markov-informed score calibration method that captures token relationships and corrects the biased scores produced by base detectors. Extensive experiments show substantial and consistent performance gains of the proposed method. Admittedly, the proposed method relies on modeling the relationships between context tokens to calibrate detection scores. Consequently, our method is not directly applicable to detection methods that do not provide this fine-grained, token-level output.

## ACKNOWLEDGMENT

This work was supported by the RGC Senior Research Fellow Scheme under the grant: SRFS2324-2S02, RGC Young Collaborative Research Grant No. C2005-24Y, and HKBU CSD Departmental Incentive Scheme.

## ETHICS STATEMENT

This paper proposes a Markov-informed calibration strategy to enhance machine-generated text detection and mitigate the potential risks posed by machine-generated text, including disinformation and phishing. Our work does not involve ethical issues such as dataset releases, potentially harmful insights, potential conflicts of interest and sponsorship, discrimination/bias/fairness concerns, privacy and security issues, legal compliance, and research integrity issues.

## REPRODUCIBILITY STATEMENT

Our code is available at `https://github.com/tmlr-group/MRF_Calibration`, and all datasets used in our experiments (Essay, Reuters, and DetectRL) are publicly available for download. In addition, we provide detailed implementation details in the Appendix, including data partitioning, the fixed seeds, the learning rate, the training batch size, and the two parameters $t_0$ and $T$ introduced by the proposed strategy, to ensure reproducibility of our work.

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

APPENDIX

## A MORE DISCUSSION OF THE PROPOSED METHOD

### A.1 CONTRIBUTION

The contributions of this paper are multifaceted.

- **We provide a unified perspective for understanding metric-based detection methods**. The diversity of these methods (e.g., using metrics based on log-likelihood or log-rank, and introducing perturbed or regenerated text) makes comparison difficult. To this end, we re-examine these methods from three perspectives: data, score calculation, and detection. This analysis provides a precise definition for each method, facilitating comparison and potential improvements. Our analysis shows that these methods employ more reasonable evaluation metrics and incorporate additional contextual information to enhance detection. However, they all fail to address the underlying token-level errors caused by inherent randomness, which limits their detection potential. Therefore, our unified analysis encourages a more nuanced characterization of token-level detection scores, which will provide guidance for future improvements.

- **We reveal the relationships between contextual token detection scores**. While the generative mechanisms of LLMs introduce dependencies between tokens, these relationships remain unclear. To this end, we theoretically reveal two relationships between contextual token scores: neighboring similarity and initial instability, which are further validated through empirical experiments. Constraining these relationships during detection has the potential to mitigate the imprecision in score calculation caused by the inherent randomness of MGTs, which is crucial for the field of MGT detection.

- **We propose a Markov-informed score calibration method to enhance MGT detection**. This involves using Markov random fields to capture the revealed relationships and, through mean-field approximation, modeling the MRF model as a lightweight component that can be stacked on existing detectors to further unlock detection potential. It is worth noting that our main technical contribution and innovation lies not only in the specific implementation but also in the conceptual token-level score calibration. While the current implementation is based on Markov random fields, this is only one possible approach; alternatives include using sequence models or graph neural networks. This conceptual insight can inspire improving MGT detection.

- **Extensive experiments consistently demonstrate the enhanced effectiveness of the proposed strategy**. We empirically verify that it not only excels on a single task but also demonstrates strong capabilities in multiple complex and challenging real-world scenarios, including generalization across LLMs and domains, and robustness to mixed text and paraphrased text. Furthermore, the proposed enhancement component incurs negligible computational overhead compared to the original detector. This combination of effectiveness and efficiency provides a solid foundation for developing practically deployable enhancement solutions for AI-generated text detection.

### A.2 DIFFERENCES FROM EXISTING METHODS

Although some works, such as FourierGPT Xu et al. (2024) and Lastde Xu et al. (2025), also learn local patterns, our approach is quite different:

- **Methodology**. Unlike FourierGPT or Lastde, which propose new standalone metrics, our method is a universal plug-in. We do not replace the metric; we calibrate the intermediate token-level scores of any existing metric (Likelihood, DetectGPT, etc.) using a Markov Random Field.

- **Theory vs. Empiricism**. While many existing local-pattern methods are empirically motivated (e.g., observing spectral power differences and token probability fluctuation), our approach is grounded in the theoretical derivation of attention dynamics (Theorem 1), which formally reveals the Neighbor Similarity and Initial Instability properties we model.

## B RELATED WORK

Existing detection methods can be categorized into active watermark-based methods and passive model-based and metric-based methods.

## B.1 WATERMARK-BASED DETECTION

Watermarking is a proactive defense technique that embeds verifiable information during the text generation stage, thereby enabling simple and reliable detection. RedList (Kirchenbauer et al., 2023) is a model-agnostic watermarking method that dynamically partitions the vocabulary into a "green-list" and "redlist" based on preceding context, slightly increasing the probability of sampling tokens from the greenlist. Subsequent works have made various improvements to this approach. For instance, SemStamp (Hou et al., 2024) introduces a sentence-level semantic hashing watermark to enhance robustness against paraphrasing attacks; DiPmark (Wu et al., 2023b) designs an unbiased watermark that does not alter the original output distribution. REMARK-LLM (Zhang et al., 2024) is a training-based watermarking method that employs a message encoding module to generate an encrypted token distribution for watermark embedding prior to inference. Beyond manually designed watermarks, directly leveraging language models to learn to generate watermarked text is also promising, including training student models (Gu et al., 2023) and semantically invariant watermarking models (Liu & Bu). In addition to the standard binary (0/1 bit) detection of AI-generated content, researchers have also explored multi-bit watermarks (Yoo et al., 2024) for embedding more information.

## B.2 MODEL-BASED DETECTION

Model-based methods represent a classical paradigm in detection, training a binary classifier on a dataset containing both human- and machine-generated texts. A series of works, such as OpenAI Detector (Solaiman et al., 2019), ChatGPT Detector (Guo et al., 2023), GPTZero (GPTZero, 2023), and G3 Detector (Zhan et al., 2023), collect texts generated by various LLMs to train a unified classifier. GPT-Pat (Yu et al., 2023a) finds that detectors trained solely on a single decoding strategy generalize poorly and enhances performance by utilizing mixed decoding strategies. In addition to original data, GLTR (Gehrmann et al., 2019) trains a simple logistic regression classifier by analyzing the predicted ranking of each word within its context. SeqXGPT (Wang et al., 2023) treats the sequence of logits as waveform signals for detection, while Sniffer (Li et al., 2023) uses the difference in logits from different models on the same text as features for detection and attribution. Beyond the data level, recent works have explored more advanced training strategies. For example, CoCo (Liu et al., 2022) introduces graph structures and contrastive learning; LLMDet (Wu et al., 2023a) leverages the perplexity of surrogate models as additional features; MPU (Tian et al., 2024) adopts a positive-unlabeled learning paradigm; and RADAR (Hu et al., 2023) incorporates adversarial training to enhance model robustness. The above methods generally assume a known text source, but when it is unknown, Ghostbuster (Verma et al., 2024) proposes training classifiers directly using texts generated from known surrogate models.

## B.3 METRIC-BASED DETECTION

Metric-based methods do not require training on specific datasets; instead, they directly leverage the inherent statistical biases or intrinsic properties of language model-generated text for distinction. The main advantage of such methods lies in their stronger generalization to new models and domains. Classic approaches in this category include the use of Log-Likelihood (Solaiman et al., 2019), Log-Rank (Mitchell et al., 2023), and Entropy (Gehrmann et al., 2019). DetectGPT (Mitchell et al., 2023) finds that AI-generated text typically lies in regions of negative curvature with respect to the model's log-probability function. By perturbing the text and observing changes in log-probability, it can effectively distinguish AI-generated text. Inspired by DetectGPT, Fast-DetectGPT (Bao et al., 2024) replaces log-probability with conditional probability curvature, significantly improving detection efficiency while maintaining performance. DetectLLM-LRR (Su et al.) proposes using the ratio of log-likelihood to log-rank for detection. Some works, such as DNA-GPT (Yang et al., 2024) and DetectGPT4Code (Yang et al., 2023), detect AI-generated text by comparing discrepancies between the original text and continuations generated by a surrogate model. PHD (Tulchinskii et al., 2024) observes that genuine human-written text possesses higher intrinsic dimensionality after encoder mapping. SimLLM (Nguyen-Son et al., 2024) is based on the observation that the similarity between the original text and its generated continuation is significantly higher than that between generated text and its re-generated version; thus, it estimates the similarity between an input sentence and its generated counterpart for detection. Given that existing methods struggle with out-of-distribution data, token coherence (Ma & Wang, 2024) has been shown to be a reliable metric since

LLM-generated text usually exhibits higher token coherence than human-written text. Yu et al. (Yu et al., 2024) capture the intrinsic features of text by identifying layers with the greatest distributional differences when projecting into the vocabulary space, and using intrinsic rather than semantic features for detection has been demonstrated to yield better results. RepreGuard (Chen et al., 2025) extracted unique activation features from the MGT using a surrogate model, and then the projection scores along the direction of these features are more discriminative. Lastde (Xu et al., 2025) introduced time series analysis to LLM-generated text detection, capturing the temporal dynamics of token probability sequences. FourierGPT (Xu et al., 2024) offered a new perspective on human and model text detection tasks by using relative rather than absolute probability values and by extracting useful features from the probability spectrum. MoSEs (Wu et al., 2025) achieved the quantification of uncertainty in style perception through conditional threshold estimation.

## C    PROOF OF THEOREM 1

**Theorem.** *1 Let $\lambda_K, \lambda_Q, \lambda_V, \lambda_O$ be the largest singular values of parameters $W_K, W_Q, W_V, W_O$, respectively, and let $W = W_V W_O W_Q W_K^\top$. For the transformer defined in Eq. (1), assuming normalized inputs ($\|x_t\|_2 = 1$ for all $t$) and constants $c, \epsilon > 0$, consider $a_t x_{t+1}^\top \geq (1 - \delta) \|a_t\|_2$ with $\delta \leq \left( \frac{c\epsilon}{\lambda_Q \lambda_K \lambda_V \lambda_O} \right)^2$. If $x_\ell$ satisfies $x_\ell W x_\ell^\top \geq c$ and $x_\ell W x_\ell \geq \epsilon^{-1} \max_{j \in [\ell], j \neq \ell} x_j W x_\ell^\top$, then*

$$\alpha_{t+1,l} \leq \frac{\exp\left( C_l \cdot \alpha_{t,l} + \eta \right)}{\exp\left( C_l \cdot \alpha_{t,l} + \eta \right) + \sum_{j \neq l} \exp\left( C_j \cdot \alpha_{t,j} - \eta \right)},$$

$$\alpha_{t+1,l} \geq \frac{\exp\left( C_l \cdot \alpha_{t,l} - \eta \right)}{\exp\left( C_l \cdot \alpha_{t,l} - \eta \right) + \sum_{j \neq l} \exp\left( C_j \cdot \alpha_{t,j} + \eta \right)},$$

*where*

$$C_j = \frac{x_j W x_j^T}{t \, |a_t|_2}, \text{and } \eta = \frac{(1 + \sqrt{2})\epsilon x_j W x_j^T}{(t + 1) \, |a_t|_2}.$$

*Proof.* Our proof follows the proof of existing work (Liu et al., 2023). First, we introduce two necessary lemmas to help our proof.

**Lemma 2** ((Liu et al., 2023)). *Let $x_1, x_2 \in \mathbb{R}^{1 \times m}$ satisfies $\|x_1\|_2 = \|x_2\|_2 = 1$ and $x_1 x_2^\top \geq 1 - \delta$ for some $\delta \in (0, 1)$. Then for all $y \in \mathbb{R}^{1 \times m}$ we have*

$$\left| x_1 y^\top - x_2 y^\top \right| \leq \sqrt{2\delta} \|y\|_2$$

**Lemma 3** ((Liu et al., 2023)). *Let $\ell \in [t]$ be given. Suppose that $x_\ell A x_\ell^\top > \epsilon^{-1} \left| x_j A x_\ell^\top \right|$ for all $j \neq \ell$. Then we have*

$$(\mathcal{S}(t)_\ell - \epsilon) \, x_\ell^\top a x_\ell \leq x_\ell^\top W_K^\top W_Q a_t \leq (\mathcal{S}(t)_\ell + \epsilon) \, x_\ell^\top a x_\ell$$

Based on these two lemmas, we can formally prove the theorem. Let $x_1 = \frac{a_t}{\|a_t\|}$, and $x_2 = x_{t+1}$. If $\frac{a_t x_{t+1}^\top}{\|a_t\|} \geq 1 - \delta$, using the conclusion of Lemma 2, we have

$$\left| \frac{a_t}{(t+1)\|a_t\|_2} W_Q W_K^T x_\ell^T - \frac{1}{t+1} x_{t+1} W_Q W_K^T x_\ell^T \right| \leq \frac{\sqrt{2\delta}}{t+1} \left\| W_Q W_K^T x_\ell^T \right\|_2$$

Since $\lambda_Q$, $\lambda_K$ are the maximum singular values, respectively. Then we have $\left\| W_Q W_K^\top x_\ell^\top \right\|_2 \leq \lambda_Q \lambda_K \|x_\ell\|_2 = \lambda_Q \lambda_K$. This leads to:

$$\left| \frac{a_t}{(t+1)\|a_t\|_2} W_Q W_K^T x_\ell^T - \frac{1}{t+1} x_{t+1} W_Q W_K^T x_\ell^T \right| \leq \frac{\sqrt{2\delta}}{t+1} \lambda_Q \lambda_K \tag{10}$$

Since

$$\|a_t\|_2 = \left\| \left( \sum_{j=1}^{t-1} \alpha_{t,j} x_j \right) W_V W_O \right\| \leq \lambda_O \lambda_V \left\| \sum_{j=1}^{t-1} \alpha_{t,j} x_j \right\|_2 \leq \lambda_O \lambda_V \sum_{j=1}^{t-1} \alpha_{t,j} \|x_j\|_2 = \lambda_O \lambda_V,$$

and the theorem assumes $\delta \leq \left(\frac{c\epsilon}{\lambda_Q \lambda_K \lambda_V \lambda_O}\right)^2$, substituting these into Eq. (10), we can obtain:

$$\frac{\sqrt{2\delta}}{t+1}\lambda_Q \lambda_K \leq \frac{\sqrt{2}c\epsilon}{(t+1)\lambda_V \lambda_O} \leq \frac{\sqrt{2}c\epsilon}{(t+1)\|a_t\|_2} \leq \frac{\sqrt{2}\epsilon}{(t+1)\|a_t\|_2}x_\ell a x_\ell^T \tag{11}$$

The last inequality is obtained from $x_\ell a x_\ell^T \geq c$. Then combining Formula (10) and Formula (11), we have

$$\left|\frac{a_t}{(t+1)\|a_t\|_2}W_Q W_K^T x_\ell^T - \frac{1}{t+1}x_{t+1}W_Q W_K^T x_\ell^T\right| \leq \frac{\sqrt{2}\epsilon}{(t+1)\|a_t\|_2}x_\ell a x_\ell^T$$

From Lemma 3, we have:

$$\left|\frac{a_t W_Q W_K^\top x_\ell^\top}{(t+1)\|a_t\|^2} - \frac{\alpha_{t,\ell}x_\ell W x_\ell^\top}{(t+1)\|a_t\|^2}\right| \leq \frac{\epsilon}{(t+1)\|a_t\|^2}x_\ell^\top a x_\ell$$

By the triangle inequality, we can combine the upper bounds:

$$\begin{aligned}\left|\frac{x_{t+1}W_Q W_K^T x_\ell^T}{t+1} - \frac{\alpha_{t,\ell}x_\ell W x_\ell^T}{(t+1)\|a_t\|_2}\right| &\leq \left|\frac{x_{t+1}W_Q W_K^T x_\ell^T}{t+1} - \frac{a_t W_Q W_K^T x_\ell^T}{(t+1)\|a_t\|_2}\right| \\ &\quad + \left|\frac{a_t W_Q W_K^T x_\ell^T}{(t+1)\|a_t\|_2} - \frac{\alpha_{t,\ell}x_\ell W x_\ell^T}{(t+1)\|a_t\|_2}\right| \\ &\leq \frac{(1+\sqrt{2})\epsilon}{(t+1)\|a_t\|^2}x_\ell^\top a x_\ell\end{aligned} \tag{12}$$

Then, we rearrange the inequality, and we can obtain:

$$\frac{x_\ell W x_\ell^\top}{(t+1)\|a_t\|_2}\left(\alpha_{t,\ell} - (1+\sqrt{2})\epsilon\right) \leq \frac{1}{t+1}x_{t+1}W_Q W_K^\top x_\ell^\top \leq \frac{x_\ell W x_\ell^\top}{(t+1)\|a_t\|_2}\left(\alpha_{t,\ell} + (1+\sqrt{2}\epsilon)\right)$$

Now we give the lower and upper bounds of $\alpha_{t+1,\ell} = \text{softmax}(1/(t+1) \cdot x_{t+1}W_Q W_K^\top X_t^\top)_l$. **Upper bound.** Let $\gamma_{t+1,\ell} = 1/(t+1) \cdot x_{t+1}W_Q W_K^\top x_\ell^\top$. For the softmax function $\alpha_{t+1,l} = \frac{\exp(\gamma_{t+1,\ell})}{\exp(\gamma_{t+1,\ell})+\sum_{k\neq\ell}\exp(\gamma_{t+1,k})}$, to get the maximum value of $\alpha_{t+1,\ell}$, we need to (1) make the numerator as big as possible, which means $\gamma_{t+1,\ell}$ should take its maximum value $\frac{x_\ell W x_\ell^\top}{(t+1)\|a_t\|_2}\left(\alpha_{t,\ell} + (1+\sqrt{2})\epsilon\right)$, (2) make the denominator as small as possible, which means that all other values $\gamma_{t+1,k}$ (when $k \neq \ell$) should be minimized $\frac{x_k W x_k^\top}{(t+1)\|a_t\|_2}\left(\alpha_{t,k} - (1+\sqrt{2})\epsilon\right)$. Therefore,

$$\alpha_{t+1,l} \leq \frac{\frac{x_\ell W x_\ell^\top}{(t+1)\|a_t\|_2}\left(\alpha_{t,\ell} + (1+\sqrt{2})\epsilon\right)}{\frac{x_\ell W x_\ell^\top}{(t+1)\|a_t\|_2}\left(\alpha_{t,\ell} + (1+\sqrt{2})\epsilon\right) + \sum_{k\neq\ell}\frac{x_k W x_k^\top}{(t+1)\|a_t\|_2}\left(\alpha_{t,k} - (1+\sqrt{2})\epsilon\right)}$$

**Lower bound.** Similarly, for the lower bound, we should (1) make the numerator as small as possible, which means $\gamma_{t+1,\ell}$ should take its minimum value $\frac{x_\ell W x_\ell^\top}{(t+1)\|a_t\|_2}\left(\alpha_{t,\ell} - (1+\sqrt{2})\epsilon\right)$, (2) make the denominator as large as possible, which means that all other values $\gamma_{t+1,k}$ (when $k \neq \ell$) should be maximized $\frac{x_k W x_k^\top}{(t+1)\|a_t\|_2}\left(\alpha_{t,k} + (1+\sqrt{2})\epsilon\right)$. Therefore,

$$\alpha_{t+1,l} \geq \frac{\frac{x_\ell W x_\ell^\top}{(t+1)\|a_t\|_2}\left(\alpha_{t,\ell} - (1+\sqrt{2})\epsilon\right)}{\frac{x_\ell W x_\ell^\top}{(t+1)\|a_t\|_2}\left(\alpha_{t,\ell} - (1+\sqrt{2})\epsilon\right) + \sum_{k\neq\ell}\frac{x_k W x_k^\top}{(t+1)\|a_t\|_2}\left(\alpha_{t,k} + (1+\sqrt{2})\epsilon\right)}$$

The proof is completed. $\qquad\square$

# D  EXPERIMENTAL DETAILS

## D.1  DATASETS

The details of the dataset used in the paper are as follows:

- **Essay** (Verma et al., 2024). Each source of this dataset (human-written texts, various LLM-generated texts) contains 1,000 samples. The HGT portion comprises original IvyPanda essays that cover a wide array of subjects and academic levels, from high school through university. For the MGT portion, a tailored prompt was first crafted for each source essay using ChatGPT-turbo, and that prompt was then submitted to several LLMs, including GPT4All, Chat-GPT, ChatGPT-turbo, ChatGLM, Dolly, and Claude, to generate machine-written essays. This workflow produced a diverse set of model-generated texts that remained aligned with the topics of their corresponding source documents.

- **Reuters** (Verma et al., 2024). Built on the Reuters 50–50 authorship benchmark, this dataset contains 1,000 articles from 50 writers, with each author contributing 20 pieces. Replicating the pipeline used for the essay corpus, the team first asked ChatGPT-turbo to invent a headline for every article. Those auto-generated headlines were then embedded into prompts and submitted to multiple LLMs, including ChatGPT, GPT-4, ChatGPT-turbo, ChatGLM, Dolly, and Claude, to create the machine-generated texts.

- **TruthfulQA** (Lin et al., 2022). It contains 817 questions covering 38 categories, including health, law, finance, and politics. The generated answers were produced by several large language models, including GPT4, ChatGPT-turbo, ChatGLM, Dolly, ChatGPT, and StableLM.

- **DetectRL** (Wu et al., 2024). In this dataset, the human-authored portion is drawn from four sources: arXiv abstracts dated 2002–2017, XSum news reports, Writing Prompts stories, and Yelp reviews. These types were chosen because they are especially vulnerable to producing convincing but misleading content when LLMs are misapplied. From each source, 2,800 human texts are selected as HGTs. The machine-generated texts are created using four widely used LLMs—GPT-3.5-turbo (ChatGPT), PaLM-2-bison (Google-PaLM), and Llama-2-70b. The dataset further models practical adversarial settings: (1) a paraphrasing attack that rewrites MGTs with the Dipper paraphraser (Krishna et al., 2023) and Polish paraphraser, and (2) a mixed-text condition where 1/4 of machine-generated sentences is randomly replaced with human-written content while the label remains "machine-generated."

## D.2  BASELINES

A detailed description of the baselines used is shown below:

- **Likelihood** (Solaiman et al., 2019). It uses an LLM to calculate the log probability of each token in a text. The average of these probabilities gives a detection score. A higher score indicates a greater chance that the text was generated by LLMs.

- **Log-Rank** (Gehrmann et al., 2019). Its detection score is created by first using an LLM to rank each token in a text based on its predicted order within a given context. The logarithm of each word's predicted rank is then calculated. The final score is an average of these values, and a lower score is a strong indicator of machine-generated text.

- **Entropy** (Gehrmann et al., 2019). Similar to Log-Rank, it calculates a score for a text by taking the average of each token's conditional entropy within its given context. A lower score suggests a higher likelihood that the text was generated by LLMs.

- **DetectGPT** (Mitchell et al., 2023). It determines if a text is machine-generated by measuring how small changes affect its log probability. The underlying idea is that text created by LLMs is already a high-probability output. So, when it is slightly altered, the new version is likely to have a lower log probability. In contrast, making similar small changes to human-written text does not consistently lower the log probability; it can just as easily stay the same or increase.

- **Fast-DetectGPT (FastGPT)** (Bao et al., 2024). To overcome the major computational expense of DetectGPT, this approach replaces DetectGPT's resource-intensive perturbation step with a more efficient sampling process. It identifies differences in token selection between humans and LLMs using a conditional probability curvature metric.

- **DNA-GPT** (Yang et al., 2024). This method involves a two-step process. First, it cuts a text in half and uses the first part to prompt an LLM to generate a new continuation. Next, it examines the differences between the newly created segment and the original one. This comparison, done via N-gram analysis for black-box models or probability divergence for white-box models, reveals a clear distinction between how humans and machines generate text.

- **Repreguard** (Chen et al., 2025). It utilizes key feature directions determined by a proxy model to project and score the representation of the text under test, comparing the result against a threshold. This approach demonstrates extremely high detection accuracy and robustness when dealing with out-of-distribution data and various types of attacks.

- **Lastde** (Xu et al., 2025). It introduces time series analysis into machine-generated text detection, overcoming the limitations of traditional methods that ignore local discriminative information by collaboratively modeling the local dynamic features and global statistical indicators of token probability sequences.

- **FourierGPT** (Xu et al., 2024). It proposes a detection method based on a likelihood spectrum perspective, which captures subtle differences between human and machine language by analyzing the relative changes in text likelihood values rather than their absolute values.

- **Binoculars** Hans et al. (2024). It is a detection algorithm that requires no training data, accurately distinguishing between human and machine-generated text by comparing the score differences of a pair of pre-trained LLMs.

### D.3 EXPERIMENTAL SCENARIO

To extensively evaluate the effectiveness of the proposed enhancement model, we conduct experiments in the following real-world scenarios:

- **Cross-LLM**. To assess how well the proposed model works across different LLMs, we trained detectors on a single LLM's text and then tested it on a variety of LLMS' texts. The main body of the paper presents the results from training detectors on the GPT4All texts (Essay dataset) and Llama-2-70b texts (DetectRL), and then testing them on various LLMs, as shown in Table 2. Complete results for every training and testing combination can be found in Appendix E.3 (Tables 3-15).

- **Cross-Domain**. The DetectRL dataset includes texts from four distinct domains: arXiv academic abstracts, XSum news articles, Writing Prompts stories, and Yelp Reviews. We utilized this dataset to evaluate the model's performance across various domains. To achieve this, we trained the detector on one domain and then tested it on the other domains. For this evaluation, all machine-generated texts were created using the default PaLM model. The heatmaps in Figs. 4, 21, and 22 illustrate the performance improvement achieved by our enhancement model compared to various baseline detection models.

- **Paraphrasing Attack**. Studies have shown that MGT detection is vulnerable to paraphrase attacks. Therefore, this scenario is used to evaluate the robustness of the MGT detection method. Using the DetectRL dataset, which includes data from the Polish and Dipper paraphraser, we trained our detector on clean, original texts and then evaluated its robustness on these paraphrased texts. Specifically, we trained the detector using clean texts from Llama-2-70b and then tested it on paraphrased texts from several different LLMs. The results can be found in Fig. 6 and 24.

- **Mixed Text**. Because a blend of human and machine-generated text is common in the real world, we use the mixed texts provided by DetectRL for evaluation. It involved randomly swapping out 25% of the sentences in an LLM-generated text with human-written ones. We conducted two separate experiments on this dataset: (1) The detector was trained on pure, non-mixed text and then tested for its ability to detect the mixed texts. (2) The detector was both trained and tested on the mixed texts themselves. The performance of the detector in these mixed settings is shown in Figs. 5 and 23. In each sub-figure, the detectors trained on original text are shown on the left, and those trained on mixed text are shown on the right.

### D.4 THREAT MODEL AND PROXY SETTINGS

We conducted the experiments in a black-box setting:

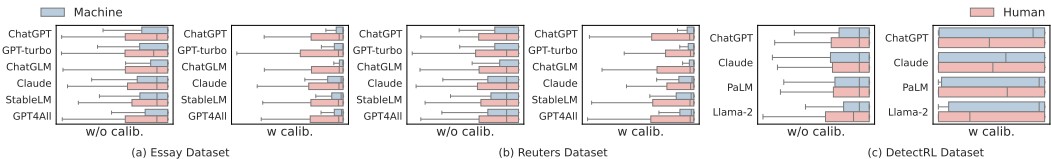

Figure 9: Distribution of token scores obtained by the Log-Likelihood method without and with enhancement. It can be observed that the proposed method enhances the discriminative nature of the token scores.

Figure 10: Distribution of token scores obtained by the Log-Rank method without and with enhancement. It can be observed that the proposed method enhances the discriminative nature of the token scores.

- **Threat Model**. We operate under a strictly Black-Box setting regarding the target LLM. We assume that the detector is entirely unknown to the LLM that generates the text (e.g., Chat-GPT, Claude). Therefore, we employ a Proxy Model to compute token-level metrics (e.g., Log-Likelihood and Rank) for the candidate text, which are reasonable due to the transferability of the extracted metrics.

- **Proxy Models**. For all baselines, we use GPT2 as the proxy model. Additionally, we use GPT2-XL as the scoring model for Fast-DetectGPT. To learn the 2x2 parameter $W_{mrf}$ introduced by our strategy, we perform training based on various LLM texts, as explicitly stated in the "Training Text" of the table. For example, in Table 2 in the paper, we learn $W_{mrf}$ on GPT4All texts of the Essay dataset and Llama-2-70b texts of the DetectRL dataset, and then test all candidate texts (from ChatGPT, Claude, or Dolly, etc.) using the learned detector.

## D.5 PARAMETER SETTINGS

We conducted five independent experiments to ensure the consistency of our results, using five fixed random seeds (1-5). For all datasets, we used 10% of the data for training, while the remaining 90% was split evenly between validation and testing. To ensure a fair comparison, the enhanced models shared the same hyperparameters as their base models. In our enhancement model, there are two hyperparameters: the transition center $t_0$ in the positional weighting function $\beta(t)$ and the number of iterations $T$ in the MRF layer. By default, we set $t_0 = 30$ and $T = 10$ for the enhanced versions of all detectors across the three datasets, highlighting the flexibility of our approach. For training the MRF layer, we use a learning rate of 0.05 and train for 10 epochs. Hyperparameter sensitivity analyses are provided in Appendix E.9.

## E  MORE EXPERIMENTAL RESULS

### E.1  MORE RESULTS OF TOKEN SCORE DISTRIBUTION BEFORE AND AFTER ENHANCEMENT

In addition to the partial results on DetectGPT presented in the main text, we also present the complete results about token-level detection score distributions for Log-Likelihood, Log-Rank, Entropy, and DetectGPT in Figs. 9, 10, 11, and 12. The results are similar to those in the main text: the original detector's scores show substantial overlap between human- and machine-generated text. However, after calibration using our proposed augmentation strategy, the scores achieve significantly improved discriminability.

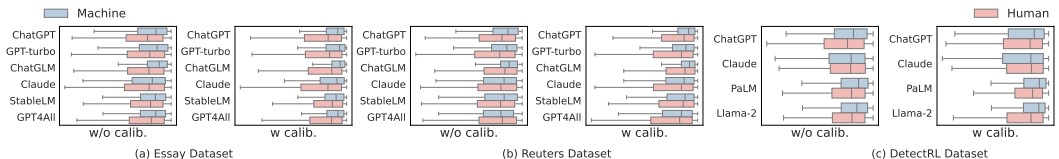

Figure 11: Distribution of token scores obtained by the Entropy method without and with enhancement. It can be observed that the proposed method enhances the discriminative nature of the token scores.

Figure 12: Distribution of token scores obtained by the DetectGPT method without and with enhancement. It can be observed that the proposed method enhances the discriminative nature of the token scores.

### E.2 MORE RESULTS OF CONTEXT TOKEN RELATIONSHIPS

In the main text, we demonstrated the existence of neighbor similarity (Fig. 2) and initial instability (Fig. 2) in token-level detection scores through experiments on the Essay dataset. Here, we provide additional supplementary results to further validate these relationships.

To verify neighbor similarity, we provide additional results using Entropy and DetectGPT detection scores on the Essay dataset (Fig. 13), as well as results on the Reuters and DetectRL datasets (Figs. 14 and 15). In addition, we provide Chinese texts on the CUDRT dataset (Fig. 16). These supplementary experiments consistently show that the closer the tokens are, the more similar their detection scores are.

Similarly, to verify initial instability, we provide additional results on the Essay dataset (Fig. 17), as well as results on the Reuters and DetectRL datasets (Figs. 18 and 19). In addition, we provide Chinese texts on the CUDRT dataset (Fig. 20). These results again demonstrate that token scores at the beginning of a text fluctuate significantly before gradually stabilizing.

### E.3 MORE PERFORMANCE COMPARISON

**Performance across Different LLMs**. In the main text, we evaluated the cross-LLM performance of detectors trained on GPT4All (Essay) and Llama-2-70b (DetectRL) on various LLM texts in terms of AUROC, as shown in Table 2. This section aims to provide more comprehensive supplementary experimental results, including: (1) cross-LLM performance performance under the same experimental settings in terms of TPR@FPR-1% (Table 3), and (2) cross-LLM performance comparisons of detectors trained on other LLMs on the Essay, DetectRL, and Reuters datasets (Tables 4 to 15). These extensive experimental results are consistent with the conclusions of the main paper. Among the 888 cross-LLM evaluation settings, our proposed enhanced model achieved better performance than the original detector in 91.4% of the cases, highlighting the generalization ability and application value of this method on different models and datasets.

**Performance across Different Domains**. In addition to the cross-domain performance improvements for Log-Likelihood and Log-Rank demonstrated in the main text, this section provides additional results for other detectors. Specifically, it includes improvements to the cross-domain performance of Entropy and DetectGPT (Fig. 21), as well as improvements to FastGPT and DNA-GPT (Fig. 22). Combining all experimental results, we reach the same conclusion as in the main text: in most experimental settings, detectors applied with our strategy significantly improve their cross-domain generalization capabilities.

**Performance against Mixed Texts**. In addition to the AUROC performance comparison for mixed texts presented in the main text, this section provides additional performance comparisons

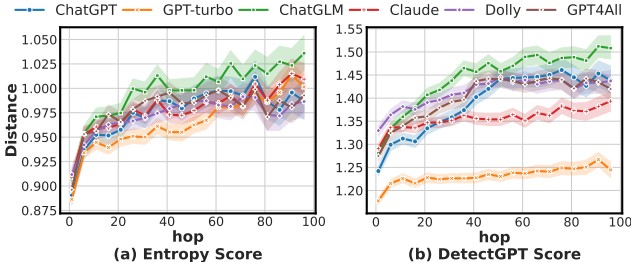

Figure 13: The detection score distances (Mean Absolute Difference) of neighbors at different hops in the Essay dataset. Entropy and DetectGPT score are used here.

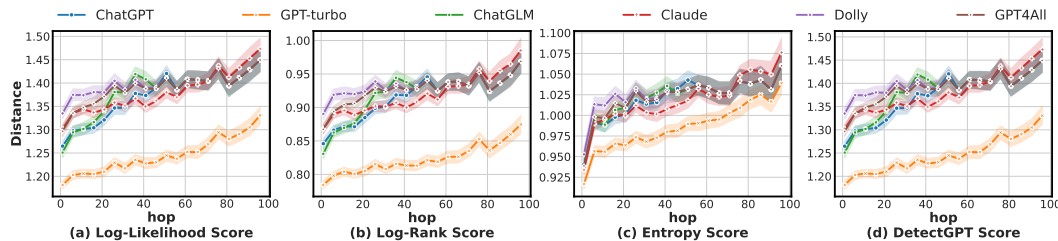

Figure 14: The detection score distances (Mean Absolute Difference) of neighbors at different hops in the Reuters dataset. Log-Likelihood, Log-Rank, Entropy, and DetectGPT score are used here.

at TPR@FPR-1%, as shown in Fig. 23. We reach consistent conclusions with those in the main text: in most experimental settings, the detector equipped with the proposed strategy significantly improves its ability to detect mixed text.

**Performance against Paraphrasing Attacks**. In addition to the AUROC performance comparison for paraphrasing texts presented in the main text, this section also presents a performance comparison under the TPR@FPR-1% metric, as shown in Fig. 24. We find that in most experimental settings, the detector applying our proposed strategy significantly improves its robustness against paraphrasing attacks, which is consistent with the conclusions drawn in the main text.

### E.4 MORE PERFORMANCE COMPARISON ON THE QA DATASET

In this section, we explore testing on a challenging QA dataset, specifically the TruthfulQA dataset (Lin et al., 2022), which covers Q&A tasks in 38 domains (e.g., health, law, and finance). The results are shown in Table 16. As can be seen, our method consistently boosts performance. For example, DNA-GPT-M achieves a significant leap from 80.17% to 88.32% (+8.15%) in average AUROC.

### E.5 MORE PERFORMANCE COMPARISON ON SHORT-TEXT

In this section, we performed a stress test on the Essay dataset with a strict 50-word limit. In Table 17, our calibration remains highly effective even with limited context. For example, we improve FastGPT by 5.35% and DNA-GPT by 3.03% on average. This confirms our calibration still holds even in short sequences.

### E.6 ENHANCEMENT EXPERIMENTS ON MORE DETECTORS

In this section, we integrated more state-of-the-art metric-based baselines: RepreGuard Chen et al. (2025), Binoculars Hans et al. (2024), Lastde Xu et al. (2025), and FourierGPT Xu et al. (2024), and applied our strategy to them as "E" suffix. As shown in Tables 18-20, our method functions as a universal plug-in, enhancing these detectors. For example, on the Reuters dataset, our method improves RepreGuard by 8.97% and Binoculars by 4.42%.

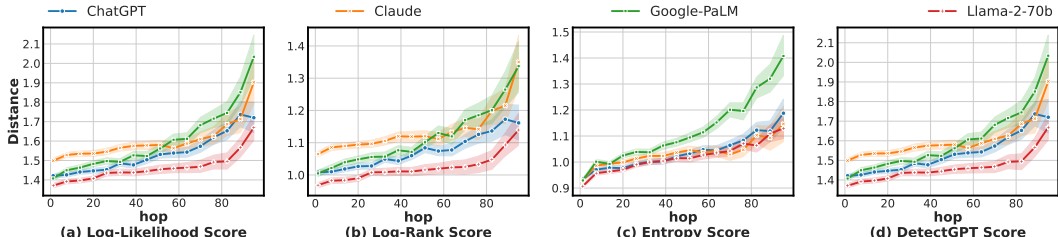

Figure 15: The detection score distances (Mean Absolute Difference) of neighbors at different hops in the DetectRL dataset. Log-Likelihood, Log-Rank, Entropy, and DetectGPT score are used here.

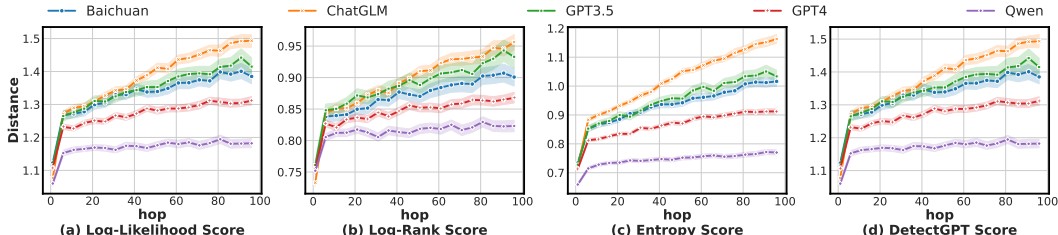

Figure 16: The detection score distances (Mean Absolute Difference) of neighbors at different hops in the CUDRT dataset. Log-Likelihood, Log-Rank, Entropy, and DetectGPT score are used here.

## E.7 PERFORMANCE COMPARISON AGAINST PERTURBATION TEXTS

In this section, we evaluated the robustness against perturbed texts. Here, we chose two distinct attack methods in the DetectRL dataset: Back-Translation and Word Perturbation. The results are shown in Tables 21. Across both perturbation scenarios, detectors equipped with our Markov-informed calibration (e.g., Likelihood-M, Log-Rank-M) consistently outperformed the uncalibrated ones, typically by margins of 10-20%. These findings align perfectly with our existing evaluation on Paraphrasing Attacks, which shows that our method effectively mitigates the performance degradation typically caused by text perturbations.

These encouraging gains are consistent with the fundamental theory of Markov random fields. The adversarial perturbation acts as token-level salt-and-pepper noise, while our Markov-informed calibration strategy (modeling Neighbor Similarity) leverages the remaining uncorrupted context to smooth out these local anomalies, recovering the latent detection signal.

## E.8 MORE RESULTS OF ABLATION STUDY

In addition to the ablation experiments on the position weight function and MRF layer presented in the main text for the Essay dataset, we also provide ablation results for the Reuters and DetectRL datasets, as shown in Figs. 25 and 26. These experimental results are consistent with the conclusions of the main text: removing either component leads to a significant performance drop; even retaining only one of them outperforms the baseline detector, strongly demonstrating the effectiveness of both components.

## E.9 SENSITIVITY ANALYSIS

**Sensitivity w.r.t. transition center $t_0$.** In our experiments, we set the default transition center $t_0$ of the position weighting function to 30. This section examines the effect of varying $t_0$ values on detection performance. The AUROC and TPR@FPR-1% results are shown in Figs. 27 and 28, respectively. The experimental results show that detection performance steadily improves with increasing $t_0$ values, which is consistent with the conclusions of the ablation experiments and demonstrates the effectiveness of the position weighting function. However, performance improvement is not infinite. When $t_0$ values are too large, performance gradually saturates or even declines. This is likely because excessively large $t_0$ filters out useful token scores. Therefore, a trade-off is necessary. Through sensitivity analysis, we recommend setting $t_0$ values between 20 and 30.

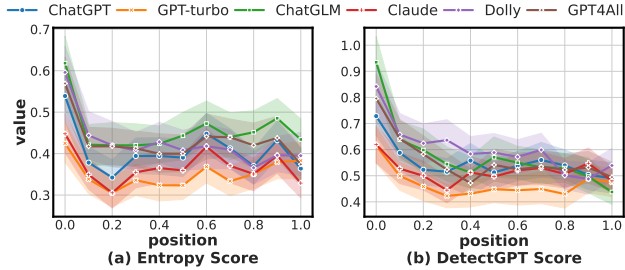

Figure 17: The score distances (Mean Absolute Difference) of 1-hop neighbors at different positions in the Essay dataset. Entropy and DetectGPT score are used here.

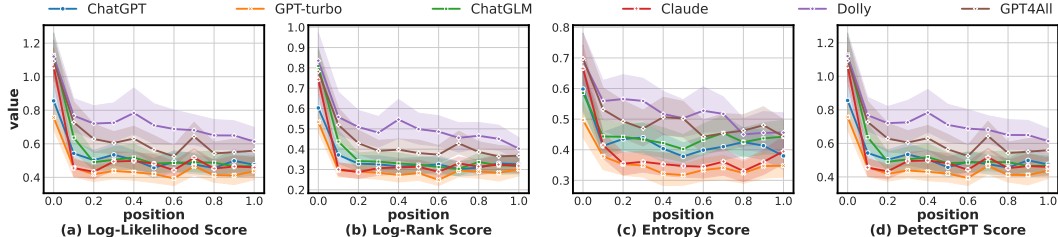

Figure 18: The score distances (Mean Absolute Difference) of 1-hop neighbors at different positions in the Reuters dataset. Log-Likelihood, Log-Rank, Entropy, and DetectGPT score are used here.

**Sensitivity w.r.t. iterations $T$ in MRF layer**. We compute the posterior probability of the Markov random field using a multi-step iterative approach. To this end, we evaluate the impact of varying the number of iterations on detection performance, as shown in Figs. 29 and 30. The experimental results demonstrate that multi-step iterative computation significantly enhances detection performance compared to single-step computation, underscoring its importance in score calibration. However, performance may degrade with increasing the number of iterations, possibly due to oversmoothing of the detection scores. Based on our sensitivity analysis, we recommend setting the number of iterations to 10.

### E.10 MORE RESULTS COMPARING WITH NEURAL NETWORK CALIBRATION

In the main text, we demonstrated, through experimental results using the Log-Likelihood score, that while neural network-based detection score correction performs well on the intra-LLM, its performance drops sharply on the cross-LLM. This section further presents comparative results using other detection scores, as shown in Figs. 31 and 32. Similar experimental findings further demonstrate that this method does not truly learn universal score correction capabilities, but rather overfits the training data. This strongly emphasizes the superiority and rationality of our proposed Markov-informed score calibration method.

### E.11 MRF VS. HMM

In the paper, we chose the MRF over Markov chain for two key reasons:

- **Theoretical rationality**. A Markov chain typically models unidirectional dependencies (current state depends on previous state). However, in MGT detection, we identified the Neighbor Similarity property, which is inherently bidirectional. An MRF naturally captures these bidirectional dependencies, whereas a standard Markov Chain struggles to utilize future context during inference without complex multi-pass algorithms.

- **Computational Efficiency**. A raw MRF may be complicated. However, as described in Section 4.2, our Mean-Field Approximation transforms the problem into lightweight iterative matrix multiplications. This allows for massive parallelization on GPUs. In contrast, exact inference in HMMs (e.g., Forward-Backward algorithm) is inherently sequential, which can be slower on modern hardware despite having similar theoretical complexity $\mathcal{O}(N)$.

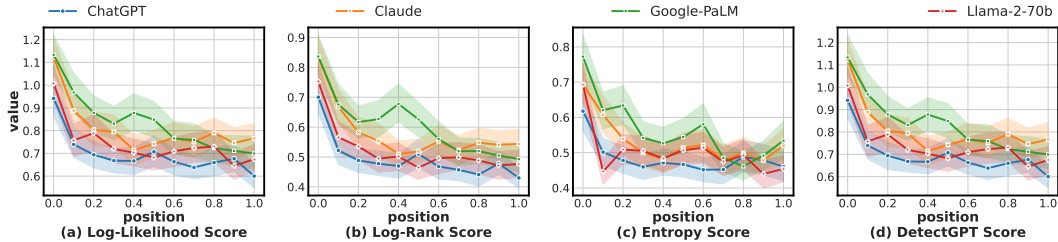

Figure 19: The score distances (Mean Absolute Difference) of 1-hop neighbors at different positions in the DetectRL dataset. Log-Likelihood, Log-Rank, Entropy, and DetectGPT score are used here.

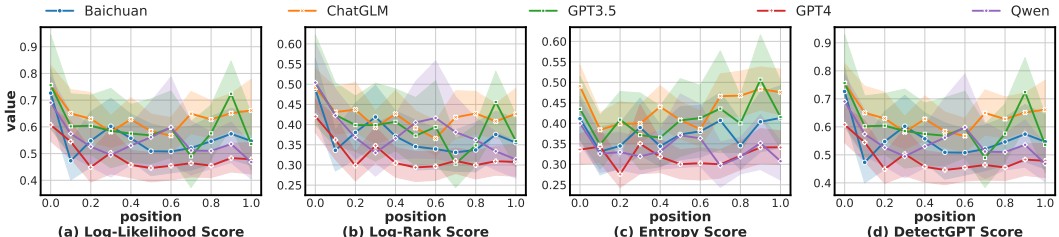

Figure 20: The score distances (Mean Absolute Difference) of 1-hop neighbors at different positions in the CUDRT dataset. Log-Likelihood, Log-Rank, Entropy, and DetectGPT score are used here.

To further verify, we implemented a Hidden Markov Model (HMM) based calibration strategy and applied it to existing detectors ("HMM" suffix). As shown in Table 22 (left part), while the HMM improves upon the original baseline, it consistently underperforms our MRF approach. This validates the necessity of modeling bidirectional dependencies. For computational efficiency (see the right part of Table 22), our MRF implementation is actually faster in practice due to GPU-friendly matrix operations.

### E.12 COMPARISON WITH ENHANCED METHOD

TOCSIN is an enhanced metric-based approach which enhances the detector by dynamically adjusting the decision threshold. Clearly, it operates on a different dimension than our method. Therefore, in addition to supplementing TOCSIN-enhanced baselines (suffix "-T"), our Markov calibration can be further applied to the top of TOCSIN (suffix "-TM") to further explore its detection potential, as shown in Table 23. It can be seen that our method further improves the performance of TOCSIN-enhanced baselines in most cases. This significant gain suggests that TOCSIN (adjusting dynamic thresholds using token cohesiveness score) and our method (calibration token-level detection score) focus on different aspects, achieving complementarity.

### E.13 ANALYSIS OF CROSS-DOMAIN FAILURE CASES

Aside from a few random performance fluctuations, most failures occur when xsum (training data) is applied to arXiv, writing, and yelp (test data). Given that our proposed calibration strategy hinges on learning a reward/penalty matrix ($W_{mrf}$) to encourage neighbor similarity, we examined the distance of neighbor detection score across various domains, as summarized in Fig. 33.

As observed, xsum exhibits considerably lower neighbor similarity (larger distance), whereas the other domains show higher similarity. This makes sense because, as a news summarization dataset, XSum is extremely information-dense, with a highly compressed syntactic structure and a lack of redundancy and transitional words seen in longer texts (such as arXiv). Consequently, adjacent tokens in XSum experience more drastic fluctuations in detection scores. Therefore, when trained on xsum, the MRF learns parameters ($W_{mrf}$) that tolerate high local variance (treating it as a normal feature of the domain). When these parameters are applied to "smoother" domains like arXiv, the MRF fails to enforce the tighter consistency constraints required for those domains, leading to the observed performance drop. Encouragingly, compared to these occasional failure modes, the extensive enhancements highlight the practicality of our method.

Table 3: Performance concerning TPR@FPR-1% (%) on Essay (left) and DetectRL (right).

| Method | Essay (Training Text: GPT4All) | | | | | | | DetectRL (Training Text: Llama-2-70b) | | | |
|---|---|---|---|---|---|---|---|---|---|---|---|
| | GPT4All | ChatGPT | Dolly | ChatGLM | Claude | ChatGPT-turbo | Avg. | Llama-2-70b | ChatGPT | Google-PaLM | Avg. |
| Likelihood | $46.33_{\pm16.49}$ | $68.62_{\pm13.32}$ | $20.67_{\pm10.79}$ | $92.86_{\pm4.84}$ | $12.31_{\pm6.29}$ | $73.60_{\pm14.63}$ | 52.4 | $38.37_{\pm0.92}$ | $10.21_{\pm1.40}$ | $25.66_{\pm2.41}$ | 24.75 |
| **Likelihood-M** | $\mathbf{87.47}_{\pm3.42}$ | $\mathbf{90.36}_{\pm2.93}$ | $\mathbf{55.47}_{\pm5.03}$ | $\mathbf{97.19}_{\pm1.11}$ | $\mathbf{40.40}_{\pm7.87}$ | $\mathbf{96.27}_{\pm1.27}$ | **77.86** | $\mathbf{46.03}_{\pm1.70}$ | $\mathbf{10.51}_{\pm3.92}$ | $\mathbf{36.37}_{\pm2.31}$ | **30.97** |
| Log-Rank | $62.69_{\pm13.36}$ | $79.07_{\pm7.89}$ | $25.11_{\pm8.35}$ | $95.71_{\pm2.05}$ | $19.20_{\pm8.26}$ | $80.89_{\pm10.73}$ | 60.44 | $42.05_{\pm0.84}$ | $12.44_{\pm1.71}$ | $22.84_{\pm1.28}$ | 25.78 |
| **Log-Rank-M** | $\mathbf{87.38}_{\pm2.50}$ | $\mathbf{88.89}_{\pm2.84}$ | $\mathbf{52.12}_{\pm4.18}$ | $\mathbf{98.04}_{\pm0.43}$ | $\mathbf{31.69}_{\pm4.15}$ | $\mathbf{94.22}_{\pm2.46}$ | **75.39** | $\mathbf{50.56}_{\pm1.44}$ | $\mathbf{13.94}_{\pm2.36}$ | $\mathbf{41.51}_{\pm2.69}$ | **35.34** |
| Entropy | $2.73_{\pm0.69}$ | $7.16_{\pm3.17}$ | $2.29_{\pm1.36}$ | $6.88_{\pm3.05}$ | $3.91_{\pm2.24}$ | $13.24_{\pm7.60}$ | 6.04 | $2.03_{\pm0.73}$ | $0.25_{\pm0.16}$ | $6.95_{\pm0.78}$ | 3.07 |
| **Entropy-M** | $\mathbf{15.63}_{\pm0.87}$ | $\mathbf{41.24}_{\pm2.25}$ | $\mathbf{16.95}_{\pm3.12}$ | $\mathbf{40.85}_{\pm4.37}$ | $\mathbf{23.20}_{\pm1.70}$ | $\mathbf{49.91}_{\pm5.02}$ | **31.30** | $\mathbf{2.69}_{\pm0.39}$ | $\mathbf{0.67}_{\pm0.10}$ | $\mathbf{9.17}_{\pm1.10}$ | **4.18** |
| DetectGPT | $0.00_{\pm0.00}$ | $0.00_{\pm0.00}$ | $0.00_{\pm0.00}$ | $0.00_{\pm0.00}$ | $0.89_{\pm0.24}$ | $0.00_{\pm0.00}$ | 0.15 | $4.15_{\pm0.40}$ | $2.94_{\pm0.62}$ | $1.71_{\pm0.47}$ | 2.93 |
| **DetectGPT-M** | $\mathbf{41.23}_{\pm29.53}$ | $\mathbf{46.98}_{\pm25.36}$ | $\mathbf{18.38}_{\pm20.21}$ | $\mathbf{41.83}_{\pm44.80}$ | $\mathbf{8.80}_{\pm9.64}$ | $\mathbf{65.87}_{\pm21.66}$ | **37.18** | $\mathbf{32.41}_{\pm9.58}$ | $\mathbf{7.59}_{\pm4.05}$ | $\mathbf{25.14}_{\pm4.53}$ | **21.71** |
| FastGPT | $1.59_{\pm0.32}$ | $1.64_{\pm0.67}$ | $0.29_{\pm0.10}$ | $2.72_{\pm0.68}$ | $3.47_{\pm1.83}$ | $16.89_{\pm6.66}$ | 4.43 | $11.35_{\pm2.01}$ | $3.78_{\pm1.20}$ | $5.02_{\pm0.85}$ | 6.72 |
| **FastGPT-M** | $\mathbf{23.92}_{\pm10.29}$ | $\mathbf{30.00}_{\pm9.20}$ | $\mathbf{32.89}_{\pm16.52}$ | $\mathbf{55.54}_{\pm7.80}$ | $0.67_{\pm0.56}$ | $16.44_{\pm31.78}$ | **26.58** | $\mathbf{15.23}_{\pm3.23}$ | $\mathbf{6.28}_{\pm1.72}$ | $\mathbf{13.52}_{\pm5.63}$ | **11.68** |
| DNA-GPT | $57.68_{\pm3.34}$ | $63.24_{\pm6.54}$ | $22.20_{\pm3.85}$ | $84.82_{\pm4.14}$ | $16.67_{\pm3.47}$ | $56.27_{\pm6.45}$ | 50.15 | $36.56_{\pm1.56}$ | $19.70_{\pm0.52}$ | $30.38_{\pm2.75}$ | 28.88 |
| **DNA-GPT-M** | $\mathbf{93.85}_{\pm0.97}$ | $\mathbf{88.44}_{\pm4.22}$ | $\mathbf{56.42}_{\pm10.82}$ | $\mathbf{96.83}_{\pm0.41}$ | $\mathbf{33.33}_{\pm7.90}$ | $\mathbf{86.22}_{\pm6.52}$ | **75.85** | $\mathbf{42.52}_{\pm1.26}$ | $19.11_{\pm2.04}$ | $\mathbf{34.81}_{\pm1.54}$ | **32.15** |

Table 4: Performance on Essay dataset. The detection models are trained on text generated by ChatGPT.

| Metric | Method | GPT4All | ChatGPT | Dolly | ChatGLM | Claude | ChatGPT-turbo | avg |
|---|---|---|---|---|---|---|---|---|
| TPR@FPR-1% | Likelihood | $46.24_{\pm16.60}$ | $68.62_{\pm13.32}$ | $20.67_{\pm10.79}$ | $92.81_{\pm4.82}$ | $12.31_{\pm6.29}$ | $73.60_{\pm14.63}$ | 52.38 |
| | **Likelihood-M** | $\mathbf{89.38}_{\pm4.05}$ | $\mathbf{92.00}_{\pm2.84}$ | $\mathbf{59.19}_{\pm5.98}$ | $\mathbf{97.10}_{\pm0.69}$ | $\mathbf{42.22}_{\pm6.32}$ | $\mathbf{96.80}_{\pm1.23}$ | **79.45** |
| | Log-Rank | $62.69_{\pm13.36}$ | $79.07_{\pm7.89}$ | $25.11_{\pm8.35}$ | $95.71_{\pm2.05}$ | $19.16_{\pm8.22}$ | $80.89_{\pm10.73}$ | 60.44 |
| | **Log-Rank-M** | $\mathbf{87.15}_{\pm2.62}$ | $\mathbf{88.80}_{\pm2.81}$ | $\mathbf{52.12}_{\pm4.32}$ | $\mathbf{98.04}_{\pm0.43}$ | $\mathbf{31.33}_{\pm4.10}$ | $\mathbf{94.22}_{\pm2.46}$ | **75.28** |
| | Entropy | $2.73_{\pm0.69}$ | $7.16_{\pm3.17}$ | $2.29_{\pm1.36}$ | $6.88_{\pm3.05}$ | $3.91_{\pm2.24}$ | $13.24_{\pm7.60}$ | 6.04 |
| | **Entropy-M** | $\mathbf{12.76}_{\pm2.18}$ | $\mathbf{34.84}_{\pm5.15}$ | $\mathbf{14.27}_{\pm2.63}$ | $\mathbf{36.96}_{\pm2.81}$ | $\mathbf{19.78}_{\pm3.76}$ | $\mathbf{43.20}_{\pm6.33}$ | **26.97** |
| | DetectGPT | $0.00_{\pm0.00}$ | $0.00_{\pm0.00}$ | $0.00_{\pm0.00}$ | $0.00_{\pm0.00}$ | $0.84_{\pm0.51}$ | $2.36_{\pm2.52}$ | 0.53 |
| | **DetectGPT-M** | $\mathbf{41.09}_{\pm30.67}$ | $\mathbf{42.04}_{\pm25.95}$ | $\mathbf{9.55}_{\pm9.01}$ | $\mathbf{39.02}_{\pm38.26}$ | $\mathbf{10.13}_{\pm10.28}$ | $\mathbf{68.49}_{\pm23.15}$ | **35.05** |
| | FastGPT | $1.59_{\pm0.32}$ | $1.64_{\pm0.67}$ | $0.29_{\pm0.10}$ | $2.72_{\pm0.68}$ | $3.47_{\pm1.83}$ | $16.89_{\pm6.66}$ | 4.43 |
| | **FastGPT-M** | $\mathbf{19.73}_{\pm11.16}$ | $\mathbf{26.18}_{\pm12.42}$ | $\mathbf{24.82}_{\pm20.23}$ | $\mathbf{45.76}_{\pm21.65}$ | $3.38_{\pm5.34}$ | $\mathbf{26.04}_{\pm32.58}$ | **24.32** |
| | DNA-GPT | $40.77_{\pm8.33}$ | $73.69_{\pm3.78}$ | $19.95_{\pm7.42}$ | $82.59_{\pm11.71}$ | $18.04_{\pm3.53}$ | $63.82_{\pm6.62}$ | 49.81 |
| | **DNA-GPT-M** | $\mathbf{89.02}_{\pm4.14}$ | $\mathbf{95.07}_{\pm1.33}$ | $\mathbf{65.97}_{\pm4.57}$ | $\mathbf{99.24}_{\pm0.36}$ | $\mathbf{39.56}_{\pm4.01}$ | $\mathbf{94.62}_{\pm1.43}$ | **80.58** |
| AUROC | Likelihood | $96.16_{\pm0.30}$ | $98.79_{\pm0.19}$ | $90.90_{\pm1.33}$ | $99.29_{\pm0.25}$ | $92.76_{\pm0.23}$ | $99.13_{\pm0.19}$ | 96.17 |
| | **Likelihood-M** | $\mathbf{98.77}_{\pm0.10}$ | $\mathbf{99.58}_{\pm0.14}$ | $\mathbf{94.80}_{\pm0.87}$ | $\mathbf{99.56}_{\pm0.19}$ | $\mathbf{94.69}_{\pm0.42}$ | $\mathbf{99.77}_{\pm0.14}$ | **97.86** |
| | Log-Rank | $96.55_{\pm0.31}$ | $98.95_{\pm0.13}$ | $90.08_{\pm1.28}$ | $99.36_{\pm0.13}$ | $92.01_{\pm0.20}$ | $99.24_{\pm0.15}$ | 96.03 |
| | **Log-Rank-M** | $\mathbf{98.56}_{\pm0.06}$ | $\mathbf{99.40}_{\pm0.09}$ | $\mathbf{93.81}_{\pm0.92}$ | $\mathbf{99.56}_{\pm0.08}$ | $\mathbf{92.88}_{\pm0.29}$ | $\mathbf{99.64}_{\pm0.10}$ | **97.31** |
| | Entropy | $74.19_{\pm1.62}$ | $89.49_{\pm0.33}$ | $73.26_{\pm1.48}$ | $84.11_{\pm0.77}$ | $86.58_{\pm0.66}$ | $95.94_{\pm0.35}$ | 83.93 |
| | **Entropy-M** | $\mathbf{83.33}_{\pm0.66}$ | $\mathbf{93.24}_{\pm0.15}$ | $\mathbf{81.17}_{\pm1.04}$ | $\mathbf{91.38}_{\pm0.37}$ | $\mathbf{88.28}_{\pm0.49}$ | $\mathbf{96.86}_{\pm0.21}$ | **89.04** |
| | DetectGPT | $50.15_{\pm0.99}$ | $50.53_{\pm3.65}$ | $48.04_{\pm7.27}$ | $49.25_{\pm1.58}$ | $51.21_{\pm8.39}$ | $55.34_{\pm31.67}$ | 50.75 |
| | **DetectGPT-M** | $\mathbf{94.98}_{\pm3.26}$ | $\mathbf{96.35}_{\pm2.60}$ | $\mathbf{85.82}_{\pm6.13}$ | $\mathbf{94.20}_{\pm4.48}$ | $\mathbf{84.06}_{\pm6.47}$ | $\mathbf{98.58}_{\pm1.24}$ | **92.33** |
| | FastGPT | $64.63_{\pm1.53}$ | $67.68_{\pm1.70}$ | $47.17_{\pm1.53}$ | $71.08_{\pm1.51}$ | $75.31_{\pm0.90}$ | $88.62_{\pm0.67}$ | 69.08 |
| | **FastGPT-M** | $\mathbf{86.68}_{\pm3.28}$ | $\mathbf{90.82}_{\pm3.86}$ | $\mathbf{72.23}_{\pm23.91}$ | $\mathbf{93.53}_{\pm3.86}$ | $65.24_{\pm12.36}$ | $71.87_{\pm21.80}$ | **80.06** |
| | DNA-GPT | $96.28_{\pm0.32}$ | $98.87_{\pm0.21}$ | $92.85_{\pm0.83}$ | $99.32_{\pm0.27}$ | $91.65_{\pm0.72}$ | $98.51_{\pm0.31}$ | 96.25 |
| | **DNA-GPT-M** | $\mathbf{99.32}_{\pm0.13}$ | $\mathbf{99.79}_{\pm0.07}$ | $\mathbf{97.42}_{\pm0.56}$ | $\mathbf{99.90}_{\pm0.06}$ | $\mathbf{95.65}_{\pm0.51}$ | $\mathbf{99.76}_{\pm0.07}$ | **98.64** |

### E.14 COMPARISON WITH MODEL-BASED DETECTORS

As shown in Fig. 34, we compare the enhanced versions of metric-based methods with model-based detection methods, including ChatGPT-D and MPU. Experimental results show that while model-based methods demonstrate superior performance on the DetectRL dataset, they underperform state-of-the-art metric-based methods, such as DNA-GPT, on the Essay and Reuters datasets. Notably, the significant performance gap between model-based methods in intra-LLM and cross-LLM further confirms their increased risk of overfitting to the training data. This observation is consistent with our intention of focusing on metric-based detection methods.

### E.15 RUNNING TIME

Table 24 shows the training and inference runtimes on different datasets. As discussed in the main text, our proposed Markov-based score refinement module can be implemented in constant time via sparse-dense matrix multiplication. Therefore, the additional time overhead introduced by this module is negligible compared to the time-consuming score calculation, highlighting the flexibility and practicality of our approach in practical applications.

## F THE USE OF LARGE LANGUAGE MODELS

In our paper, we used LLMs to polish the language and correct grammatical errors. LLMs were not used to generate novel research ideas, design experiments, analyze results, or write substantive

Table 5: Performance on Essay dataset. The detection models are trained on text generated by ChatGPT-turbo.

| Metric | Method | GPT4All | ChatGPT | Dolly | ChatGLM | Claude | ChatGPT-turbo | avg |
|---|---|---|---|---|---|---|---|---|
| TPR@FPR-1% | Likelihood | $46.29_{\pm16.55}$ | $68.62_{\pm13.32}$ | $20.67_{\pm10.79}$ | $92.86_{\pm4.84}$ | $12.31_{\pm6.29}$ | $73.60_{\pm14.63}$ | 52.39 |
| | **Likelihood-M** | $\mathbf{87.61}_{\pm3.74}$ | $\mathbf{88.31}_{\pm3.64}$ | $\mathbf{61.29}_{\pm7.12}$ | $\mathbf{94.20}_{\pm2.40}$ | $\mathbf{41.11}_{\pm7.93}$ | $\mathbf{95.38}_{\pm2.27}$ | **77.98** |
| | Log-Rank | $62.69_{\pm13.36}$ | $79.07_{\pm7.89}$ | $25.11_{\pm8.35}$ | $95.71_{\pm2.05}$ | $19.16_{\pm8.22}$ | $80.89_{\pm10.73}$ | 60.44 |
| | **Log-Rank-M** | $\mathbf{86.70}_{\pm0.59}$ | $\mathbf{87.47}_{\pm3.17}$ | $\mathbf{54.61}_{\pm4.05}$ | $93.75_{\pm3.55}$ | $\mathbf{35.91}_{\pm2.65}$ | $\mathbf{94.13}_{\pm1.68}$ | **75.43** |
| | Entropy | $2.73_{\pm0.69}$ | $7.16_{\pm3.17}$ | $2.29_{\pm1.36}$ | $6.88_{\pm3.05}$ | $3.91_{\pm2.24}$ | $13.24_{\pm7.60}$ | 6.04 |
| | **Entropy-M** | $\mathbf{11.62}_{\pm0.59}$ | $\mathbf{30.67}_{\pm6.02}$ | $\mathbf{13.56}_{\pm1.59}$ | $\mathbf{32.54}_{\pm2.63}$ | $\mathbf{17.33}_{\pm5.32}$ | $\mathbf{39.91}_{\pm8.11}$ | **24.27** |
| | DetectGPT | $0.00_{\pm0.00}$ | $0.00_{\pm0.00}$ | $0.00_{\pm0.00}$ | $0.00_{\pm0.00}$ | $0.53_{\pm0.52}$ | $3.07_{\pm1.96}$ | 0.60 |
| | **DetectGPT-M** | $\mathbf{14.03}_{\pm13.72}$ | $\mathbf{17.07}_{\pm8.79}$ | $\mathbf{3.05}_{\pm2.54}$ | $\mathbf{6.43}_{\pm3.80}$ | $\mathbf{5.11}_{\pm3.98}$ | $\mathbf{44.71}_{\pm11.72}$ | **15.07** |
| | FastGPT | $1.59_{\pm0.32}$ | $1.64_{\pm0.67}$ | $0.29_{\pm0.10}$ | $2.72_{\pm0.68}$ | $3.47_{\pm1.83}$ | $16.89_{\pm6.66}$ | 4.43 |
| | **FastGPT-M** | $\mathbf{8.97}_{\pm4.52}$ | $\mathbf{15.33}_{\pm15.63}$ | $\mathbf{0.72}_{\pm0.60}$ | $\mathbf{20.09}_{\pm18.21}$ | $\mathbf{7.51}_{\pm6.83}$ | $\mathbf{49.82}_{\pm22.08}$ | **17.07** |
| | DNA-GPT | $0.73_{\pm0.33}$ | $0.76_{\pm0.30}$ | $2.63_{\pm0.81}$ | $0.13_{\pm0.11}$ | $21.64_{\pm3.45}$ | $0.00_{\pm0.00}$ | 4.31 |
| | **DNA-GPT-M** | $\mathbf{39.41}_{\pm2.68}$ | $\mathbf{38.62}_{\pm2.34}$ | $\mathbf{39.47}_{\pm2.11}$ | $\mathbf{38.04}_{\pm2.49}$ | $\mathbf{39.38}_{\pm1.86}$ | $\mathbf{38.67}_{\pm2.19}$ | **38.93** |
| AUROC | Likelihood | $96.16_{\pm0.30}$ | $98.79_{\pm0.19}$ | $90.90_{\pm1.33}$ | $99.29_{\pm0.25}$ | $92.76_{\pm0.23}$ | $99.13_{\pm0.19}$ | 96.17 |
| | **Likelihood-M** | $\mathbf{98.65}_{\pm0.21}$ | $\mathbf{99.39}_{\pm0.20}$ | $\mathbf{93.95}_{\pm1.15}$ | $99.13_{\pm0.43}$ | $\mathbf{94.66}_{\pm0.39}$ | $\mathbf{99.77}_{\pm0.12}$ | **97.59** |
| | Log-Rank | $96.55_{\pm0.31}$ | $98.95_{\pm0.13}$ | $90.08_{\pm1.28}$ | $99.36_{\pm0.13}$ | $92.01_{\pm0.20}$ | $99.24_{\pm0.15}$ | 96.03 |
| | **Log-Rank-M** | $\mathbf{98.37}_{\pm0.21}$ | $\mathbf{99.17}_{\pm0.29}$ | $\mathbf{92.96}_{\pm1.51}$ | $99.17_{\pm0.37}$ | $\mathbf{92.96}_{\pm0.43}$ | $\mathbf{99.62}_{\pm0.11}$ | **97.04** |
| | Entropy | $74.19_{\pm1.62}$ | $89.49_{\pm0.34}$ | $73.26_{\pm1.48}$ | $84.11_{\pm0.77}$ | $86.58_{\pm0.66}$ | $95.94_{\pm0.35}$ | 83.93 |
| | **Entropy-M** | $\mathbf{82.88}_{\pm0.85}$ | $\mathbf{93.08}_{\pm0.29}$ | $\mathbf{80.90}_{\pm0.89}$ | $\mathbf{91.18}_{\pm0.37}$ | $\mathbf{88.57}_{\pm0.54}$ | $\mathbf{96.90}_{\pm0.24}$ | **88.92** |
| | DetectGPT | $49.19_{\pm0.58}$ | $53.60_{\pm0.77}$ | $42.52_{\pm0.84}$ | $49.59_{\pm1.70}$ | $58.46_{\pm0.60}$ | $82.10_{\pm1.25}$ | 55.91 |
| | **DetectGPT-M** | $\mathbf{91.65}_{\pm2.28}$ | $\mathbf{93.38}_{\pm2.31}$ | $\mathbf{77.58}_{\pm5.63}$ | $\mathbf{90.54}_{\pm2.74}$ | $\mathbf{79.82}_{\pm9.52}$ | $\mathbf{97.61}_{\pm0.92}$ | **88.43** |
| | FastGPT | $64.63_{\pm1.53}$ | $67.68_{\pm1.70}$ | $47.17_{\pm1.53}$ | $71.08_{\pm1.51}$ | $75.31_{\pm0.90}$ | $88.62_{\pm0.67}$ | 69.08 |
| | **FastGPT-M** | $\mathbf{85.05}_{\pm3.63}$ | $\mathbf{86.41}_{\pm10.08}$ | $\mathbf{49.85}_{\pm11.80}$ | $\mathbf{89.08}_{\pm6.67}$ | $\mathbf{75.84}_{\pm17.10}$ | $\mathbf{96.80}_{\pm2.60}$ | **80.51** |
| | DNA-GPT | $53.28_{\pm1.23}$ | $43.65_{\pm0.76}$ | $61.90_{\pm0.39}$ | $40.46_{\pm1.26}$ | $67.53_{\pm0.55}$ | $43.31_{\pm0.94}$ | 51.69 |
| | **DNA-GPT-M** | $\mathbf{83.94}_{\pm1.09}$ | $\mathbf{84.02}_{\pm1.06}$ | $\mathbf{83.93}_{\pm1.20}$ | $\mathbf{83.89}_{\pm1.10}$ | $\mathbf{84.23}_{\pm1.07}$ | $\mathbf{84.07}_{\pm1.06}$ | **84.01** |

Table 6: Performance on Essay dataset. The detection models are trained on text generated by ChatGLM.

| Metric | Method | GPT4All | ChatGPT | Dolly | ChatGLM | Claude | ChatGPT-turbo | avg |
|---|---|---|---|---|---|---|---|---|
| TPR@FPR-1% | Likelihood | $46.29_{\pm16.54}$ | $68.62_{\pm13.32}$ | $20.67_{\pm10.79}$ | $92.81_{\pm4.82}$ | $12.31_{\pm6.29}$ | $73.60_{\pm14.63}$ | 52.38 |
| | **Likelihood-M** | $\mathbf{78.82}_{\pm5.30}$ | $\mathbf{87.87}_{\pm3.22}$ | $\mathbf{46.73}_{\pm10.16}$ | $\mathbf{98.71}_{\pm1.11}$ | $\mathbf{31.64}_{\pm10.61}$ | $\mathbf{93.78}_{\pm3.41}$ | **72.92** |
| | Log-Rank | $62.69_{\pm13.36}$ | $79.07_{\pm7.89}$ | $25.11_{\pm8.35}$ | $95.71_{\pm2.05}$ | $19.20_{\pm8.26}$ | $80.89_{\pm10.73}$ | 60.44 |
| | **Log-Rank-M** | $\mathbf{81.32}_{\pm7.05}$ | $\mathbf{86.76}_{\pm3.68}$ | $\mathbf{47.35}_{\pm8.02}$ | $\mathbf{98.62}_{\pm0.26}$ | $\mathbf{25.24}_{\pm7.46}$ | $\mathbf{90.62}_{\pm5.22}$ | **71.65** |
| | Entropy | $2.73_{\pm0.69}$ | $7.16_{\pm3.17}$ | $2.29_{\pm1.36}$ | $6.88_{\pm3.05}$ | $3.91_{\pm2.24}$ | $13.24_{\pm7.60}$ | 6.04 |
| | **Entropy-M** | $\mathbf{12.76}_{\pm1.21}$ | $\mathbf{32.09}_{\pm4.82}$ | $\mathbf{13.99}_{\pm3.01}$ | $\mathbf{35.62}_{\pm4.65}$ | $\mathbf{18.40}_{\pm4.10}$ | $\mathbf{41.42}_{\pm6.98}$ | **25.71** |
| | DetectGPT | $0.00_{\pm0.00}$ | $0.00_{\pm0.00}$ | $0.00_{\pm0.00}$ | $0.00_{\pm0.00}$ | $0.89_{\pm0.24}$ | $0.00_{\pm0.00}$ | 0.15 |
| | **DetectGPT-M** | $\mathbf{18.36}_{\pm0.94}$ | $\mathbf{17.87}_{\pm5.87}$ | $\mathbf{20.67}_{\pm3.81}$ | $\mathbf{60.98}_{\pm6.42}$ | $\mathbf{4.89}_{\pm2.75}$ | $\mathbf{6.36}_{\pm2.01}$ | **21.52** |
| | FastGPT | $1.59_{\pm0.32}$ | $1.64_{\pm0.67}$ | $0.29_{\pm0.10}$ | $2.72_{\pm0.68}$ | $3.47_{\pm1.83}$ | $16.89_{\pm6.66}$ | 4.43 |
| | **FastGPT-M** | $\mathbf{23.87}_{\pm10.38}$ | $\mathbf{29.87}_{\pm9.03}$ | $\mathbf{32.89}_{\pm16.52}$ | $\mathbf{55.49}_{\pm7.87}$ | $\mathbf{0.67}_{\pm0.56}$ | $\mathbf{16.22}_{\pm31.34}$ | **26.50** |
| | DNA-GPT | $57.54_{\pm4.18}$ | $69.91_{\pm9.92}$ | $39.19_{\pm4.23}$ | $97.32_{\pm0.76}$ | $31.24_{\pm5.75}$ | $65.16_{\pm10.11}$ | 60.06 |
| | **DNA-GPT-M** | $\mathbf{93.58}_{\pm1.29}$ | $\mathbf{92.53}_{\pm3.44}$ | $\mathbf{66.54}_{\pm9.55}$ | $\mathbf{99.55}_{\pm0.20}$ | $\mathbf{44.36}_{\pm9.19}$ | $\mathbf{91.47}_{\pm4.41}$ | **81.34** |
| AUROC | Likelihood | $96.16_{\pm0.30}$ | $98.79_{\pm0.19}$ | $90.90_{\pm1.33}$ | $99.29_{\pm0.25}$ | $92.76_{\pm0.23}$ | $99.13_{\pm0.19}$ | 96.17 |
| | **Likelihood-M** | $\mathbf{98.03}_{\pm0.26}$ | $\mathbf{99.22}_{\pm0.20}$ | $\mathbf{93.10}_{\pm1.43}$ | $\mathbf{99.65}_{\pm0.07}$ | $\mathbf{93.47}_{\pm1.14}$ | $\mathbf{99.54}_{\pm0.16}$ | **97.17** |
| | Log-Rank | $96.55_{\pm0.31}$ | $98.95_{\pm0.13}$ | $90.08_{\pm1.28}$ | $99.36_{\pm0.13}$ | $92.01_{\pm0.20}$ | $99.24_{\pm0.15}$ | 96.03 |
| | **Log-Rank-M** | $\mathbf{98.14}_{\pm0.30}$ | $\mathbf{99.20}_{\pm0.22}$ | $\mathbf{92.86}_{\pm1.66}$ | $\mathbf{99.54}_{\pm0.10}$ | $\mathbf{91.39}_{\pm1.08}$ | $\mathbf{99.53}_{\pm0.16}$ | **96.78** |
| | Entropy | $74.19_{\pm1.62}$ | $89.49_{\pm0.33}$ | $73.26_{\pm1.48}$ | $84.11_{\pm0.77}$ | $86.58_{\pm0.66}$ | $95.94_{\pm0.35}$ | 83.93 |
| | **Entropy-M** | $\mathbf{82.99}_{\pm1.01}$ | $\mathbf{93.13}_{\pm0.23}$ | $\mathbf{81.03}_{\pm1.11}$ | $\mathbf{91.15}_{\pm0.28}$ | $\mathbf{88.45}_{\pm0.44}$ | $\mathbf{96.88}_{\pm0.18}$ | **88.94** |
| | DetectGPT | $50.81_{\pm0.58}$ | $46.40_{\pm0.77}$ | $57.48_{\pm0.84}$ | $50.41_{\pm1.70}$ | $41.54_{\pm0.60}$ | $17.90_{\pm1.25}$ | 44.09 |
| | **DetectGPT-M** | $\mathbf{80.39}_{\pm1.33}$ | $\mathbf{80.53}_{\pm0.58}$ | $\mathbf{80.67}_{\pm1.98}$ | $\mathbf{94.02}_{\pm0.56}$ | $\mathbf{62.63}_{\pm1.24}$ | $\mathbf{67.25}_{\pm1.24}$ | **77.58** |
| | FastGPT | $64.63_{\pm1.53}$ | $67.68_{\pm1.70}$ | $47.17_{\pm1.53}$ | $71.08_{\pm1.51}$ | $75.31_{\pm0.90}$ | $88.62_{\pm0.67}$ | 69.08 |
| | **FastGPT-M** | $\mathbf{87.19}_{\pm3.46}$ | $\mathbf{91.55}_{\pm3.32}$ | $\mathbf{82.58}_{\pm18.04}$ | $\mathbf{95.35}_{\pm0.49}$ | $\mathbf{59.28}_{\pm1.90}$ | $\mathbf{63.48}_{\pm18.12}$ | **79.91** |
| | DNA-GPT | $96.97_{\pm0.20}$ | $98.09_{\pm0.47}$ | $94.11_{\pm0.47}$ | $99.87_{\pm0.04}$ | $92.84_{\pm0.71}$ | $97.85_{\pm0.37}$ | 96.62 |
| | **DNA-GPT-M** | $\mathbf{99.37}_{\pm0.05}$ | $\mathbf{99.58}_{\pm0.10}$ | $\mathbf{97.68}_{\pm0.29}$ | $\mathbf{99.92}_{\pm0.02}$ | $\mathbf{96.24}_{\pm0.47}$ | $\mathbf{99.54}_{\pm0.09}$ | **98.72** |

Table 7: Performance on Essay dataset. The detection models are trained on text generated by Dolly.

| Metric | Method | GPT4All | ChatGPT | Dolly | ChatGLM | Claude | ChatGPT-turbo | avg |
|---|---|---|---|---|---|---|---|---|
| TPR@FPR-1% | Likelihood | $46.29_{\pm16.54}$ | $68.62_{\pm13.32}$ | $20.67_{\pm10.79}$ | $92.86_{\pm4.84}$ | $12.31_{\pm6.29}$ | $73.60_{\pm14.63}$ | 52.39 |
| | **Likelihood-M** | $\textbf{89.02}_{\pm1.68}$ | $\textbf{88.09}_{\pm4.50}$ | $\textbf{57.14}_{\pm5.55}$ | $\textbf{97.59}_{\pm1.09}$ | $\textbf{34.36}_{\pm13.54}$ | $\textbf{94.36}_{\pm3.67}$ | 76.76 |
| | Log-Rank | $62.69_{\pm13.36}$ | $79.07_{\pm7.89}$ | $25.11_{\pm8.35}$ | $95.71_{\pm2.05}$ | $19.16_{\pm8.22}$ | $80.89_{\pm10.73}$ | 60.44 |
| | **Log-Rank-M** | $\textbf{87.70}_{\pm2.60}$ | $\textbf{89.02}_{\pm2.13}$ | $\textbf{51.65}_{\pm3.99}$ | $\textbf{98.12}_{\pm0.54}$ | $\textbf{31.69}_{\pm3.32}$ | $\textbf{94.36}_{\pm1.93}$ | 75.42 |
| | Entropy | $2.73_{\pm0.69}$ | $7.16_{\pm3.17}$ | $2.29_{\pm1.36}$ | $6.88_{\pm3.05}$ | $3.91_{\pm2.24}$ | $13.24_{\pm7.60}$ | 6.04 |
| | **Entropy-M** | $\textbf{11.94}_{\pm1.85}$ | $\textbf{32.09}_{\pm4.77}$ | $\textbf{13.22}_{\pm4.31}$ | $\textbf{34.78}_{\pm8.04}$ | $\textbf{18.49}_{\pm3.20}$ | $\textbf{40.62}_{\pm6.34}$ | 25.19 |
| | DetectGPT | $0.00_{\pm0.00}$ | $0.00_{\pm0.00}$ | $0.00_{\pm0.00}$ | $0.00_{\pm0.00}$ | $0.89_{\pm0.24}$ | $0.00_{\pm0.00}$ | 0.15 |
| | **DetectGPT-M** | $\textbf{32.89}_{\pm22.87}$ | $\textbf{46.09}_{\pm19.75}$ | $\textbf{24.44}_{\pm9.98}$ | $\textbf{68.84}_{\pm21.48}$ | $\textbf{6.04}_{\pm4.69}$ | $\textbf{48.09}_{\pm25.20}$ | 37.73 |
| | FastGPT | $0.82_{\pm0.31}$ | $0.00_{\pm0.00}$ | $0.38_{\pm0.42}$ | $0.09_{\pm0.11}$ | $0.00_{\pm0.00}$ | $0.00_{\pm0.00}$ | 0.22 |
| | **FastGPT-M** | $\textbf{27.24}_{\pm2.77}$ | $\textbf{28.04}_{\pm7.40}$ | $\textbf{39.43}_{\pm3.59}$ | $\textbf{55.89}_{\pm5.30}$ | $\textbf{0.98}_{\pm0.65}$ | $\textbf{0.84}_{\pm0.71}$ | 25.41 |
| | DNA-GPT | $58.95_{\pm4.76}$ | $64.00_{\pm6.45}$ | $48.64_{\pm4.04}$ | $84.29_{\pm3.64}$ | $29.24_{\pm3.40}$ | $59.02_{\pm8.29}$ | 57.36 |
| | **DNA-GPT-M** | $\textbf{88.70}_{\pm1.78}$ | $\textbf{87.82}_{\pm1.27}$ | $\textbf{72.60}_{\pm4.17}$ | $\textbf{92.59}_{\pm0.38}$ | $\textbf{44.71}_{\pm4.67}$ | $\textbf{86.93}_{\pm2.13}$ | 78.89 |
| AUROC | Likelihood | $96.16_{\pm0.30}$ | $98.79_{\pm0.19}$ | $90.90_{\pm1.33}$ | $99.29_{\pm0.25}$ | $92.76_{\pm0.24}$ | $99.13_{\pm0.19}$ | 96.17 |
| | **Likelihood-M** | $\textbf{98.63}_{\pm0.29}$ | $\textbf{99.48}_{\pm0.16}$ | $\textbf{94.44}_{\pm0.71}$ | $\textbf{99.62}_{\pm0.08}$ | $\textbf{94.49}_{\pm0.73}$ | $\textbf{99.70}_{\pm0.15}$ | 97.73 |
| | Log-Rank | $96.55_{\pm0.31}$ | $98.95_{\pm0.13}$ | $90.08_{\pm1.28}$ | $99.36_{\pm0.13}$ | $92.01_{\pm0.20}$ | $99.24_{\pm0.15}$ | 96.03 |
| | **Log-Rank-M** | $\textbf{98.55}_{\pm0.08}$ | $\textbf{99.40}_{\pm0.09}$ | $\textbf{93.78}_{\pm0.09}$ | $\textbf{99.56}_{\pm0.09}$ | $\textbf{92.81}_{\pm0.38}$ | $\textbf{99.65}_{\pm0.10}$ | 97.29 |
| | Entropy | $74.19_{\pm1.62}$ | $89.49_{\pm0.33}$ | $73.26_{\pm1.48}$ | $84.11_{\pm0.77}$ | $86.58_{\pm0.66}$ | $95.94_{\pm0.35}$ | 83.93 |
| | **Entropy-M** | $\textbf{83.09}_{\pm0.84}$ | $\textbf{93.15}_{\pm0.17}$ | $\textbf{81.05}_{\pm1.14}$ | $\textbf{91.26}_{\pm0.24}$ | $\textbf{88.42}_{\pm0.52}$ | $\textbf{96.88}_{\pm0.20}$ | 88.97 |
| | DetectGPT | $50.81_{\pm0.58}$ | $46.40_{\pm0.77}$ | $57.48_{\pm0.84}$ | $50.41_{\pm1.70}$ | $41.54_{\pm0.60}$ | $17.90_{\pm1.25}$ | 44.09 |
| | **DetectGPT-M** | $\textbf{93.62}_{\pm5.70}$ | $\textbf{93.95}_{\pm6.36}$ | $\textbf{90.01}_{\pm3.69}$ | $\textbf{97.82}_{\pm1.98}$ | $\textbf{81.42}_{\pm9.59}$ | $\textbf{92.20}_{\pm11.34}$ | 91.50 |
| | FastGPT | $35.37_{\pm1.53}$ | $32.32_{\pm1.70}$ | $52.83_{\pm1.53}$ | $28.92_{\pm1.51}$ | $24.69_{\pm0.90}$ | $11.38_{\pm0.67}$ | 30.92 |
| | **FastGPT-M** | $\textbf{88.35}_{\pm0.89}$ | $\textbf{89.67}_{\pm1.11}$ | $\textbf{91.55}_{\pm0.28}$ | $\textbf{95.27}_{\pm0.38}$ | $\textbf{59.65}_{\pm1.54}$ | $\textbf{54.61}_{\pm2.30}$ | 79.85 |
| | DNA-GPT | $94.50_{\pm0.07}$ | $93.95_{\pm0.19}$ | $95.91_{\pm0.49}$ | $95.04_{\pm0.15}$ | $89.35_{\pm0.39}$ | $93.81_{\pm0.20}$ | 93.76 |
| | **DNA-GPT-M** | $\textbf{98.23}_{\pm0.21}$ | $\textbf{97.67}_{\pm0.16}$ | $\textbf{98.01}_{\pm0.33}$ | $\textbf{98.00}_{\pm0.14}$ | $\textbf{94.63}_{\pm0.40}$ | $\textbf{97.64}_{\pm0.17}$ | 97.37 |

Table 8: Performance on Essay dataset. The detection models are trained on text generated by Claude.

| Metric | Method | GPT4All | ChatGPT | Dolly | ChatGLM | Claude | ChatGPT-turbo | avg |
|---|---|---|---|---|---|---|---|---|
| TPR@FPR-1% | Likelihood | $46.20_{\pm16.47}$ | $68.58_{\pm13.28}$ | $20.67_{\pm10.79}$ | $92.86_{\pm4.84}$ | $12.31_{\pm6.29}$ | $73.60_{\pm14.63}$ | 52.37 |
| | **Likelihood-M** | $\textbf{86.74}_{\pm3.37}$ | $\textbf{90.09}_{\pm2.80}$ | $\textbf{50.07}_{\pm9.80}$ | $\textbf{98.08}_{\pm0.27}$ | $\textbf{34.00}_{\pm6.66}$ | $\textbf{95.78}_{\pm1.40}$ | 75.79 |
| | Log-Rank | $62.69_{\pm13.36}$ | $79.07_{\pm7.89}$ | $25.11_{\pm8.35}$ | $95.71_{\pm2.05}$ | $19.20_{\pm8.26}$ | $80.89_{\pm10.73}$ | 60.44 |
| | **Log-Rank-M** | $\textbf{89.20}_{\pm2.33}$ | $\textbf{88.18}_{\pm2.58}$ | $\textbf{54.42}_{\pm3.96}$ | $95.00_{\pm3.56}$ | $\textbf{37.82}_{\pm2.28}$ | $\textbf{94.80}_{\pm1.37}$ | 76.57 |
| | Entropy | $2.73_{\pm0.69}$ | $7.16_{\pm3.17}$ | $2.29_{\pm1.36}$ | $6.88_{\pm3.05}$ | $3.91_{\pm2.24}$ | $13.24_{\pm7.60}$ | 6.04 |
| | **Entropy-M** | $\textbf{12.16}_{\pm0.86}$ | $\textbf{29.07}_{\pm3.22}$ | $\textbf{10.74}_{\pm4.12}$ | $\textbf{30.94}_{\pm5.62}$ | $\textbf{16.80}_{\pm2.73}$ | $\textbf{39.69}_{\pm4.99}$ | 23.23 |
| | DetectGPT | $0.00_{\pm0.00}$ | $0.00_{\pm0.00}$ | $0.00_{\pm0.00}$ | $0.00_{\pm0.00}$ | $0.84_{\pm0.51}$ | $2.36_{\pm2.52}$ | 0.53 |
| | **DetectGPT-M** | $\textbf{34.62}_{\pm17.61}$ | $\textbf{38.71}_{\pm23.27}$ | $\textbf{24.73}_{\pm8.38}$ | $\textbf{74.06}_{\pm17.99}$ | $\textbf{3.91}_{\pm2.05}$ | $\textbf{29.11}_{\pm24.44}$ | 34.19 |
| | FastGPT | $1.59_{\pm0.32}$ | $1.64_{\pm0.67}$ | $0.29_{\pm0.10}$ | $2.72_{\pm0.68}$ | $3.47_{\pm1.83}$ | $16.89_{\pm6.66}$ | 4.43 |
| | **FastGPT-M** | $\textbf{27.47}_{\pm2.92}$ | $\textbf{28.04}_{\pm7.40}$ | $\textbf{39.57}_{\pm3.76}$ | $\textbf{56.47}_{\pm6.13}$ | $\textbf{0.98}_{\pm0.65}$ | $\textbf{0.84}_{\pm0.71}$ | 25.56 |
| | DNA-GPT | $0.50_{\pm0.17}$ | $0.31_{\pm0.11}$ | $0.67_{\pm0.55}$ | $0.00_{\pm0.00}$ | $0.13_{\pm0.11}$ | $0.53_{\pm0.36}$ | 0.36 |
| | **DNA-GPT-M** | $\textbf{12.76}_{\pm2.00}$ | $\textbf{13.16}_{\pm2.36}$ | $\textbf{14.03}_{\pm3.12}$ | $\textbf{12.14}_{\pm1.73}$ | $\textbf{13.69}_{\pm2.59}$ | $\textbf{13.33}_{\pm2.38}$ | 13.19 |
| AUROC | Likelihood | $96.16_{\pm0.30}$ | $98.79_{\pm0.19}$ | $90.90_{\pm1.33}$ | $99.29_{\pm0.25}$ | $92.76_{\pm0.23}$ | $99.13_{\pm0.19}$ | 96.17 |
| | **Likelihood-M** | $\textbf{98.52}_{\pm0.24}$ | $\textbf{99.45}_{\pm0.09}$ | $\textbf{94.32}_{\pm0.95}$ | $\textbf{99.63}_{\pm0.09}$ | $\textbf{94.35}_{\pm0.42}$ | $\textbf{99.66}_{\pm0.16}$ | 97.66 |
| | Log-Rank | $96.55_{\pm0.31}$ | $98.95_{\pm0.13}$ | $90.08_{\pm1.28}$ | $\textbf{99.36}_{\pm0.13}$ | $92.01_{\pm0.20}$ | $99.24_{\pm0.15}$ | 96.03 |
| | **Log-Rank-M** | $\textbf{98.47}_{\pm0.29}$ | $\textbf{99.28}_{\pm0.22}$ | $\textbf{93.15}_{\pm1.37}$ | $99.25_{\pm0.32}$ | $\textbf{92.93}_{\pm0.43}$ | $\textbf{99.66}_{\pm0.10}$ | 97.13 |
| | Entropy | $74.19_{\pm1.62}$ | $89.49_{\pm0.33}$ | $73.26_{\pm1.48}$ | $84.11_{\pm0.77}$ | $86.58_{\pm0.66}$ | $95.94_{\pm0.35}$ | 83.93 |
| | **Entropy-M** | $\textbf{82.36}_{\pm1.15}$ | $\textbf{92.90}_{\pm0.26}$ | $\textbf{80.64}_{\pm1.12}$ | $\textbf{90.88}_{\pm0.52}$ | $\textbf{88.62}_{\pm0.48}$ | $\textbf{96.91}_{\pm0.23}$ | 88.72 |
| | DetectGPT | $50.15_{\pm0.99}$ | $50.53_{\pm3.65}$ | $48.04_{\pm7.27}$ | $49.25_{\pm1.58}$ | $51.21_{\pm8.39}$ | $55.34_{\pm31.67}$ | 50.75 |
| | **DetectGPT-M** | $\textbf{86.96}_{\pm7.59}$ | $\textbf{87.54}_{\pm7.67}$ | $\textbf{85.22}_{\pm5.50}$ | $\textbf{96.46}_{\pm2.24}$ | $\textbf{71.40}_{\pm9.73}$ | $\textbf{80.33}_{\pm14.18}$ | 84.65 |
| | FastGPT | $64.63_{\pm1.53}$ | $67.68_{\pm1.70}$ | $47.17_{\pm1.53}$ | $71.08_{\pm1.51}$ | $75.31_{\pm0.90}$ | $88.62_{\pm0.67}$ | 69.08 |
| | **FastGPT-M** | $\textbf{88.45}_{\pm0.98}$ | $\textbf{89.73}_{\pm1.25}$ | $\textbf{91.61}_{\pm0.36}$ | $\textbf{95.35}_{\pm0.47}$ | $59.77_{\pm1.62}$ | $54.76_{\pm2.39}$ | 79.95 |
| | DNA-GPT | $45.85_{\pm2.15}$ | $39.61_{\pm1.52}$ | $53.93_{\pm1.47}$ | $36.19_{\pm1.75}$ | $53.01_{\pm1.67}$ | $40.65_{\pm1.60}$ | 44.87 |
| | **DNA-GPT-M** | $\textbf{76.26}_{\pm1.36}$ | $\textbf{76.25}_{\pm1.29}$ | $\textbf{76.38}_{\pm1.33}$ | $\textbf{75.98}_{\pm1.39}$ | $\textbf{76.74}_{\pm1.26}$ | $\textbf{76.30}_{\pm1.29}$ | 76.32 |

Table 9: Performance on DetectRL. The detection models are trained on text generated by ChatGPT and Google-PaLM.

| Metric | Method | DetectRL (Training Text: ChatGPT) | | | | DetectRL (Training Text: Google-PaLM) | | | |
|---|---|---|---|---|---|---|---|---|---|
| | | Llama-2-70b | ChatGPT | Google-PaLM | Avg. | Llama-2-70b | ChatGPT | Google-PaLM | Avg. |
| TPR@FPR-1% | Likelihood | $38.37_{\pm0.92}$ | $10.21_{\pm1.40}$ | $25.66_{\pm2.41}$ | 24.75 | $38.32_{\pm0.88}$ | $10.19_{\pm1.38}$ | $25.66_{\pm2.41}$ | 24.72 |
| | **Likelihood-M** | $\mathbf{45.61}_{\pm3.57}$ | $\mathbf{12.14}_{\pm2.66}$ | $\mathbf{33.18}_{\pm2.46}$ | **30.31** | $\mathbf{42.94}_{\pm4.14}$ | $9.91_{\pm3.87}$ | $\mathbf{35.48}_{\pm2.70}$ | **29.44** |
| | Log-Rank | $42.05_{\pm0.84}$ | $12.44_{\pm1.71}$ | $22.84_{\pm1.28}$ | 25.78 | $42.05_{\pm0.84}$ | $12.41_{\pm1.72}$ | $22.84_{\pm1.28}$ | 25.77 |
| | **Log-Rank-M** | $\mathbf{50.66}_{\pm1.43}$ | $\mathbf{12.73}_{\pm1.65}$ | $\mathbf{41.41}_{\pm2.56}$ | **34.93** | $\mathbf{50.58}_{\pm1.35}$ | $\mathbf{13.00}_{\pm2.43}$ | $\mathbf{41.09}_{\pm4.07}$ | **34.89** |
| | Entropy | $2.03_{\pm0.73}$ | $0.25_{\pm0.16}$ | $6.95_{\pm0.78}$ | 3.07 | $2.03_{\pm0.73}$ | $0.25_{\pm0.16}$ | $6.95_{\pm0.78}$ | 3.07 |
| | **Entropy-M** | $\mathbf{2.13}_{\pm0.30}$ | $\mathbf{0.27}_{\pm0.20}$ | $\mathbf{9.27}_{\pm0.87}$ | **3.89** | $\mathbf{2.22}_{\pm0.37}$ | $\mathbf{0.27}_{\pm0.20}$ | $\mathbf{9.37}_{\pm0.93}$ | **3.96** |
| | DetectGPT | $3.29_{\pm0.97}$ | $5.36_{\pm2.20}$ | $2.79_{\pm1.32}$ | 3.82 | $2.87_{\pm0.56}$ | $6.77_{\pm1.50}$ | $3.86_{\pm1.44}$ | 4.50 |
| | **DetectGPT-M** | $\mathbf{25.29}_{\pm10.05}$ | $\mathbf{7.66}_{\pm4.17}$ | $\mathbf{14.17}_{\pm11.10}$ | **15.71** | $\mathbf{22.00}_{\pm4.30}$ | $4.57_{\pm0.37}$ | $\mathbf{24.94}_{\pm2.68}$ | **17.17** |
| | FastGPT | $11.35_{\pm2.01}$ | $3.78_{\pm1.20}$ | $5.02_{\pm0.85}$ | 6.72 | $11.35_{\pm2.01}$ | $3.78_{\pm1.20}$ | $5.02_{\pm0.85}$ | 6.72 |
| | **FastGPT-M** | $\mathbf{27.71}_{\pm15.63}$ | $\mathbf{13.42}_{\pm8.91}$ | $\mathbf{17.68}_{\pm11.92}$ | **19.60** | $12.41_{\pm0.89}$ | $6.18_{\pm0.83}$ | $20.10_{\pm1.78}$ | 12.90 |
| | DNA-GPT | $31.03_{\pm1.59}$ | $11.05_{\pm1.69}$ | $26.40_{\pm2.08}$ | 22.83 | $41.71_{\pm1.32}$ | $29.49_{\pm0.84}$ | $38.84_{\pm1.58}$ | 36.68 |
| | **DNA-GPT-M** | $\mathbf{41.16}_{\pm1.63}$ | $\mathbf{12.53}_{\pm2.06}$ | $\mathbf{33.57}_{\pm2.15}$ | **29.09** | $\mathbf{47.10}_{\pm0.93}$ | $22.72_{\pm1.50}$ | $\mathbf{40.54}_{\pm1.98}$ | 36.79 |
| AUROC | Likelihood | $78.58_{\pm0.41}$ | $66.61_{\pm0.99}$ | $71.42_{\pm0.49}$ | 72.2 | $78.57_{\pm0.41}$ | $66.61_{\pm0.99}$ | $71.42_{\pm0.49}$ | 72.20 |
| | **Likelihood-M** | $78.15_{\pm4.58}$ | $\mathbf{67.06}_{\pm4.19}$ | $\mathbf{73.40}_{\pm4.38}$ | **72.87** | $\mathbf{87.44}_{\pm0.70}$ | $\mathbf{73.70}_{\pm0.57}$ | $\mathbf{81.58}_{\pm0.93}$ | **80.91** |
| | Log-Rank | $79.67_{\pm0.46}$ | $65.85_{\pm0.94}$ | $70.66_{\pm0.40}$ | 72.06 | $79.67_{\pm0.46}$ | $65.85_{\pm0.94}$ | $70.66_{\pm0.40}$ | 72.06 |
| | **Log-Rank-M** | $\mathbf{90.23}_{\pm0.45}$ | $\mathbf{75.40}_{\pm0.97}$ | $\mathbf{84.44}_{\pm0.52}$ | **83.35** | $\mathbf{90.14}_{\pm0.53}$ | $\mathbf{75.61}_{\pm0.83}$ | $\mathbf{84.63}_{\pm0.59}$ | **83.46** |
| | Entropy | $66.21_{\pm0.88}$ | $63.09_{\pm1.05}$ | $60.73_{\pm0.91}$ | 63.34 | $66.21_{\pm0.88}$ | $63.09_{\pm1.05}$ | $60.73_{\pm0.91}$ | 63.34 |
| | **Entropy-M** | $\mathbf{68.61}_{\pm0.94}$ | $\mathbf{66.83}_{\pm0.95}$ | $\mathbf{67.93}_{\pm1.08}$ | **67.79** | $\mathbf{68.91}_{\pm0.83}$ | $\mathbf{66.86}_{\pm0.95}$ | $\mathbf{67.94}_{\pm1.04}$ | **67.90** |
| | DetectGPT | $49.21_{\pm2.31}$ | $49.64_{\pm0.53}$ | $50.72_{\pm6.91}$ | 49.85 | $47.63_{\pm0.59}$ | $49.78_{\pm0.60}$ | $56.81_{\pm1.41}$ | 51.40 |
| | **DetectGPT-M** | $\mathbf{73.50}_{\pm9.13}$ | $\mathbf{62.93}_{\pm4.59}$ | $\mathbf{61.30}_{\pm15.58}$ | **65.91** | $\mathbf{77.77}_{\pm4.05}$ | $60.34_{\pm3.37}$ | $\mathbf{76.40}_{\pm1.66}$ | **71.50** |
| | FastGPT | $67.72_{\pm1.02}$ | $58.50_{\pm1.22}$ | $56.68_{\pm1.21}$ | 60.97 | $67.72_{\pm1.02}$ | $58.50_{\pm1.22}$ | $56.68_{\pm1.21}$ | 60.97 |
| | **FastGPT-M** | $\mathbf{71.60}_{\pm6.86}$ | $\mathbf{61.26}_{\pm5.62}$ | $\mathbf{66.05}_{\pm9.70}$ | **66.30** | $62.35_{\pm0.49}$ | $51.87_{\pm1.38}$ | $\mathbf{68.88}_{\pm1.00}$ | 61.03 |
| | DNA-GPT | $72.53_{\pm1.19}$ | $66.07_{\pm1.38}$ | $68.35_{\pm0.98}$ | 68.98 | $79.00_{\pm0.86}$ | $72.27_{\pm0.81}$ | $77.96_{\pm0.94}$ | 76.41 |
| | **DNA-GPT-M** | $\mathbf{75.37}_{\pm1.13}$ | $\mathbf{66.19}_{\pm1.40}$ | $\mathbf{71.77}_{\pm0.79}$ | **71.11** | $\mathbf{80.09}_{\pm1.26}$ | $71.09_{\pm1.12}$ | $77.15_{\pm1.02}$ | 76.11 |

Table 10: Performance on Reuters dataset. The detection models are trained on text generated by GPT4All.

| Metric | Method | GPT4All | ChatGPT | Dolly | ChatGLM | Claude | ChatGPT-turbo | Avg. |
|---|---|---|---|---|---|---|---|---|
| TPR@FPR-1% | Likelihood | $19.47_{\pm2.45}$ | $80.89_{\pm2.17}$ | $12.44_{\pm1.38}$ | $94.36_{\pm1.15}$ | $18.89_{\pm3.41}$ | $90.22_{\pm1.23}$ | 52.71 |
| | **Likelihood-M** | $\mathbf{61.42}_{\pm5.74}$ | $\mathbf{88.44}_{\pm2.19}$ | $\mathbf{35.56}_{\pm2.86}$ | $\mathbf{95.24}_{\pm1.16}$ | $\mathbf{32.36}_{\pm5.93}$ | $\mathbf{92.76}_{\pm0.71}$ | **67.63** |
| | Log-Rank | $29.91_{\pm3.27}$ | $86.22_{\pm1.87}$ | $15.20_{\pm1.70}$ | $97.33_{\pm0.64}$ | $24.58_{\pm3.35}$ | $93.16_{\pm1.07}$ | 57.73 |
| | **Log-Rank-M** | $\mathbf{73.82}_{\pm3.32}$ | $\mathbf{89.56}_{\pm1.79}$ | $\mathbf{38.31}_{\pm2.26}$ | $96.71_{\pm1.62}$ | $\mathbf{35.47}_{\pm4.63}$ | $\mathbf{94.49}_{\pm0.62}$ | **71.39** |
| | Entropy | $6.18_{\pm1.40}$ | $0.04_{\pm0.09}$ | $13.07_{\pm1.76}$ | $0.22_{\pm0.20}$ | $0.00_{\pm0.00}$ | $0.22_{\pm0.00}$ | 3.29 |
| | **Entropy-M** | $\mathbf{17.42}_{\pm2.84}$ | $\mathbf{0.36}_{\pm0.30}$ | $12.36_{\pm2.86}$ | $\mathbf{0.98}_{\pm0.87}$ | $0.00_{\pm0.00}$ | $\mathbf{0.40}_{\pm0.38}$ | **5.25** |
| | DetectGPT | $0.00_{\pm0.00}$ | $0.00_{\pm0.00}$ | $0.00_{\pm0.00}$ | $0.00_{\pm0.00}$ | $0.44_{\pm0.40}$ | $0.04_{\pm0.09}$ | 0.08 |
| | **DetectGPT-M** | $\mathbf{40.89}_{\pm6.07}$ | $\mathbf{15.33}_{\pm8.81}$ | $\mathbf{46.40}_{\pm2.46}$ | $\mathbf{40.00}_{\pm8.10}$ | $\mathbf{2.40}_{\pm0.38}$ | $\mathbf{6.40}_{\pm2.50}$ | **25.24** |
| | FastGPT | $3.56_{\pm3.08}$ | $0.04_{\pm0.09}$ | $12.18_{\pm6.70}$ | $0.09_{\pm0.11}$ | $0.09_{\pm0.11}$ | $0.04_{\pm0.09}$ | 2.67 |
| | **FastGPT-M** | $\mathbf{61.69}_{\pm7.18}$ | $\mathbf{21.78}_{\pm6.77}$ | $\mathbf{65.24}_{\pm5.12}$ | $\mathbf{31.78}_{\pm9.97}$ | $\mathbf{1.38}_{\pm0.84}$ | $\mathbf{1.51}_{\pm0.55}$ | **30.56** |
| | DNA-GPT | $85.20_{\pm6.79}$ | $92.58_{\pm3.94}$ | $68.49_{\pm4.37}$ | $96.93_{\pm3.14}$ | $79.24_{\pm4.93}$ | $95.29_{\pm3.53}$ | 86.29 |
| | **DNA-GPT-M** | $\mathbf{97.51}_{\pm1.31}$ | $\mathbf{98.40}_{\pm0.88}$ | $\mathbf{81.29}_{\pm3.75}$ | $\mathbf{99.87}_{\pm0.11}$ | $\mathbf{89.07}_{\pm3.57}$ | $\mathbf{99.24}_{\pm0.30}$ | **94.23** |
| AUROC | Likelihood | $75.19_{\pm1.39}$ | $97.55_{\pm0.33}$ | $58.77_{\pm1.76}$ | $99.62_{\pm0.11}$ | $86.45_{\pm0.70}$ | $98.42_{\pm0.25}$ | 86.00 |
| | **Likelihood-M** | $\mathbf{93.74}_{\pm1.23}$ | $\mathbf{98.42}_{\pm0.26}$ | $\mathbf{77.05}_{\pm0.53}$ | $99.61_{\pm0.11}$ | $\mathbf{90.68}_{\pm0.56}$ | $\mathbf{98.78}_{\pm0.33}$ | **93.05** |
| | Log-Rank | $78.39_{\pm1.23}$ | $97.77_{\pm0.32}$ | $58.22_{\pm2.03}$ | $99.61_{\pm0.19}$ | $85.68_{\pm0.67}$ | $98.67_{\pm0.24}$ | 86.39 |
| | **Log-Rank-M** | $\mathbf{95.50}_{\pm0.81}$ | $\mathbf{98.60}_{\pm0.23}$ | $\mathbf{76.60}_{\pm1.33}$ | $\mathbf{99.64}_{\pm0.29}$ | $\mathbf{90.45}_{\pm0.69}$ | $\mathbf{98.99}_{\pm0.31}$ | **93.30** |
| | Entropy | $72.48_{\pm1.24}$ | $\mathbf{35.35}_{\pm1.65}$ | $\mathbf{61.92}_{\pm1.32}$ | $\mathbf{50.96}_{\pm1.57}$ | $23.76_{\pm1.05}$ | $17.55_{\pm0.92}$ | 43.67 |
| | **Entropy-M** | $70.31_{\pm1.98}$ | $34.84_{\pm2.95}$ | $50.09_{\pm1.57}$ | $46.62_{\pm1.12}$ | $\mathbf{27.55}_{\pm2.60}$ | $\mathbf{29.08}_{\pm4.18}$ | 43.08 |
| | DetectGPT | $75.62_{\pm1.20}$ | $57.18_{\pm1.91}$ | $77.25_{\pm0.66}$ | $61.97_{\pm0.90}$ | $58.01_{\pm1.75}$ | $20.42_{\pm1.07}$ | 58.41 |
| | **DetectGPT-M** | $\mathbf{87.58}_{\pm0.93}$ | $\mathbf{78.19}_{\pm2.33}$ | $\mathbf{86.29}_{\pm1.18}$ | $\mathbf{89.73}_{\pm0.97}$ | $\mathbf{58.40}_{\pm1.54}$ | $\mathbf{65.72}_{\pm1.36}$ | **77.65** |
| | FastGPT | $76.50_{\pm1.39}$ | $39.56_{\pm0.54}$ | $82.45_{\pm0.47}$ | $33.21_{\pm1.82}$ | $41.31_{\pm0.89}$ | $11.29_{\pm0.49}$ | 47.39 |
| | **FastGPT-M** | $\mathbf{96.46}_{\pm0.31}$ | $\mathbf{90.21}_{\pm0.75}$ | $\mathbf{96.23}_{\pm0.36}$ | $\mathbf{92.36}_{\pm0.74}$ | $\mathbf{66.98}_{\pm1.06}$ | $\mathbf{56.28}_{\pm1.80}$ | **83.09** |
| | DNA-GPT | $99.51_{\pm0.14}$ | $99.72_{\pm0.07}$ | $\mathbf{98.41}_{\pm0.21}$ | $99.85_{\pm0.05}$ | $99.21_{\pm0.14}$ | $99.81_{\pm0.06}$ | 99.42 |
| | **DNA-GPT-M** | $\mathbf{99.80}_{\pm0.09}$ | $\mathbf{99.88}_{\pm0.07}$ | $98.39_{\pm0.17}$ | $\mathbf{99.94}_{\pm0.06}$ | $\mathbf{99.53}_{\pm0.09}$ | $\mathbf{99.89}_{\pm0.09}$ | **99.57** |

Table 11: Performance on Reuters dataset. The detection models are trained on text generated by ChatGPT.

| Metric | Method | GPT4All | ChatGPT | Dolly | ChatGLM | Claude | ChatGPT-turbo | Avg. |
|---|---|---|---|---|---|---|---|---|
| TPR@FPR-1% | Likelihood | $19.78_{\pm2.21}$ | $81.07_{\pm2.05}$ | $12.53_{\pm1.35}$ | $94.36_{\pm1.15}$ | $19.42_{\pm3.17}$ | $90.31_{\pm1.11}$ | 52.91 |
| | **Likelihood-M** | $\mathbf{59.73}_{\pm8.56}$ | $\mathbf{88.98}_{\pm1.58}$ | $\mathbf{35.91}_{\pm3.82}$ | $\mathbf{96.00}_{\pm2.14}$ | $\mathbf{26.71}_{\pm5.93}$ | $\mathbf{93.24}_{\pm0.94}$ | **66.76** |
| | Log-Rank | $29.96_{\pm3.29}$ | $86.22_{\pm1.87}$ | $15.20_{\pm1.70}$ | $97.33_{\pm0.64}$ | $24.58_{\pm3.35}$ | $93.20_{\pm1.13}$ | 57.75 |
| | **Log-Rank-M** | $\mathbf{63.91}_{\pm8.04}$ | $\mathbf{90.62}_{\pm2.38}$ | $\mathbf{34.58}_{\pm4.71}$ | $\mathbf{98.13}_{\pm0.46}$ | $\mathbf{29.73}_{\pm8.71}$ | $\mathbf{94.53}_{\pm0.67}$ | **68.59** |
| | Entropy | $0.09_{\pm0.11}$ | $2.49_{\pm1.25}$ | $2.04_{\pm0.57}$ | $1.69_{\pm0.54}$ | $4.36_{\pm1.81}$ | $9.56_{\pm2.90}$ | 3.37 |
| | **Entropy-M** | $\mathbf{7.07}_{\pm3.05}$ | $\mathbf{17.47}_{\pm5.59}$ | $\mathbf{13.60}_{\pm2.23}$ | $\mathbf{9.02}_{\pm3.21}$ | $\mathbf{18.09}_{\pm4.26}$ | $\mathbf{33.29}_{\pm7.84}$ | **16.42** |
| | DetectGPT | $0.00_{\pm0.00}$ | $0.00_{\pm0.00}$ | $0.00_{\pm0.00}$ | $0.00_{\pm0.00}$ | $0.44_{\pm0.40}$ | $0.04_{\pm0.09}$ | 0.08 |
| | **DetectGPT-M** | $\mathbf{22.76}_{\pm25.66}$ | $\mathbf{47.91}_{\pm28.49}$ | $\mathbf{11.69}_{\pm12.56}$ | $\mathbf{23.64}_{\pm28.60}$ | $\mathbf{8.27}_{\pm5.92}$ | $\mathbf{71.24}_{\pm17.28}$ | **30.92** |
| | FastGPT | $0.31_{\pm0.23}$ | $2.04_{\pm0.86}$ | $0.04_{\pm0.09}$ | $3.16_{\pm1.03}$ | $1.42_{\pm0.57}$ | $23.51_{\pm4.15}$ | 5.08 |
| | **FastGPT-M** | $\mathbf{56.04}_{\pm11.84}$ | $\mathbf{17.07}_{\pm10.24}$ | $\mathbf{61.51}_{\pm9.34}$ | $\mathbf{26.67}_{\pm13.24}$ | $\mathbf{0.89}_{\pm0.92}$ | $\mathbf{1.29}_{\pm0.68}$ | **27.24** |
| | DNA-GPT | $29.82_{\pm18.89}$ | $69.78_{\pm22.51}$ | $15.64_{\pm12.02}$ | $83.47_{\pm14.87}$ | $21.96_{\pm19.50}$ | $82.22_{\pm15.63}$ | 50.48 |
| | **DNA-GPT-M** | $\mathbf{86.80}_{\pm5.37}$ | $\mathbf{96.76}_{\pm1.16}$ | $\mathbf{59.33}_{\pm8.10}$ | $\mathbf{99.29}_{\pm0.38}$ | $\mathbf{61.91}_{\pm13.48}$ | $\mathbf{97.51}_{\pm1.24}$ | **83.60** |
| AUROC | Likelihood | $75.19_{\pm1.39}$ | $97.55_{\pm0.33}$ | $58.77_{\pm1.76}$ | $99.62_{\pm0.11}$ | $86.46_{\pm0.70}$ | $98.42_{\pm0.25}$ | 86.00 |
| | **Likelihood-M** | $\mathbf{93.97}_{\pm1.27}$ | $\mathbf{98.68}_{\pm0.41}$ | $\mathbf{79.54}_{\pm1.12}$ | $\mathbf{99.70}_{\pm0.24}$ | $\mathbf{90.39}_{\pm1.08}$ | $\mathbf{98.94}_{\pm0.23}$ | **93.54** |
| | Log-Rank | $78.39_{\pm1.23}$ | $97.77_{\pm0.32}$ | $58.22_{\pm2.03}$ | $99.61_{\pm0.19}$ | $85.68_{\pm0.67}$ | $98.67_{\pm0.24}$ | 86.39 |
| | **Log-Rank-M** | $\mathbf{93.41}_{\pm1.37}$ | $\mathbf{98.69}_{\pm0.25}$ | $\mathbf{76.76}_{\pm1.45}$ | $\mathbf{99.75}_{\pm0.20}$ | $\mathbf{89.55}_{\pm1.04}$ | $\mathbf{99.03}_{\pm0.27}$ | **92.86** |
| | Entropy | $27.52_{\pm1.24}$ | $64.65_{\pm1.65}$ | $38.08_{\pm1.32}$ | $49.04_{\pm1.57}$ | $76.24_{\pm1.05}$ | $82.45_{\pm0.92}$ | 56.33 |
| | **Entropy-M** | $\mathbf{55.30}_{\pm2.65}$ | $\mathbf{80.07}_{\pm1.89}$ | $\mathbf{56.54}_{\pm2.20}$ | $\mathbf{69.75}_{\pm2.62}$ | $\mathbf{83.33}_{\pm1.09}$ | $\mathbf{90.46}_{\pm1.17}$ | **72.58** |
| | DetectGPT | $75.62_{\pm1.20}$ | $57.18_{\pm1.91}$ | $\mathbf{77.25}_{\pm0.66}$ | $61.97_{\pm0.90}$ | $58.01_{\pm1.75}$ | $20.42_{\pm1.07}$ | 58.41 |
| | **DetectGPT-M** | $\mathbf{84.60}_{\pm11.69}$ | $\mathbf{92.37}_{\pm5.63}$ | $69.47_{\pm12.90}$ | $\mathbf{80.47}_{\pm13.77}$ | $\mathbf{75.78}_{\pm9.49}$ | $\mathbf{97.20}_{\pm1.23}$ | **83.32** |
| | FastGPT | $23.50_{\pm1.39}$ | $60.44_{\pm0.54}$ | $17.55_{\pm0.47}$ | $66.79_{\pm1.82}$ | $58.69_{\pm0.89}$ | $\mathbf{88.71}_{\pm0.49}$ | 52.61 |
| | **FastGPT-M** | $\mathbf{96.25}_{\pm0.33}$ | $\mathbf{89.97}_{\pm0.74}$ | $\mathbf{96.04}_{\pm0.33}$ | $\mathbf{92.12}_{\pm0.76}$ | $\mathbf{66.72}_{\pm1.11}$ | $56.04_{\pm1.73}$ | **82.86** |
| | DNA-GPT | $97.34_{\pm0.54}$ | $99.17_{\pm0.30}$ | $95.82_{\pm0.62}$ | $99.45_{\pm0.23}$ | $97.21_{\pm0.52}$ | $99.38_{\pm0.26}$ | 98.06 |
| | **DNA-GPT-M** | $\mathbf{99.42}_{\pm0.15}$ | $\mathbf{99.78}_{\pm0.09}$ | $\mathbf{97.31}_{\pm0.24}$ | $\mathbf{99.89}_{\pm0.10}$ | $\mathbf{98.78}_{\pm0.24}$ | $\mathbf{99.82}_{\pm0.12}$ | **99.16** |

Table 12: Performance on Reuters dataset. The detection models are trained on text generated by ChatGPT-turbo.

| Metric | Method | GPT4All | ChatGPT | Dolly | ChatGLM | Claude | ChatGPT-turbo | Avg. |
|---|---|---|---|---|---|---|---|---|
| TPR@FPR-1% | Likelihood | $19.82_{\pm2.17}$ | $81.07_{\pm2.05}$ | $12.53_{\pm1.35}$ | $94.44_{\pm1.10}$ | $19.42_{\pm3.17}$ | $90.27_{\pm1.07}$ | 52.93 |
| | **Likelihood-M** | $\mathbf{66.62}_{\pm5.21}$ | $\mathbf{90.13}_{\pm1.15}$ | $\mathbf{38.62}_{\pm2.74}$ | $\mathbf{95.96}_{\pm1.29}$ | $\mathbf{29.60}_{\pm2.32}$ | $\mathbf{93.91}_{\pm1.03}$ | **69.14** |
| | Log-Rank | $29.96_{\pm3.29}$ | $86.22_{\pm1.87}$ | $15.20_{\pm1.70}$ | $97.33_{\pm0.64}$ | $24.58_{\pm3.35}$ | $93.20_{\pm1.13}$ | 57.75 |
| | **Log-Rank-M** | $\mathbf{60.18}_{\pm8.59}$ | $\mathbf{90.27}_{\pm1.98}$ | $\mathbf{33.07}_{\pm3.84}$ | $\mathbf{98.22}_{\pm0.40}$ | $\mathbf{27.29}_{\pm6.31}$ | $\mathbf{94.00}_{\pm0.40}$ | **67.17** |
| | Entropy | $0.09_{\pm0.11}$ | $2.49_{\pm1.25}$ | $2.04_{\pm0.57}$ | $1.69_{\pm0.54}$ | $4.36_{\pm1.81}$ | $9.60_{\pm2.94}$ | 3.38 |
| | **Entropy-M** | $\mathbf{5.11}_{\pm2.70}$ | $\mathbf{16.18}_{\pm4.91}$ | $\mathbf{12.27}_{\pm1.51}$ | $\mathbf{7.69}_{\pm2.70}$ | $\mathbf{17.42}_{\pm3.93}$ | $\mathbf{31.69}_{\pm6.61}$ | **15.06** |
| | DetectGPT | $0.00_{\pm0.00}$ | $0.00_{\pm0.00}$ | $0.00_{\pm0.00}$ | $0.00_{\pm0.00}$ | $0.62_{\pm0.53}$ | $9.24_{\pm2.65}$ | 1.64 |
| | **DetectGPT-M** | $\mathbf{6.22}_{\pm3.99}$ | $\mathbf{26.71}_{\pm14.32}$ | $\mathbf{2.49}_{\pm0.97}$ | $\mathbf{4.04}_{\pm3.33}$ | $\mathbf{4.18}_{\pm2.22}$ | $\mathbf{60.67}_{\pm14.38}$ | **17.39** |
| | FastGPT | $0.31_{\pm0.23}$ | $2.04_{\pm0.86}$ | $0.04_{\pm0.09}$ | $3.16_{\pm1.03}$ | $1.42_{\pm0.57}$ | $23.51_{\pm4.15}$ | 5.08 |
| | **FastGPT-M** | $0.00_{\pm0.00}$ | $0.22_{\pm0.28}$ | $0.04_{\pm0.09}$ | $0.53_{\pm0.33}$ | $0.62_{\pm0.29}$ | $7.38_{\pm2.42}$ | 1.47 |
| | DNA-GPT | $2.09_{\pm1.29}$ | $4.31_{\pm1.94}$ | $1.69_{\pm1.21}$ | $5.51_{\pm1.67}$ | $1.96_{\pm1.60}$ | $5.24_{\pm2.29}$ | 3.47 |
| | **DNA-GPT-M** | $\mathbf{5.82}_{\pm1.06}$ | $\mathbf{6.44}_{\pm0.97}$ | $\mathbf{3.56}_{\pm1.27}$ | $\mathbf{6.49}_{\pm0.87}$ | $\mathbf{3.78}_{\pm1.29}$ | $\mathbf{6.67}_{\pm0.84}$ | **5.46** |
| AUROC | Likelihood | $75.20_{\pm1.39}$ | $97.55_{\pm0.33}$ | $58.77_{\pm1.77}$ | $99.62_{\pm0.11}$ | $86.46_{\pm0.70}$ | $98.42_{\pm0.25}$ | 86.00 |
| | **Likelihood-M** | $\mathbf{94.79}_{\pm0.60}$ | $\mathbf{98.72}_{\pm0.40}$ | $\mathbf{79.93}_{\pm1.44}$ | $\mathbf{99.72}_{\pm0.16}$ | $\mathbf{90.83}_{\pm0.53}$ | $\mathbf{98.96}_{\pm0.26}$ | **93.82** |
| | Log-Rank | $78.39_{\pm1.23}$ | $97.77_{\pm0.32}$ | $58.22_{\pm2.03}$ | $99.61_{\pm0.19}$ | $85.68_{\pm0.67}$ | $98.67_{\pm0.24}$ | 86.39 |
| | **Log-Rank-M** | $\mathbf{92.21}_{\pm2.45}$ | $\mathbf{98.63}_{\pm0.23}$ | $\mathbf{76.27}_{\pm1.24}$ | $\mathbf{99.73}_{\pm0.21}$ | $\mathbf{89.02}_{\pm0.89}$ | $\mathbf{98.99}_{\pm0.27}$ | **92.48** |
| | Entropy | $27.52_{\pm1.24}$ | $64.65_{\pm1.65}$ | $38.08_{\pm1.32}$ | $49.04_{\pm1.57}$ | $76.24_{\pm1.05}$ | $82.45_{\pm0.92}$ | 56.33 |
| | **Entropy-M** | $\mathbf{54.11}_{\pm3.57}$ | $\mathbf{79.38}_{\pm1.96}$ | $\mathbf{55.94}_{\pm1.82}$ | $\mathbf{68.88}_{\pm2.66}$ | $\mathbf{83.14}_{\pm1.07}$ | $\mathbf{89.91}_{\pm0.91}$ | **71.89** |
| | DetectGPT | $24.38_{\pm1.20}$ | $42.82_{\pm1.91}$ | $22.75_{\pm0.66}$ | $38.03_{\pm0.90}$ | $41.99_{\pm1.75}$ | $79.58_{\pm1.07}$ | 41.59 |
| | **DetectGPT-M** | $\mathbf{79.17}_{\pm7.47}$ | $\mathbf{89.36}_{\pm4.32}$ | $\mathbf{59.50}_{\pm5.37}$ | $\mathbf{71.41}_{\pm7.62}$ | $\mathbf{73.92}_{\pm8.27}$ | $\mathbf{96.53}_{\pm1.14}$ | **78.32** |
| | FastGPT | $23.50_{\pm1.39}$ | $60.44_{\pm0.54}$ | $17.55_{\pm0.47}$ | $66.79_{\pm1.82}$ | $58.69_{\pm0.89}$ | $88.71_{\pm0.49}$ | 52.61 |
| | **FastGPT-M** | $12.73_{\pm1.55}$ | $35.93_{\pm2.31}$ | $9.43_{\pm0.56}$ | $36.46_{\pm3.11}$ | $50.18_{\pm2.00}$ | $76.28_{\pm1.01}$ | 36.83 |
| | DNA-GPT | $45.62_{\pm1.73}$ | $63.85_{\pm1.00}$ | $35.96_{\pm1.28}$ | $71.24_{\pm1.14}$ | $43.56_{\pm1.49}$ | $70.63_{\pm1.33}$ | 55.14 |
| | **DNA-GPT-M** | $\mathbf{79.05}_{\pm1.13}$ | $\mathbf{87.32}_{\pm0.51}$ | $\mathbf{64.13}_{\pm1.27}$ | $\mathbf{91.42}_{\pm0.64}$ | $\mathbf{67.14}_{\pm1.48}$ | $\mathbf{89.37}_{\pm0.79}$ | **79.74** |

Table 13: Performance on Reuters dataset. The detection models are trained on text generated by ChatGLM.

| Metric | Method | GPT4All | ChatGPT | Dolly | ChatGLM | Claude | ChatGPT-turbo | Avg. |
|---|---|---|---|---|---|---|---|---|
| TPR@FPR-1% | Likelihood | $19.78_{\pm2.13}$ | $81.07_{\pm2.05}$ | $12.49_{\pm1.33}$ | $94.44_{\pm1.10}$ | $19.42_{\pm3.17}$ | $90.36_{\pm1.06}$ | 52.93 |
| | **Likelihood-M** | $\mathbf{41.42}_{\pm8.12}$ | $\mathbf{86.84}_{\pm2.44}$ | $\mathbf{31.47}_{\pm2.22}$ | $\mathbf{98.58}_{\pm0.23}$ | $\mathbf{20.84}_{\pm2.97}$ | $\mathbf{91.56}_{\pm1.28}$ | **61.79** |
| | Log-Rank | $29.96_{\pm3.29}$ | $86.22_{\pm1.87}$ | $15.20_{\pm1.70}$ | $97.33_{\pm0.64}$ | $24.58_{\pm3.35}$ | $93.20_{\pm1.13}$ | 57.75 |
| | **Log-Rank-M** | $\mathbf{62.40}_{\pm4.54}$ | $\mathbf{90.49}_{\pm2.34}$ | $\mathbf{33.82}_{\pm3.87}$ | $\mathbf{98.22}_{\pm0.47}$ | $\mathbf{30.44}_{\pm6.61}$ | $\mathbf{94.13}_{\pm0.75}$ | **68.25** |
| | Entropy | $4.89_{\pm2.71}$ | $0.36_{\pm0.71}$ | $10.49_{\pm4.75}$ | $0.44_{\pm0.37}$ | $0.58_{\pm1.16}$ | $1.38_{\pm2.31}$ | 3.02 |
| | **Entropy-M** | $\mathbf{6.13}_{\pm4.80}$ | $\mathbf{17.20}_{\pm4.05}$ | $\mathbf{13.11}_{\pm3.21}$ | $\mathbf{9.73}_{\pm4.98}$ | $\mathbf{18.76}_{\pm2.64}$ | $\mathbf{33.87}_{\pm5.03}$ | **16.47** |
| | DetectGPT | $0.00_{\pm0.00}$ | $0.00_{\pm0.00}$ | $0.00_{\pm0.00}$ | $0.00_{\pm0.00}$ | $0.44_{\pm0.40}$ | $0.04_{\pm0.09}$ | 0.08 |
| | **DetectGPT-M** | $\mathbf{40.53}_{\pm6.43}$ | $\mathbf{14.18}_{\pm7.67}$ | $\mathbf{45.73}_{\pm2.50}$ | $\mathbf{38.98}_{\pm8.03}$ | $\mathbf{2.18}_{\pm0.53}$ | $\mathbf{5.47}_{\pm2.15}$ | **24.51** |
| | FastGPT | $0.31_{\pm0.23}$ | $2.04_{\pm0.86}$ | $0.04_{\pm0.09}$ | $3.16_{\pm1.03}$ | $1.42_{\pm0.57}$ | $23.51_{\pm4.15}$ | 5.08 |
| | **FastGPT-M** | $\mathbf{56.04}_{\pm11.84}$ | $\mathbf{17.07}_{\pm10.24}$ | $\mathbf{61.51}_{\pm9.34}$ | $\mathbf{26.67}_{\pm13.24}$ | $0.89_{\pm0.92}$ | $1.29_{\pm0.68}$ | **27.24** |
| | DNA-GPT | $27.96_{\pm18.31}$ | $66.84_{\pm18.61}$ | $16.00_{\pm12.45}$ | $85.51_{\pm12.79}$ | $20.49_{\pm17.96}$ | $80.40_{\pm13.38}$ | 49.53 |
| | **DNA-GPT-M** | $\mathbf{82.89}_{\pm7.12}$ | $\mathbf{94.93}_{\pm1.99}$ | $\mathbf{56.31}_{\pm9.16}$ | $\mathbf{98.98}_{\pm0.18}$ | $\mathbf{57.78}_{\pm15.40}$ | $\mathbf{96.53}_{\pm1.23}$ | **81.24** |
| AUROC | Likelihood | $75.19_{\pm1.39}$ | $97.55_{\pm0.33}$ | $58.77_{\pm1.77}$ | $99.62_{\pm0.11}$ | $86.46_{\pm0.70}$ | $98.42_{\pm0.25}$ | 86.00 |
| | **Likelihood-M** | $\mathbf{87.15}_{\pm3.71}$ | $\mathbf{98.44}_{\pm0.43}$ | $\mathbf{77.92}_{\pm1.74}$ | $\mathbf{99.95}_{\pm0.00}$ | $\mathbf{87.99}_{\pm1.16}$ | $\mathbf{98.69}_{\pm0.26}$ | **91.69** |
| | Log-Rank | $78.39_{\pm1.23}$ | $97.77_{\pm0.32}$ | $58.22_{\pm2.03}$ | $99.61_{\pm0.19}$ | $85.68_{\pm0.67}$ | $98.67_{\pm0.24}$ | 86.39 |
| | **Log-Rank-M** | $\mathbf{93.39}_{\pm0.75}$ | $\mathbf{98.68}_{\pm0.24}$ | $\mathbf{76.11}_{\pm1.61}$ | $\mathbf{99.75}_{\pm0.19}$ | $\mathbf{89.70}_{\pm0.41}$ | $\mathbf{98.99}_{\pm0.25}$ | **92.77** |
| | Entropy | $62.76_{\pm18.55}$ | $40.53_{\pm11.30}$ | | $49.39_{\pm1.74}$ | $33.61_{\pm20.52}$ | $30.23_{\pm25.75}$ | 45.45 |
| | **Entropy-M** | $54.30_{\pm3.63}$ | $\mathbf{79.38}_{\pm1.33}$ | $\mathbf{55.89}_{\pm1.46}$ | $\mathbf{69.08}_{\pm2.00}$ | $\mathbf{83.03}_{\pm0.44}$ | $\mathbf{89.77}_{\pm1.24}$ | **71.91** |
| | DetectGPT | $75.62_{\pm1.20}$ | $57.18_{\pm1.91}$ | $77.25_{\pm0.66}$ | $61.97_{\pm0.90}$ | $58.01_{\pm1.75}$ | $20.42_{\pm1.07}$ | 58.41 |
| | **DetectGPT-M** | $\mathbf{87.17}_{\pm0.76}$ | $\mathbf{77.46}_{\pm1.85}$ | $\mathbf{86.14}_{\pm1.14}$ | $\mathbf{89.36}_{\pm1.25}$ | $57.83_{\pm1.55}$ | $\mathbf{64.40}_{\pm1.89}$ | **77.06** |
| | FastGPT | $23.50_{\pm1.39}$ | $60.44_{\pm0.54}$ | $17.55_{\pm0.47}$ | $66.79_{\pm1.82}$ | $58.69_{\pm0.89}$ | $88.71_{\pm0.49}$ | 52.61 |
| | **FastGPT-M** | $\mathbf{96.27}_{\pm0.33}$ | $\mathbf{89.98}_{\pm0.74}$ | $\mathbf{96.07}_{\pm0.33}$ | $\mathbf{92.13}_{\pm0.76}$ | $\mathbf{66.74}_{\pm1.10}$ | $56.05_{\pm1.74}$ | **82.87** |
| | DNA-GPT | $97.10_{\pm0.48}$ | $98.94_{\pm0.26}$ | $95.56_{\pm0.57}$ | $99.29_{\pm0.17}$ | $96.95_{\pm0.46}$ | $99.22_{\pm0.22}$ | 97.85 |
| | **DNA-GPT-M** | $\mathbf{99.33}_{\pm0.15}$ | $\mathbf{99.72}_{\pm0.09}$ | $\mathbf{97.15}_{\pm0.21}$ | $\mathbf{99.82}_{\pm0.10}$ | $\mathbf{98.68}_{\pm0.24}$ | $\mathbf{99.77}_{\pm0.12}$ | **99.08** |

Table 14: Performance on Reuters dataset. The detection models are trained on text generated by Dolly.

| Metric | Method | GPT4All | ChatGPT | Dolly | ChatGLM | Claude | ChatGPT-turbo | Avg. |
|---|---|---|---|---|---|---|---|---|
| TPR@FPR-1% | Likelihood | $19.78_{\pm2.21}$ | $81.07_{\pm2.05}$ | $12.40_{\pm1.37}$ | $94.44_{\pm1.10}$ | $19.38_{\pm3.23}$ | $90.27_{\pm1.07}$ | 52.89 |
| | **Likelihood-M** | $\mathbf{59.96}_{\pm5.45}$ | $\mathbf{88.58}_{\pm2.29}$ | $\mathbf{35.42}_{\pm2.84}$ | $\mathbf{95.91}_{\pm1.21}$ | $\mathbf{31.96}_{\pm6.22}$ | $\mathbf{92.93}_{\pm0.77}$ | **67.46** |
| | Log-Rank | $6.62_{\pm11.36}$ | $17.24_{\pm34.49}$ | $10.49_{\pm2.42}$ | $19.42_{\pm38.62}$ | $4.93_{\pm9.87}$ | $18.58_{\pm37.16}$ | 12.88 |
| | **Log-Rank-M** | $\mathbf{66.76}_{\pm7.01}$ | $\mathbf{90.49}_{\pm1.80}$ | $\mathbf{35.42}_{\pm4.26}$ | $\mathbf{98.09}_{\pm0.72}$ | $\mathbf{31.78}_{\pm6.22}$ | $\mathbf{94.31}_{\pm0.39}$ | **69.47** |
| | Entropy | $\mathbf{6.18}_{\pm1.40}$ | $0.04_{\pm0.09}$ | $\mathbf{13.07}_{\pm1.76}$ | $0.22_{\pm0.20}$ | $0.00_{\pm0.00}$ | $0.22_{\pm0.00}$ | 3.29 |
| | **Entropy-M** | $2.40_{\pm0.74}$ | $\mathbf{7.29}_{\pm5.90}$ | $9.91_{\pm1.09}$ | $\mathbf{3.16}_{\pm2.61}$ | $\mathbf{9.24}_{\pm7.67}$ | $\mathbf{16.13}_{\pm13.02}$ | **8.02** |
| | DetectGPT | $0.00_{\pm0.00}$ | $0.00_{\pm0.00}$ | $0.00_{\pm0.00}$ | $0.00_{\pm0.00}$ | $0.44_{\pm0.40}$ | $0.04_{\pm0.09}$ | 0.08 |
| | **DetectGPT-M** | $\mathbf{42.93}_{\pm25.42}$ | $\mathbf{37.20}_{\pm35.68}$ | $\mathbf{39.69}_{\pm19.14}$ | $\mathbf{51.73}_{\pm37.94}$ | $\mathbf{6.27}_{\pm7.20}$ | $\mathbf{31.56}_{\pm34.65}$ | **34.90** |
| | FastGPT | $3.56_{\pm3.08}$ | $0.04_{\pm0.09}$ | $12.18_{\pm6.70}$ | $0.09_{\pm0.11}$ | $0.09_{\pm0.11}$ | $0.04_{\pm0.09}$ | 2.67 |
| | **FastGPT-M** | $\mathbf{61.69}_{\pm7.18}$ | $\mathbf{21.78}_{\pm6.77}$ | $\mathbf{65.24}_{\pm5.12}$ | $\mathbf{31.78}_{\pm9.97}$ | $\mathbf{1.38}_{\pm0.84}$ | $\mathbf{1.51}_{\pm0.55}$ | **30.56** |
| | DNA-GPT | $78.71_{\pm8.56}$ | $91.47_{\pm7.33}$ | $70.00_{\pm10.40}$ | $95.51_{\pm4.74}$ | $76.76_{\pm8.45}$ | $94.80_{\pm5.80}$ | 84.54 |
| | **DNA-GPT-M** | $\mathbf{93.24}_{\pm5.57}$ | $\mathbf{98.09}_{\pm1.34}$ | $\mathbf{79.07}_{\pm7.29}$ | $\mathbf{99.87}_{\pm0.27}$ | $\mathbf{85.20}_{\pm6.66}$ | $\mathbf{98.31}_{\pm1.35}$ | **92.30** |
| AUROC | Likelihood | $75.19_{\pm1.39}$ | $97.55_{\pm0.33}$ | $58.77_{\pm1.76}$ | $99.62_{\pm0.10}$ | $86.45_{\pm0.70}$ | $98.42_{\pm0.25}$ | 86.00 |
| | **Likelihood-M** | $\mathbf{93.42}_{\pm0.91}$ | $\mathbf{98.45}_{\pm0.29}$ | $\mathbf{77.41}_{\pm1.04}$ | $\mathbf{99.67}_{\pm0.16}$ | $\mathbf{90.57}_{\pm0.61}$ | $\mathbf{98.78}_{\pm0.33}$ | **93.05** |
| | Log-Rank | $32.84_{\pm22.66}$ | $21.28_{\pm38.17}$ | $43.51_{\pm5.44}$ | $20.22_{\pm39.67}$ | $28.30_{\pm28.33}$ | $20.77_{\pm38.91}$ | 27.82 |
| | **Log-Rank-M** | $\mathbf{94.18}_{\pm0.71}$ | $\mathbf{98.71}_{\pm0.19}$ | $\mathbf{76.52}_{\pm1.64}$ | $\mathbf{99.73}_{\pm0.23}$ | $\mathbf{89.98}_{\pm0.70}$ | $\mathbf{99.02}_{\pm0.31}$ | **93.02** |
| | Entropy | $\mathbf{72.48}_{\pm1.24}$ | $35.35_{\pm1.65}$ | $\mathbf{61.92}_{\pm1.32}$ | $50.96_{\pm1.57}$ | $23.76_{\pm1.05}$ | $17.55_{\pm0.92}$ | 43.67 |
| | **Entropy-M** | $49.80_{\pm2.95}$ | $\mathbf{56.16}_{\pm26.44}$ | $52.47_{\pm1.86}$ | $\mathbf{53.09}_{\pm16.63}$ | $\mathbf{56.78}_{\pm31.30}$ | $\mathbf{58.33}_{\pm37.31}$ | **54.44** |
| | DetectGPT | $75.62_{\pm1.20}$ | $57.18_{\pm1.91}$ | $77.25_{\pm0.66}$ | $61.97_{\pm0.90}$ | $58.01_{\pm1.75}$ | $20.42_{\pm1.07}$ | 58.41 |
| | **DetectGPT-M** | $\mathbf{80.34}_{\pm21.91}$ | $\mathbf{73.22}_{\pm28.90}$ | $\mathbf{81.89}_{\pm11.41}$ | $\mathbf{91.04}_{\pm8.24}$ | $57.53_{\pm18.26}$ | $\mathbf{66.30}_{\pm30.95}$ | **75.05** |
| | FastGPT | $76.50_{\pm1.39}$ | $39.56_{\pm0.54}$ | $82.45_{\pm0.47}$ | $33.21_{\pm1.82}$ | $41.31_{\pm0.89}$ | $11.29_{\pm0.49}$ | 47.39 |
| | **FastGPT-M** | $\mathbf{96.47}_{\pm0.30}$ | $\mathbf{90.23}_{\pm0.73}$ | $\mathbf{96.24}_{\pm0.37}$ | $\mathbf{92.37}_{\pm0.72}$ | $\mathbf{67.04}_{\pm1.07}$ | $56.34_{\pm1.90}$ | **83.12** |
| | DNA-GPT | $99.19_{\pm0.21}$ | $99.74_{\pm0.09}$ | $98.90_{\pm0.24}$ | $99.83_{\pm0.07}$ | $99.21_{\pm0.16}$ | $99.80_{\pm0.08}$ | 99.45 |
| | **DNA-GPT-M** | $\mathbf{99.66}_{\pm0.19}$ | $\mathbf{99.84}_{\pm0.13}$ | $98.81_{\pm0.22}$ | $\mathbf{99.91}_{\pm0.10}$ | $\mathbf{99.47}_{\pm0.20}$ | $\mathbf{99.86}_{\pm0.14}$ | **99.59** |

Table 15: Performance on Reuters dataset. The detection models are trained on text generated by Claude.

| Metric | Method | GPT4All | ChatGPT | ChatGPT-turbo | ChatGLM | StableLM | Claude | Avg. |
|---|---|---|---|---|---|---|---|---|
| TPR@FPR-1% | Likelihood | $19.82_{\pm2.17}$ | $81.07_{\pm2.05}$ | $12.53_{\pm1.35}$ | $94.44_{\pm1.10}$ | $19.42_{\pm3.17}$ | $90.36_{\pm1.06}$ | 52.94 |
| | **Likelihood-M** | $\mathbf{60.18}_{\pm5.26}$ | $\mathbf{84.84}_{\pm2.91}$ | $\mathbf{32.13}_{\pm2.78}$ | $91.78_{\pm1.51}$ | $\mathbf{34.58}_{\pm4.58}$ | $\mathbf{91.47}_{\pm1.34}$ | **65.83** |
| | Log-Rank | $29.96_{\pm3.29}$ | $86.22_{\pm1.87}$ | $15.20_{\pm1.70}$ | $97.33_{\pm0.64}$ | $24.58_{\pm3.35}$ | $93.20_{\pm1.13}$ | 57.75 |
| | **Log-Rank-M** | $\mathbf{71.33}_{\pm4.00}$ | $\mathbf{89.82}_{\pm2.47}$ | $\mathbf{37.60}_{\pm1.64}$ | $97.07_{\pm1.44}$ | $\mathbf{34.84}_{\pm5.56}$ | $\mathbf{94.36}_{\pm0.87}$ | **70.84** |
| | Entropy | $0.09_{\pm0.11}$ | $2.49_{\pm1.25}$ | $2.04_{\pm0.57}$ | $1.69_{\pm0.54}$ | $4.36_{\pm1.81}$ | $9.60_{\pm2.94}$ | 3.38 |
| | **Entropy-M** | $\mathbf{5.51}_{\pm2.56}$ | $\mathbf{17.42}_{\pm5.92}$ | $\mathbf{13.24}_{\pm2.62}$ | $\mathbf{7.91}_{\pm2.58}$ | $\mathbf{18.00}_{\pm4.28}$ | $\mathbf{34.09}_{\pm8.23}$ | **16.03** |
| | DetectGPT | $0.00_{\pm0.00}$ | $0.00_{\pm0.00}$ | $0.00_{\pm0.00}$ | $0.00_{\pm0.00}$ | $0.44_{\pm0.40}$ | $0.04_{\pm0.09}$ | 0.08 |
| | **DetectGPT-M** | $\mathbf{39.91}_{\pm20.48}$ | $\mathbf{51.60}_{\pm22.14}$ | $\mathbf{28.80}_{\pm20.74}$ | $\mathbf{57.47}_{\pm35.18}$ | $6.00_{\pm3.22}$ | $\mathbf{78.53}_{\pm8.07}$ | **43.72** |
| | FastGPT | $0.31_{\pm0.23}$ | $2.04_{\pm0.86}$ | $0.04_{\pm0.09}$ | $3.16_{\pm1.03}$ | $1.42_{\pm0.57}$ | $23.51_{\pm4.15}$ | 5.08 |
| | **FastGPT-M** | $\mathbf{59.96}_{\pm10.36}$ | $\mathbf{33.11}_{\pm28.98}$ | $\mathbf{56.09}_{\pm16.45}$ | $\mathbf{42.44}_{\pm27.14}$ | $\mathbf{8.62}_{\pm15.04}$ | $19.64_{\pm36.52}$ | **36.64** |
| | DNA-GPT | $3.64_{\pm1.52}$ | $6.58_{\pm1.66}$ | $2.18_{\pm1.15}$ | $7.78_{\pm0.95}$ | $2.44_{\pm2.03}$ | $7.20_{\pm1.15}$ | 4.97 |
| | **DNA-GPT-M** | $\mathbf{6.67}_{\pm2.96}$ | $\mathbf{7.16}_{\pm3.18}$ | $\mathbf{4.22}_{\pm2.00}$ | $7.51_{\pm3.26}$ | $\mathbf{5.38}_{\pm2.87}$ | $7.16_{\pm3.17}$ | **6.35** |
| AUROC | Likelihood | $75.19_{\pm1.39}$ | $97.55_{\pm0.33}$ | $58.77_{\pm1.77}$ | $99.62_{\pm0.10}$ | $86.45_{\pm0.70}$ | $98.42_{\pm0.25}$ | 86.00 |
| | **Likelihood-M** | $\mathbf{91.12}_{\pm2.58}$ | $\mathbf{97.69}_{\pm0.49}$ | $\mathbf{69.45}_{\pm3.64}$ | $98.38_{\pm0.57}$ | $\mathbf{90.57}_{\pm0.59}$ | $\mathbf{98.54}_{\pm0.43}$ | **90.96** |
| | Log-Rank | $78.40_{\pm1.23}$ | $97.77_{\pm0.32}$ | $58.22_{\pm2.03}$ | $99.61_{\pm0.19}$ | $85.68_{\pm0.67}$ | $98.67_{\pm0.24}$ | 86.39 |
| | **Log-Rank-M** | $\mathbf{95.06}_{\pm0.64}$ | $\mathbf{98.59}_{\pm0.28}$ | $\mathbf{76.35}_{\pm1.49}$ | $\mathbf{99.65}_{\pm0.22}$ | $\mathbf{90.36}_{\pm0.68}$ | $\mathbf{98.98}_{\pm0.27}$ | **93.16** |
| | Entropy | $27.52_{\pm1.24}$ | $64.65_{\pm1.65}$ | $38.08_{\pm1.32}$ | $49.04_{\pm1.57}$ | $76.24_{\pm1.05}$ | $82.45_{\pm0.92}$ | 56.33 |
| | **Entropy-M** | $\mathbf{53.47}_{\pm2.35}$ | $\mathbf{79.33}_{\pm1.73}$ | $\mathbf{55.92}_{\pm1.77}$ | $\mathbf{68.50}_{\pm2.29}$ | $\mathbf{83.03}_{\pm1.03}$ | $\mathbf{90.09}_{\pm0.94}$ | **71.72** |
| | DetectGPT | $75.62_{\pm1.20}$ | $57.18_{\pm1.91}$ | $77.25_{\pm0.66}$ | $61.97_{\pm0.90}$ | $58.01_{\pm1.75}$ | $20.42_{\pm1.07}$ | 58.41 |
| | **DetectGPT-M** | $\mathbf{91.23}_{\pm5.89}$ | $\mathbf{94.13}_{\pm1.77}$ | $\mathbf{81.29}_{\pm8.98}$ | $\mathbf{94.63}_{\pm5.95}$ | $\mathbf{70.99}_{\pm6.72}$ | $\mathbf{96.97}_{\pm1.09}$ | **88.21** |
| | FastGPT | $23.50_{\pm1.39}$ | $60.44_{\pm0.54}$ | $17.55_{\pm0.47}$ | $66.79_{\pm1.82}$ | $58.69_{\pm0.89}$ | $88.71_{\pm0.49}$ | 52.61 |
| | **FastGPT-M** | $\mathbf{96.53}_{\pm0.51}$ | $\mathbf{91.64}_{\pm3.61}$ | $\mathbf{94.46}_{\pm2.87}$ | $\mathbf{93.63}_{\pm3.03}$ | $\mathbf{71.74}_{\pm10.00}$ | $63.82_{\pm17.64}$ | **85.30** |
| | DNA-GPT | $65.77_{\pm1.29}$ | $78.38_{\pm0.64}$ | $54.85_{\pm1.20}$ | $82.40_{\pm0.73}$ | $65.30_{\pm0.67}$ | $81.64_{\pm0.76}$ | 71.39 |
| | **DNA-GPT-M** | $\mathbf{78.84}_{\pm12.13}$ | $\mathbf{83.32}_{\pm13.75}$ | $\mathbf{68.13}_{\pm8.88}$ | $\mathbf{85.53}_{\pm14.38}$ | $\mathbf{72.95}_{\pm9.83}$ | $\mathbf{84.29}_{\pm14.01}$ | **78.84** |

Table 16: Performance on TruthfulQA dataset. The detection models are trained on text generated by GPT4.

| Metric | Method | GPT4 | ChatGPT-turbo | ChatGLM | Dolly | ChatGPT | StableLM | Avg. |
|---|---|---|---|---|---|---|---|---|
| TPR@FPR-1% | Likelihood | $38.45_{\pm3.34}$ | $41.67_{\pm1.51}$ | $73.54_{\pm7.83}$ | $17.74_{\pm2.72}$ | $52.01_{\pm10.78}$ | $50.14_{\pm8.09}$ | 45.59 |
| | **Likelihood-E** | $29.17_{\pm11.92}$ | $\mathbf{56.89}_{\pm19.43}$ | $\mathbf{75.91}_{\pm17.62}$ | $\mathbf{52.17}_{\pm2.62}$ | $42.04_{\pm19.77}$ | $\mathbf{54.79}_{\pm10.29}$ | **51.83** |
| | Log-Rank | $40.17_{\pm4.46}$ | $39.77_{\pm7.89}$ | $83.22_{\pm4.08}$ | $16.45_{\pm5.07}$ | $54.50_{\pm2.09}$ | $59.26_{\pm2.20}$ | 48.9 |
| | **Log-Rank-E** | $\mathbf{47.74}_{\pm1.72}$ | $\mathbf{75.39}_{\pm4.05}$ | $\mathbf{90.55}_{\pm2.10}$ | $44.48_{\pm4.99}$ | $\mathbf{83.57}_{\pm1.73}$ | $\mathbf{79.26}_{\pm2.52}$ | **70.16** |
| | Entropy | $24.30_{\pm4.47}$ | $38.39_{\pm6.85}$ | $60.48_{\pm15.23}$ | $18.38_{\pm6.58}$ | $38.02_{\pm9.81}$ | $32.29_{\pm14.76}$ | 35.31 |
| | **Entropy-E** | $\mathbf{39.94}_{\pm3.25}$ | $\mathbf{78.44}_{\pm6.63}$ | $\mathbf{84.65}_{\pm18.12}$ | $\mathbf{52.70}_{\pm13.35}$ | $\mathbf{78.87}_{\pm9.18}$ | $\mathbf{69.52}_{\pm11.82}$ | **67.35** |
| | DetectGPT | $21.66_{\pm7.94}$ | $42.65_{\pm6.28}$ | $60.82_{\pm6.22}$ | $13.75_{\pm3.45}$ | $52.41_{\pm1.77}$ | $39.49_{\pm4.15}$ | 38.46 |
| | **DetectGPT-E** | $\mathbf{46.30}_{\pm2.60}$ | $\mathbf{82.31}_{\pm4.48}$ | $\mathbf{94.85}_{\pm0.92}$ | $\mathbf{59.52}_{\pm4.78}$ | $\mathbf{86.91}_{\pm3.54}$ | $\mathbf{84.14}_{\pm2.65}$ | **75.67** |
| | FastGPT | $21.20_{\pm4.54}$ | $47.15_{\pm3.59}$ | $66.61_{\pm6.47}$ | $20.63_{\pm4.37}$ | $52.58_{\pm9.86}$ | $49.92_{\pm6.27}$ | 43.01 |
| | **FastGPT-E** | $\mathbf{37.25}_{\pm4.38}$ | $\mathbf{74.64}_{\pm4.96}$ | $\mathbf{89.80}_{\pm5.01}$ | $\mathbf{49.06}_{\pm2.23}$ | $\mathbf{83.80}_{\pm4.03}$ | $\mathbf{76.32}_{\pm5.07}$ | **68.48** |
| | **DNAGPT** | $10.37_{\pm3.51}$ | $15.62_{\pm4.68}$ | $30.59_{\pm5.19}$ | $8.28_{\pm2.27}$ | $20.74_{\pm5.36}$ | $22.04_{\pm1.97}$ | **17.94** |
| | **DNAGPT** | $8.08_{\pm2.57}$ | $\mathbf{15.68}_{\pm6.48}$ | $20.39_{\pm3.96}$ | $11.52_{\pm6.76}$ | $21.47_{\pm10.11}$ | $13.99_{\pm6.07}$ | 15.19 |
| AUROC | Likelihood | $84.67_{\pm0.88}$ | $91.95_{\pm0.30}$ | $97.08_{\pm0.29}$ | $79.28_{\pm0.48}$ | $\mathbf{95.65}_{\pm0.40}$ | $94.17_{\pm0.30}$ | 90.47 |
| | **Likelihood-E** | $\mathbf{86.61}_{\pm0.56}$ | $92.72_{\pm2.67}$ | $\mathbf{97.14}_{\pm0.92}$ | $\mathbf{83.70}_{\pm0.57}$ | $94.20_{\pm1.87}$ | $\mathbf{94.37}_{\pm0.92}$ | **91.46** |
| | Log-Rank | $82.55_{\pm0.80}$ | $90.60_{\pm0.30}$ | $97.03_{\pm0.31}$ | $78.08_{\pm0.70}$ | $95.11_{\pm0.44}$ | $94.09_{\pm0.38}$ | 89.57 |
| | **Log-Rank-E** | $84.16_{\pm0.43}$ | $\mathbf{93.54}_{\pm0.60}$ | $\mathbf{97.56}_{\pm0.58}$ | $82.14_{\pm0.91}$ | $\mathbf{96.36}_{\pm0.58}$ | $\mathbf{95.10}_{\pm0.51}$ | **91.48** |
| | Entropy | $79.29_{\pm0.53}$ | $92.19_{\pm0.39}$ | $96.02_{\pm0.49}$ | $80.50_{\pm0.40}$ | $94.04_{\pm0.88}$ | $91.30_{\pm0.58}$ | 88.89 |
| | **Entropy-E** | $\mathbf{85.48}_{\pm0.50}$ | $\mathbf{95.17}_{\pm0.79}$ | $\mathbf{97.45}_{\pm1.27}$ | $\mathbf{84.57}_{\pm0.93}$ | $\mathbf{96.83}_{\pm0.88}$ | $\mathbf{95.42}_{\pm0.84}$ | **92.49** |
| | DetectGPT | $75.73_{\pm1.30}$ | $86.86_{\pm0.70}$ | $92.84_{\pm0.66}$ | $72.25_{\pm1.87}$ | $92.84_{\pm1.08}$ | $86.70_{\pm0.65}$ | 84.54 |
| | **DetectGPT-E** | $70.03_{\pm0.91}$ | $\mathbf{90.38}_{\pm1.10}$ | $\mathbf{97.23}_{\pm0.41}$ | $\mathbf{78.97}_{\pm0.76}$ | $\mathbf{93.82}_{\pm1.07}$ | $\mathbf{91.37}_{\pm0.42}$ | **86.96** |
| | FastGPT | $82.61_{\pm1.26}$ | $90.99_{\pm0.86}$ | $96.26_{\pm0.29}$ | $81.63_{\pm0.76}$ | $94.38_{\pm0.65}$ | $91.84_{\pm0.57}$ | 89.62 |
| | **FastGPT-E** | $82.80_{\pm0.75}$ | $\mathbf{94.26}_{\pm0.28}$ | $\mathbf{97.68}_{\pm0.68}$ | $\mathbf{84.02}_{\pm0.99}$ | $\mathbf{96.47}_{\pm0.41}$ | $\mathbf{94.84}_{\pm1.08}$ | **91.68** |
| | DNAGPT | $77.49_{\pm1.00}$ | $79.64_{\pm1.25}$ | $88.69_{\pm0.51}$ | $67.60_{\pm0.40}$ | $84.15_{\pm1.03}$ | $83.46_{\pm0.91}$ | 80.17 |
| | **DNAGPT** | $\mathbf{83.92}_{\pm0.38}$ | $\mathbf{90.18}_{\pm0.76}$ | $\mathbf{92.96}_{\pm0.50}$ | $\mathbf{79.81}_{\pm1.07}$ | $\mathbf{92.16}_{\pm0.66}$ | $\mathbf{90.90}_{\pm0.46}$ | **88.32** |

Table 17: Performance on short texts on Essay. The detection models are trained on text generated by GPT4All.

| Metric | Method | GPT4All | ChatGPT | Dolly | ChatGLM | Claude | ChatGPT-turbo | avg |
|---|---|---|---|---|---|---|---|---|
| **TPR@FPR-1%** | Likelihood | $19.86_{\pm5.55}$ | $40.93_{\pm4.86}$ | $11.69_{\pm2.93}$ | $58.75_{\pm6.73}$ | $15.38_{\pm1.78}$ | $23.47_{\pm4.56}$ | 28.35 |
| | **Likelihood-E** | $\mathbf{30.80}_{\pm6.67}$ | $\mathbf{45.91}_{\pm7.98}$ | $\mathbf{14.99}_{\pm4.58}$ | $\mathbf{74.51}_{\pm4.38}$ | $\mathbf{18.98}_{\pm5.26}$ | $\mathbf{27.24}_{\pm6.18}$ | **35.40** |
| | Log-Rank | $15.85_{\pm2.24}$ | $36.49_{\pm6.97}$ | $9.98_{\pm1.81}$ | $55.58_{\pm5.15}$ | $11.33_{\pm3.00}$ | $19.07_{\pm5.24}$ | 24.72 |
| | **Log-Rank-E** | $\mathbf{33.30}_{\pm2.53}$ | $\mathbf{44.18}_{\pm4.07}$ | $\mathbf{13.27}_{\pm3.77}$ | $\mathbf{72.41}_{\pm2.13}$ | $\mathbf{17.60}_{\pm1.81}$ | $\mathbf{24.22}_{\pm2.48}$ | **34.16** |
| | Entropy | $0.77_{\pm0.31}$ | $4.62_{\pm2.03}$ | $1.43_{\pm1.14}$ | $1.70_{\pm1.13}$ | $3.07_{\pm1.77}$ | $3.47_{\pm1.83}$ | 2.51 |
| | **Entropy-E** | $\mathbf{1.96}_{\pm0.42}$ | $\mathbf{5.11}_{\pm1.82}$ | $\mathbf{2.86}_{\pm1.17}$ | $\mathbf{3.26}_{\pm1.79}$ | $2.89_{\pm1.79}$ | $3.11_{\pm1.63}$ | **3.20** |
| | DetectGPT | $8.25_{\pm2.34}$ | $13.42_{\pm2.70}$ | $7.16_{\pm3.80}$ | $18.57_{\pm4.32}$ | $9.73_{\pm4.84}$ | $8.58_{\pm2.09}$ | 10.95 |
| | **DetectGPT-E** | $\mathbf{23.92}_{\pm7.43}$ | $\mathbf{41.96}_{\pm6.95}$ | $\mathbf{15.32}_{\pm3.89}$ | $\mathbf{53.44}_{\pm7.22}$ | $\mathbf{22.36}_{\pm5.76}$ | $\mathbf{30.49}_{\pm5.42}$ | **31.25** |
| | FastGPT | $5.28_{\pm2.94}$ | $11.02_{\pm1.41}$ | $3.96_{\pm2.41}$ | $14.20_{\pm6.19}$ | $7.42_{\pm2.08}$ | $6.27_{\pm1.27}$ | 8.03 |
| | **FastGPT-E** | $\mathbf{13.71}_{\pm9.14}$ | $\mathbf{20.89}_{\pm1.26}$ | $\mathbf{8.78}_{\pm3.14}$ | $\mathbf{27.50}_{\pm4.97}$ | $\mathbf{15.73}_{\pm1.90}$ | $\mathbf{12.84}_{\pm2.35}$ | **16.58** |
| | DNAGPT | $8.02_{\pm1.30}$ | $20.04_{\pm4.42}$ | $10.02_{\pm2.96}$ | $35.98_{\pm5.07}$ | $6.84_{\pm1.77}$ | $11.11_{\pm2.36}$ | 15.34 |
| | **DNAGPT** | $\mathbf{13.85}_{\pm1.23}$ | $\mathbf{23.87}_{\pm4.14}$ | $\mathbf{16.09}_{\pm5.47}$ | $\mathbf{49.78}_{\pm4.61}$ | $\mathbf{8.09}_{\pm1.80}$ | $\mathbf{13.38}_{\pm2.56}$ | **20.84** |
| **AUROC** | Likelihood | $84.84_{\pm0.81}$ | $93.57_{\pm0.68}$ | $83.34_{\pm1.31}$ | $96.34_{\pm0.44}$ | $81.26_{\pm1.02}$ | $89.30_{\pm0.82}$ | 88.11 |
| | **Likelihood-E** | $\mathbf{86.95}_{\pm0.80}$ | $\mathbf{93.65}_{\pm0.76}$ | $\mathbf{84.16}_{\pm1.29}$ | $\mathbf{97.04}_{\pm0.32}$ | $\mathbf{83.43}_{\pm0.74}$ | $\mathbf{89.59}_{\pm0.87}$ | **89.14** |
| | Log-Rank | $82.90_{\pm0.94}$ | $92.29_{\pm0.84}$ | $81.18_{\pm1.11}$ | $95.97_{\pm0.52}$ | $77.58_{\pm0.92}$ | $87.83_{\pm0.92}$ | 86.29 |
| | **Log-Rank-E** | $\mathbf{85.88}_{\pm0.83}$ | $\mathbf{92.86}_{\pm0.78}$ | $\mathbf{83.27}_{\pm1.08}$ | $\mathbf{97.01}_{\pm0.40}$ | $\mathbf{81.40}_{\pm0.71}$ | $\mathbf{88.28}_{\pm0.93}$ | **88.12** |
| | Entropy | $58.86_{\pm1.57}$ | $72.56_{\pm1.03}$ | $62.13_{\pm0.86}$ | $66.71_{\pm1.77}$ | $67.08_{\pm1.13}$ | $73.88_{\pm0.80}$ | 66.87 |
| | **Entropy-E** | $\mathbf{62.02}_{\pm1.48}$ | $\mathbf{74.75}_{\pm1.22}$ | $\mathbf{65.11}_{\pm1.67}$ | $\mathbf{71.53}_{\pm1.72}$ | $\mathbf{70.74}_{\pm1.19}$ | $\mathbf{74.98}_{\pm1.01}$ | **69.86** |
| | DetectGPT | $78.67_{\pm1.06}$ | $84.93_{\pm0.44}$ | $76.56_{\pm0.97}$ | $89.40_{\pm0.75}$ | $79.27_{\pm0.74}$ | $81.54_{\pm0.96}$ | 81.73 |
| | **DetectGPT-E** | $\mathbf{86.13}_{\pm1.27}$ | $\mathbf{92.12}_{\pm0.78}$ | $\mathbf{82.57}_{\pm1.32}$ | $\mathbf{96.33}_{\pm0.63}$ | $\mathbf{83.40}_{\pm0.90}$ | $\mathbf{88.57}_{\pm0.75}$ | **88.19** |
| | FastGPT | $79.78_{\pm0.78}$ | $85.87_{\pm0.73}$ | $81.15_{\pm0.62}$ | $91.22_{\pm1.03}$ | $82.62_{\pm0.55}$ | $80.05_{\pm1.18}$ | 83.45 |
| | **FastGPT-E** | $\mathbf{86.30}_{\pm0.72}$ | $\mathbf{90.49}_{\pm0.69}$ | $\mathbf{86.03}_{\pm0.77}$ | $\mathbf{95.26}_{\pm0.78}$ | $\mathbf{88.05}_{\pm0.61}$ | $\mathbf{86.65}_{\pm0.68}$ | **88.80** |
| | DNAGPT | $79.59_{\pm0.70}$ | $86.99_{\pm1.18}$ | $79.45_{\pm1.14}$ | $90.86_{\pm0.80}$ | $76.84_{\pm1.03}$ | $83.48_{\pm1.08}$ | 82.87 |
| | **DNAGPT** | $\mathbf{83.46}_{\pm0.91}$ | $\mathbf{88.99}_{\pm1.03}$ | $\mathbf{83.06}_{\pm1.30}$ | $\mathbf{93.40}_{\pm0.58}$ | $\mathbf{80.65}_{\pm0.95}$ | $\mathbf{85.84}_{\pm1.02}$ | **85.90** |

Table 18: Enhancement results on more detectors in the Essay dataset. The detection models are trained on text generated by GPT4All.

| Method | GPT4All | ChatGPT | Dolly | ChatGLM | Claude | ChatGPT-turbo | Avg. |
|---|---|---|---|---|---|---|---|
| RepreGuard | $94.66_{\pm0.35}$ | $97.35_{\pm0.28}$ | $90.40_{\pm0.83}$ | $91.95_{\pm0.71}$ | $47.51_{\pm1.53}$ | $43.42_{\pm1.34}$ | 77.55 |
| RepreGuard-M | $\mathbf{95.46}_{\pm0.80}$ | $\mathbf{99.55}_{\pm0.18}$ | $90.55_{\pm0.70}$ | $\mathbf{99.95}_{\pm0.04}$ | $47.18_{\pm1.47}$ | $43.12_{\pm1.28}$ | **79.30** |
| Binoculars | $96.33_{\pm0.27}$ | $98.59_{\pm0.18}$ | $88.57_{\pm1.56}$ | $99.57_{\pm0.43}$ | $\mathbf{87.21}_{\pm0.43}$ | $\mathbf{98.84}_{\pm0.18}$ | 94.85 |
| Binoculars-M | $\mathbf{97.34}_{\pm0.13}$ | $\mathbf{98.65}_{\pm0.19}$ | $\mathbf{90.19}_{\pm1.38}$ | $97.68_{\pm2.43}$ | $86.80_{\pm0.45}$ | $98.82_{\pm0.18}$ | **94.91** |
| Lastde | $99.54_{\pm0.08}$ | $99.72_{\pm0.08}$ | $97.38_{\pm0.36}$ | $99.98_{\pm0.01}$ | $75.98_{\pm1.22}$ | $\mathbf{91.16}_{\pm0.72}$ | 93.96 |
| Lastde-M | $\mathbf{99.68}_{\pm0.07}$ | $\mathbf{99.77}_{\pm0.07}$ | $\mathbf{97.87}_{\pm0.35}$ | $\mathbf{99.99}_{\pm0.01}$ | $\mathbf{76.21}_{\pm1.29}$ | $90.96_{\pm0.75}$ | **94.08** |
| FourierGPT | $91.70_{\pm16.06}$ | $90.18_{\pm19.03}$ | $93.74_{\pm12.07}$ | $98.26_{\pm2.82}$ | $\mathbf{44.90}_{\pm6.18}$ | $\mathbf{42.11}_{\pm6.15}$ | 76.81 |
| FourierGPT-M | $\mathbf{97.10}_{\pm4.19}$ | $\mathbf{95.27}_{\pm6.46}$ | $\mathbf{97.64}_{\pm3.19}$ | $\mathbf{99.44}_{\pm0.58}$ | $41.14_{\pm6.81}$ | $41.08_{\pm6.90}$ | **78.61** |

Table 19: Enhancement results on more detectors in the Reuters dataset. The detection models are trained on text generated by GPT4All.

| Method | GPT4All | ChatGPT | Dolly | ChatGLM | Claude | ChatGPT-turbo | Avg. |
|---|---|---|---|---|---|---|---|
| RepreGuard | $97.21_{\pm0.29}$ | $\mathbf{98.32}_{\pm0.30}$ | $8.24_{\pm0.46}$ | $\mathbf{97.26}_{\pm0.27}$ | $21.81_{\pm0.62}$ | $44.32_{\pm1.80}$ | 61.19 |
| RepreGuard-M | $\mathbf{98.33}_{\pm0.35}$ | $97.87_{\pm0.21}$ | $\mathbf{48.12}_{\pm5.04}$ | $97.00_{\pm0.29}$ | $\mathbf{29.08}_{\pm1.08}$ | $\mathbf{50.59}_{\pm1.84}$ | **70.16** |
| Binoculars | $80.96_{\pm1.11}$ | $97.99_{\pm0.30}$ | $61.25_{\pm1.77}$ | $99.80_{\pm0.06}$ | $83.23_{\pm0.80}$ | $98.70_{\pm0.25}$ | 86.99 |
| Binoculars-M | $\mathbf{90.90}_{\pm0.73}$ | $\mathbf{98.49}_{\pm0.19}$ | $\mathbf{74.10}_{\pm0.95}$ | $\mathbf{99.86}_{\pm0.04}$ | $\mathbf{86.37}_{\pm0.75}$ | $\mathbf{98.72}_{\pm0.32}$ | **91.41** |
| Lastde | $99.21_{\pm0.18}$ | $99.67_{\pm0.04}$ | $94.25_{\pm0.63}$ | $99.83_{\pm0.07}$ | $83.27_{\pm0.49}$ | $\mathbf{99.09}_{\pm0.18}$ | 95.89 |
| Lastde-M | $\mathbf{99.55}_{\pm0.16}$ | $\mathbf{99.68}_{\pm0.05}$ | $\mathbf{96.09}_{\pm0.35}$ | $99.82_{\pm0.05}$ | $\mathbf{85.64}_{\pm0.59}$ | $99.00_{\pm0.19}$ | **96.63** |
| FourierGPT | $99.28_{\pm0.35}$ | $96.71_{\pm3.25}$ | $99.06_{\pm0.41}$ | $98.45_{\pm0.25}$ | $56.82_{\pm10.56}$ | $49.84_{\pm0.89}$ | 83.36 |
| FourierGPT-M | $\mathbf{99.36}_{\pm0.29}$ | $\mathbf{98.29}_{\pm0.72}$ | $\mathbf{99.29}_{\pm0.27}$ | $98.48_{\pm0.43}$ | $55.13_{\pm10.81}$ | $\mathbf{49.88}_{\pm1.12}$ | **83.40** |

Table 20: Enhancement results on more detectors in the DetectRL dataset. The detection models are trained on text generated by GPT4All. Note that DetectRL lacks paired data for RepreGuard; therefore, this method is not compared.

| Method | Llama-2-70b | ChatGPT | Google-PaLM | Avg. |
|---|---|---|---|---|
| Binoculars | $79.10_{\pm0.37}$ | $66.55_{\pm0.80}$ | $73.85_{\pm0.93}$ | 73.17 |
| Binoculars-M | $\mathbf{81.22}_{\pm0.39}$ | $\mathbf{67.43}_{\pm1.08}$ | $\mathbf{77.80}_{\pm0.79}$ | **75.49** |
| Lastde | $78.36_{\pm0.39}$ | $51.93_{\pm0.51}$ | $69.04_{\pm0.90}$ | 66.45 |
| Lastde-M | $\mathbf{78.70}_{\pm0.36}$ | $\mathbf{52.55}_{\pm0.43}$ | $\mathbf{71.06}_{\pm0.76}$ | **67.44** |
| FourierGPT | $58.11_{\pm7.29}$ | $50.57_{\pm3.63}$ | $59.99_{\pm8.53}$ | 56.23 |
| FourierGPT-M | $\mathbf{64.67}_{\pm8.42}$ | $\mathbf{52.64}_{\pm5.70}$ | $\mathbf{64.07}_{\pm7.25}$ | **60.46** |

Table 21: Performance (AUROC) on perturbation texts (Back-Translation and DeepWordBug) on DetectRL. All detectors are trained on Llama-2-70b texts.

| Perturbation | Method | Llama-2-70b | ChatGPT | Google-PaLM | Avg. |
|---|---|---|---|---|---|
| Back-Translation | Likelihood | $53.88_{\pm0.76}$ | $\mathbf{29.91}_{\pm1.34}$ | $48.00_{\pm0.58}$ | 43.93 |
| | Likelihood-M | $\mathbf{58.85}_{\pm0.95}$ | $29.71_{\pm1.26}$ | $\mathbf{55.80}_{\pm0.90}$ | $\mathbf{48.12}$ |
| | Log-Rank | $56.40_{\pm0.74}$ | $30.91_{\pm1.32}$ | $46.85_{\pm0.75}$ | 44.72 |
| | Log-Rank-M | $\mathbf{62.41}_{\pm0.61}$ | $\mathbf{32.17}_{\pm1.19}$ | $\mathbf{59.55}_{\pm0.69}$ | $\mathbf{51.37}$ |
| | Entropy | $46.43_{\pm1.06}$ | $\mathbf{32.54}_{\pm1.08}$ | $41.99_{\pm0.61}$ | 40.32 |
| | Entropy-M | $\mathbf{48.99}_{\pm1.33}$ | $31.64_{\pm1.18}$ | $\mathbf{46.16}_{\pm0.49}$ | $\mathbf{42.26}$ |
| | DetectGPT | $20.30_{\pm0.72}$ | $10.90_{\pm0.68}$ | $16.59_{\pm0.56}$ | 15.93 |
| | DetectGPT-M | $\mathbf{62.24}_{\pm2.17}$ | $\mathbf{38.63}_{\pm6.75}$ | $\mathbf{61.72}_{\pm4.74}$ | $\mathbf{54.20}$ |
| | FastGPT | $35.37_{\pm1.40}$ | $\mathbf{24.10}_{\pm1.08}$ | $31.23_{\pm0.57}$ | 30.23 |
| | FastGPT-M | $\mathbf{51.87}_{\pm3.25}$ | $21.35_{\pm3.21}$ | $\mathbf{53.16}_{\pm3.04}$ | $\mathbf{42.13}$ |
| | DNA-GPT | $70.82_{\pm0.84}$ | $\mathbf{55.34}_{\pm0.73}$ | $65.37_{\pm0.51}$ | 63.84 |
| | DNA-GPT-M | $\mathbf{71.83}_{\pm0.94}$ | $54.53_{\pm1.32}$ | $\mathbf{67.63}_{\pm0.62}$ | $\mathbf{64.67}$ |
| DeepWordBug | Likelihood | $69.62_{\pm0.71}$ | $51.56_{\pm0.73}$ | $62.19_{\pm1.42}$ | 61.12 |
| | Likelihood-M | $\mathbf{85.71}_{\pm1.22}$ | $\mathbf{68.11}_{\pm1.64}$ | $\mathbf{79.18}_{\pm0.63}$ | $\mathbf{77.67}$ |
| | Log-Rank | $72.76_{\pm0.64}$ | $50.60_{\pm0.44}$ | $62.64_{\pm1.86}$ | 62 |
| | Log-Rank-M | $\mathbf{90.93}_{\pm1.14}$ | $\mathbf{71.16}_{\pm2.75}$ | $\mathbf{83.41}_{\pm1.12}$ | $\mathbf{81.84}$ |
| | Entropy | $65.85_{\pm0.97}$ | $63.63_{\pm1.71}$ | $58.00_{\pm1.31}$ | 62.49 |
| | Entropy-M | $\mathbf{69.35}_{\pm0.95}$ | $\mathbf{66.54}_{\pm1.58}$ | $\mathbf{63.95}_{\pm1.25}$ | $\mathbf{66.61}$ |
| | DetectGPT | $34.33_{\pm2.82}$ | $33.28_{\pm3.72}$ | $32.72_{\pm2.14}$ | 33.44 |
| | DetectGPT-M | $\mathbf{82.95}_{\pm1.70}$ | $\mathbf{64.13}_{\pm2.85}$ | $\mathbf{79.67}_{\pm2.41}$ | $\mathbf{75.58}$ |
| | FastGPT | $39.57_{\pm2.14}$ | $\mathbf{34.59}_{\pm2.49}$ | $36.51_{\pm2.30}$ | 36.89 |
| | FastGPT-M | $\mathbf{53.71}_{\pm1.90}$ | $30.93_{\pm2.38}$ | $\mathbf{54.63}_{\pm2.88}$ | $\mathbf{46.42}$ |
| | DNA-GPT | $67.76_{\pm2.07}$ | $58.74_{\pm1.64}$ | $63.69_{\pm1.06}$ | 63.4 |
| | DNA-GPT-M | $\mathbf{71.43}_{\pm6.30}$ | $\mathbf{59.59}_{\pm4.16}$ | $\mathbf{67.61}_{\pm4.09}$ | $\mathbf{66.21}$ |

Table 22: Performance and running time comparison with HMM-based method on Essay. The detection models are traind on text generated by GPT4All.

| Method | Detection Performance | | | | | | | Running Time | |
|---|---|---|---|---|---|---|---|---|---|
| | GPT4All | ChatGPT | Dolly | ChatGLM | Claude | ChatGPT-turbo | Avg. | Train | Inference |
| Likelihood | 96.16 | 98.79 | 90.90 | 99.29 | 92.76 | 99.13 | 96.17 | 13.58 | 61.21 |
| Likelihood-HMM | 96.39 | 98.87 | 91.24 | 99.33 | 92.91 | 99.17 | 96.32 | 20.64 | 90.63 |
| **Likelihood-M** | **98.58** | **99.47** | **94.59** | **99.54** | **94.82** | **99.72** | **97.79** | 15.01 | 72.67 |
| Log-Rank | 96.55 | 98.95 | 90.08 | 99.36 | 92.01 | 99.24 | 96.03 | 15.98 | 75.60 |
| Log-Rank-HMM | 96.68 | 98.99 | 90.24 | 99.38 | 92.08 | 99.27 | 96.10 | 21.77 | 103.70 |
| **Log-Rank-M** | **98.57** | **99.41** | **93.82** | **99.55** | **92.91** | **99.64** | **97.32** | 17.40 | 87.55 |
| Entropy | 74.19 | 89.49 | 73.26 | 84.11 | 86.58 | 95.94 | 83.93 | 13.26 | 64.49 |
| Entropy-HMM | 74.47 | 89.70 | 73.56 | 84.39 | 86.61 | 95.98 | 84.12 | 16.87 | 88.26 |
| **Entropy-M** | **83.52** | **93.28** | **81.11** | **91.44** | **87.96** | **96.75** | **89.01** | 14.69 | 73.66 |

Table 23: Performance (AUROC) on TOCSIN-enhanced baselines and our method on top of TOCSIN (suffix "-TM"). Detectors are trained on GPT4All texts.

| Method | GPT4 | ChatGPT-turbo | ChatGLM | Dolly | ChatGPT | StableLM | Avg. |
|---|---|---|---|---|---|---|---|
| Likelihood-T | $90.43_{\pm1.03}$ | $98.66_{\pm0.19}$ | $89.50_{\pm0.79}$ | $98.52_{\pm0.55}$ | $90.46_{\pm0.71}$ | $98.44_{\pm0.31}$ | 94.33 |
| **Likelihood-TM** | $\mathbf{98.08}_{\pm0.19}$ | $\mathbf{99.19}_{\pm0.15}$ | $\mathbf{92.83}_{\pm1.06}$ | $\mathbf{99.01}_{\pm0.51}$ | $\mathbf{91.06}_{\pm1.26}$ | $\mathbf{99.20}_{\pm0.09}$ | **96.56** |
| Log-Rank-T | $93.96_{\pm0.46}$ | $98.78_{\pm0.17}$ | $89.90_{\pm0.52}$ | $98.69_{\pm0.52}$ | $91.01_{\pm0.48}$ | $98.57_{\pm0.27}$ | 95.15 |
| **Log-Rank-TM** | $\mathbf{98.07}_{\pm0.35}$ | $\mathbf{99.10}_{\pm0.16}$ | $\mathbf{92.32}_{\pm0.91}$ | $\mathbf{99.01}_{\pm0.43}$ | $\mathbf{91.46}_{\pm0.62}$ | $\mathbf{99.05}_{\pm0.13}$ | **96.50** |
| Entropy-T | $76.90_{\pm1.57}$ | $98.41_{\pm0.25}$ | $87.20_{\pm0.92}$ | $98.11_{\pm0.61}$ | $90.91_{\pm0.40}$ | $98.72_{\pm0.19}$ | 91.71 |
| **Entropy-TM** | $\mathbf{83.89}_{\pm0.82}$ | $98.34_{\pm0.35}$ | $\mathbf{87.23}_{\pm0.96}$ | $\mathbf{98.25}_{\pm0.53}$ | $89.70_{\pm0.44}$ | $98.63_{\pm0.19}$ | **92.67** |
| DetectGPT-T | $48.95_{\pm1.79}$ | $40.69_{\pm11.74}$ | $51.46_{\pm2.98}$ | $40.67_{\pm13.55}$ | $43.99_{\pm7.23}$ | $27.04_{\pm30.41}$ | 42.13 |
| **DetectGPT-TM** | $\mathbf{95.51}_{\pm2.81}$ | $\mathbf{93.27}_{\pm9.08}$ | $\mathbf{79.76}_{\pm14.11}$ | $\mathbf{97.67}_{\pm2.00}$ | $\mathbf{70.25}_{\pm11.65}$ | $\mathbf{93.59}_{\pm10.94}$ | **88.34** |
| FastGPT-T | $53.53_{\pm5.59}$ | $70.36_{\pm27.49}$ | $56.26_{\pm8.54}$ | $71.83_{\pm28.44}$ | $72.06_{\pm29.44}$ | $77.26_{\pm36.41}$ | 66.88 |
| **FastGPT-TM** | $\mathbf{87.73}_{\pm0.15}$ | $66.40_{\pm1.22}$ | $\mathbf{80.65}_{\pm0.81}$ | $\mathbf{75.36}_{\pm1.49}$ | $41.91_{\pm1.52}$ | $28.10_{\pm2.13}$ | 63.36 |
| DNAGPT-T | $98.32_{\pm0.21}$ | $97.97_{\pm0.19}$ | $96.60_{\pm0.27}$ | $98.34_{\pm0.17}$ | $95.13_{\pm0.46}$ | $97.93_{\pm0.18}$ | 97.38 |
| **DNAGPT-TM** | $\mathbf{99.56}_{\pm0.10}$ | $\mathbf{99.04}_{\pm0.18}$ | $\mathbf{97.70}_{\pm0.33}$ | $\mathbf{99.24}_{\pm0.11}$ | $\mathbf{96.30}_{\pm0.40}$ | $\mathbf{99.06}_{\pm0.19}$ | **98.48** |

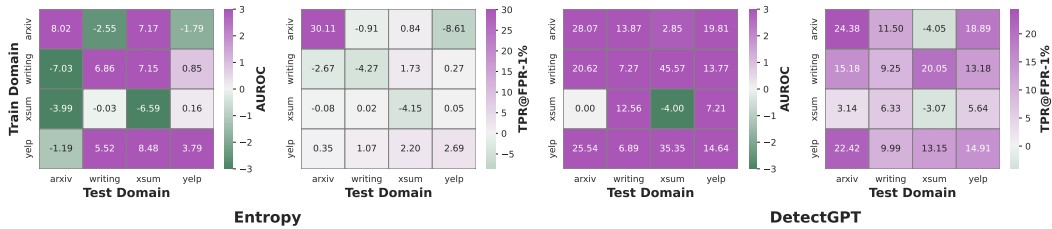

Figure 21: The performance improvement of the proposed method on Entropy and DetectGPT. Values greater than 0 indicate an enhanced effect.

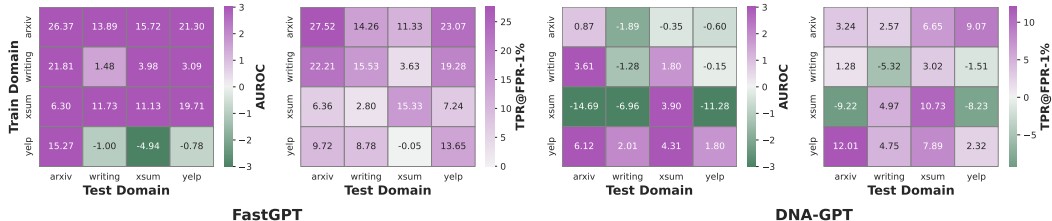

Figure 22: The performance improvement of the proposed method on FastGPT and DNA-GPT. Values greater than 0 indicate an enhanced effect.

technical content. We ensure that the use of large language models is responsible and adheres to academic and ethical standards.

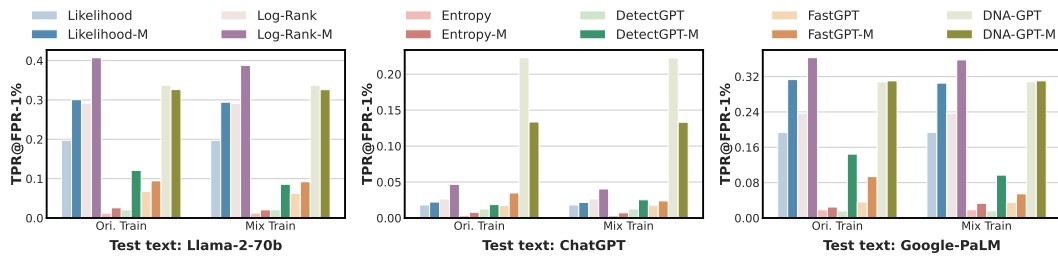

Figure 23: Detection performance concerning TPR@FPR-1% under different LLM mixed texts. All detectors are trained on Llama-2-70b texts.

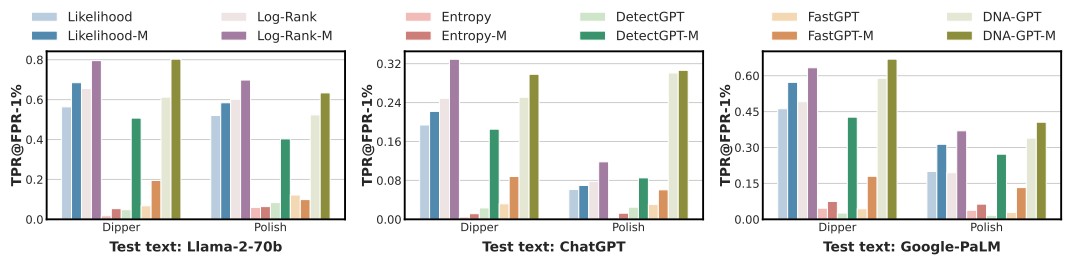

Figure 24: Detection performance concerning TPR@FPR-1% under different paraphrasing texts (Dipper and Polish). All detectors are trained on Llama-2-70b texts.

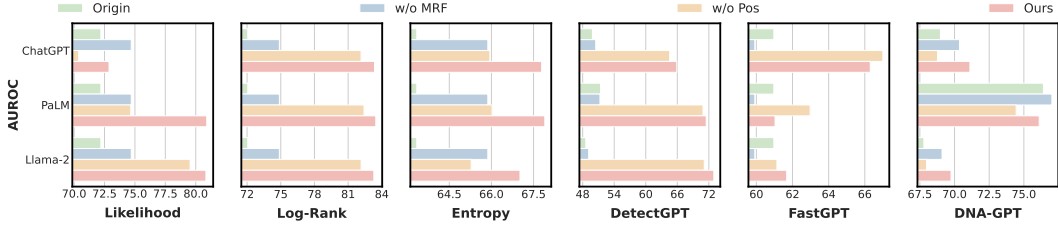

Figure 25: Ablation results on the DetectRL dataset. The y-axis represents the LLM text on which the detector was trained, and the x-axis represents the average performance across LLMs.

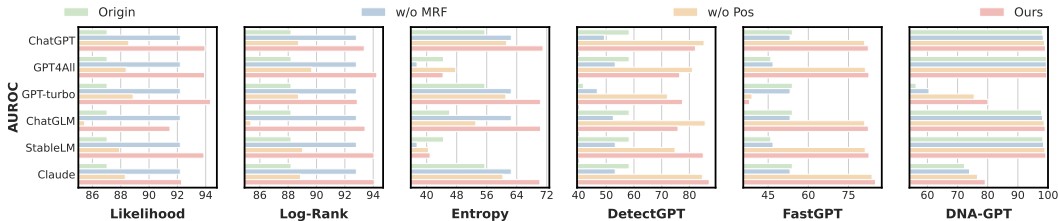

Figure 26: Ablation results on the Reuters dataset. The y-axis represents the LLM text on which the detector was trained, and the x-axis represents the average performance across LLMs.

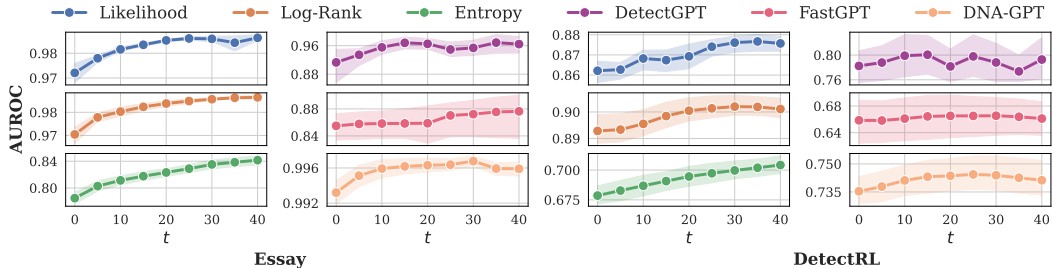

Figure 27: Detection performance of AUROC under different transition center $t_0$.

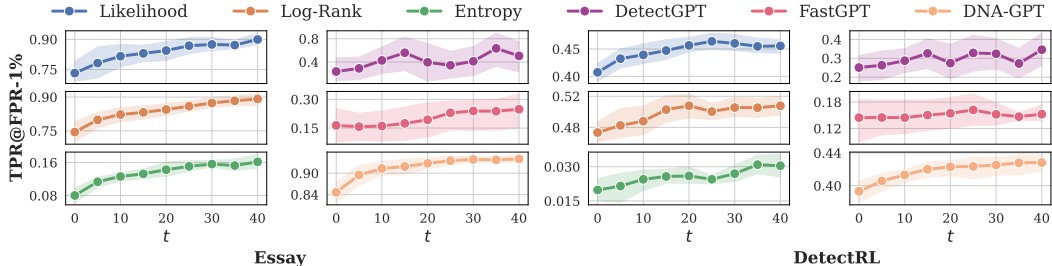

Figure 28: Detection performance of TPR@FPR-1% under different transition center $t_0$.

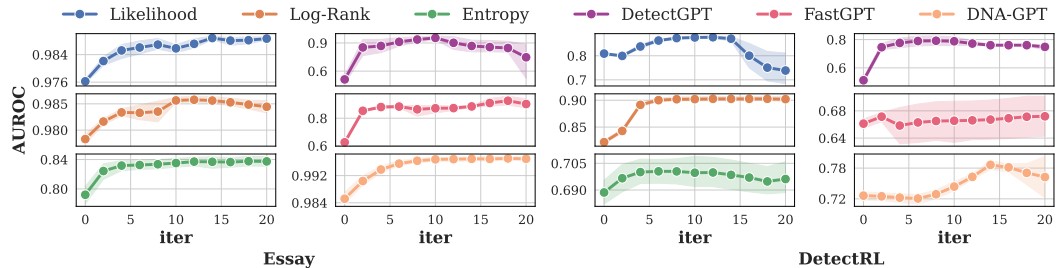

Figure 29: Detection performance of AUROC at different numbers of MRF layer iterations.

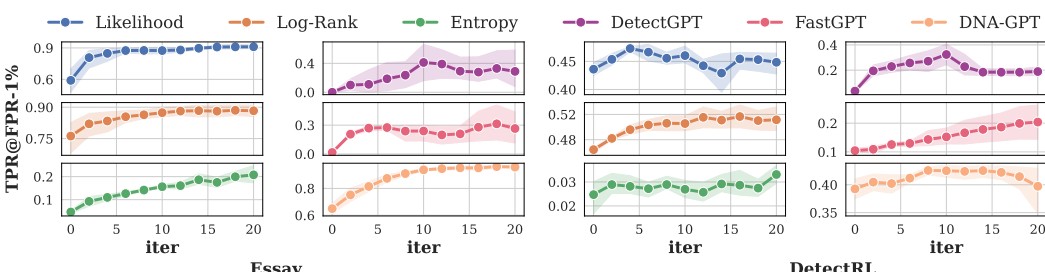

Figure 30: Detection performance of TPR@FPR-1% at different numbers of MRF layer iterations.

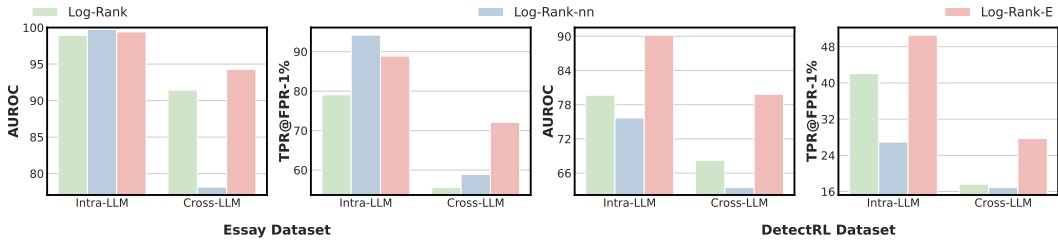

Figure 31: Comparison with NN-based methods. The detector used is Log-Rank.

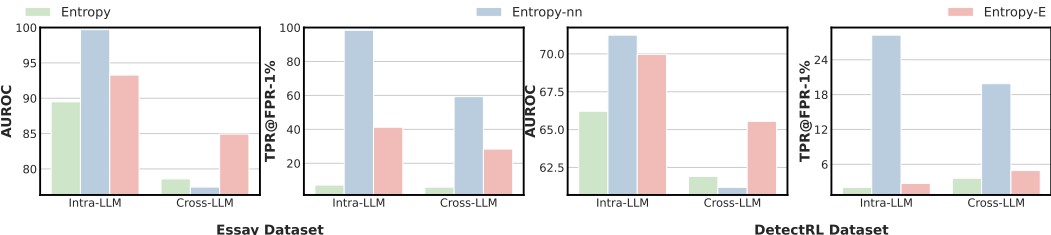

Figure 32: Comparison with NN-based methods. The detector used is Entropy.

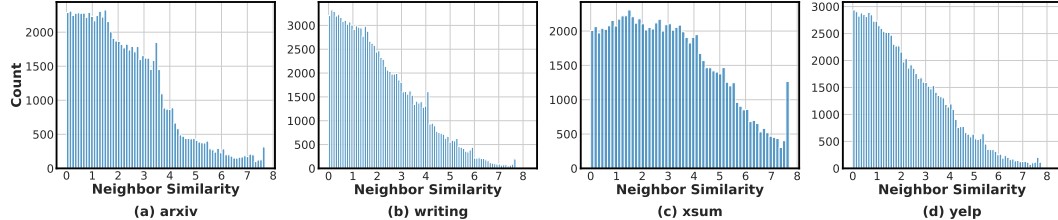

Figure 33: Statistical information on the distance of neighbor detection scores.

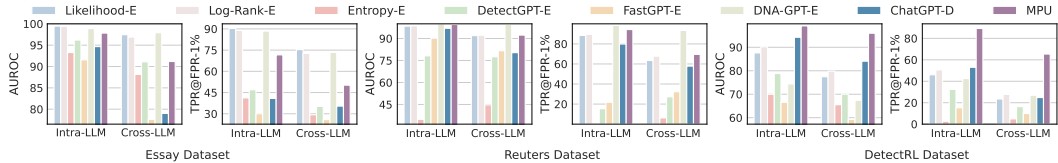

Figure 34: Comparison with Model-based detectors. Detectors are trained on GPT4All (Essay and Reuters) and Llama-2-70b (DetectRL).

Table 24: Running time (s) of training and inference phases.

| Method | Train | | | Inference | | |
|---|---|---|---|---|---|---|
| | Essay | Reuters | DetectRL | Essay | Reuters | DetectRL |
| Likelihood | 13.58 | 12.77 | 12.40 | 61.21 | 61.05 | 59.34 |
| **Likelihood-M** | 15.01 | 14.19 | 13.10 | 72.67 | 72.38 | 61.66 |
| Log-Rank | 15.98 | 14.38 | 14.61 | 75.60 | 72.99 | 66.20 |
| **Log-Rank-M** | 17.40 | 15.81 | 16.29 | 87.55 | 72.06 | 82.21 |
| Entropy | 13.26 | 12.93 | 13.79 | 64.49 | 63.69 | 63.19 |
| **Entropy-M** | 14.69 | 14.38 | 15.61 | 73.66 | 72.80 | 72.63 |
| DetectGPT | 204.32 | 216.36 | 370.44 | 924.63 | 982.55 | 1672.85 |
| **DetectGPT-M** | 206.44 | 218.73 | 373.48 | 929.46 | 984.70 | 1682.78 |
| FastGPT | 60.21 | 58.67 | 52.97 | 279.50 | 272.18 | 240.20 |
| **FastGPT-M** | 63.00 | 61.47 | 55.76 | 286.44 | 284.75 | 254.85 |
| DNA-GPT | 469.10 | 532.02 | 267.84 | 2113.65 | 2398.61 | 1207.18 |
| **DNA-GPT-M** | 471.42 | 534.47 | 271.11 | 2125.17 | 2405.37 | 1221.74 |

