# OpenReview forum: "Beyond Raw Detection Scores: Markov-Informed Calibration for Boosting Machine-Generated Text Detection"
_ICLR.cc/2026/Conference — ICLR 2026 Poster_

### Official Review · Reviewer_Qk3v · 2025-10-16

**Soundness:** 4
**Presentation:** 4
**Contribution:** 4
**Rating:** 8
**Confidence:** 4

**Summary:**

The authors make an important contribution to zero-shot detection of machine-generated text (MGT). Building on two well-substantiated phenomena—(i) adjacent tokens tend to have similar detection scores, and (ii) detection scores exhibit randomness with typically higher values at the beginning of a sequence—the paper proposes a lightweight calibration method for metric-based MGT detectors. The approach is backed by strong theoretical analysis and extensive empirical validation. The exposition is clear, the methodology is sound, and each claim is convincingly supported by evidence.

**Strengths:**

1. The paper is very well written and logically coherent, with polished and visually appealing figures and tables.
2. The empirical results are strong and substantiate the authors’ theoretical claims.
3. The appendix is exceptionally well organized (I particularly appreciate its structure--the table of contents makes the appendix easy to navigate).

**Weaknesses:**

This paper, in my view, has no particularly obvious weaknesses, and I am casting a clear accept vote. A few minor improvements could further enhance the paper’s quality, though they are not required:

1. **Analyze failure modes.** In Figure 4, I observe performance drops in certain domains. Could the authors provide an empirical analysis of the underlying causes of these degradations?
2. **Include more attack methods.** The current evaluation considers only paraphrasing, a common attack. Extending the analysis to additional attacks—such as back-translation and synonym substitution—would further strengthen the contribution.

**Questions:**

1. In multilingual settings, does your theory still hold?
2. Could you include an analysis of **Binoculars**, a SOTA zero-shot detection baseline?

---

> ### Author Response · Authors · 2025-11-22
> **Response to Reviewer Qk3v [1/3]**
>
> Thank you for your positive and valuable feedback, which we really cherish. Please see the below responses to your comments. Any further comments and discussions are welcome!
>
> > **[W1]** **Analyze failure modes.** In Figure 4, I observe performance drops in certain domains. Could the authors provide an empirical analysis of the underlying causes of these degradations?
> >
>
> Thank you for the reviewer’s insightful comments. Our analysis reveals that these failures mainly arise from **an asymmetry in neighbor similarity between the training and test domains**. Specifically, aside from a few random performance fluctuations, most failures occur when xsum (training data) is applied to arXiv, writing, and yelp (test data). Given that our proposed calibration strategy hinges on learning a reward/penalty matrix ($W_{mrf}$) to encourage neighbor similarity, we examined the distance of neighbor detection score across various domains, as summarized in Table 1 below (see Figure 33 of the revised version for the full distribution).
>
> As observed, **xsum exhibits considerably lower neighbor similarity** (larger distance), whereas the other domains show higher similarity. This makes sense because, as a news summarization dataset, XSum is extremely information-dense, with a highly compressed syntactic structure and a lack of redundancy and transitional words seen in longer texts (such as arXiv). Consequently, adjacent tokens in XSum experience more drastic fluctuations in detection scores. Therefore, when trained on xsum, the MRF learns parameters ($W_{mrf}$) that tolerate high local variance (treating it as a normal feature of the domain). When these parameters are applied to "smoother" domains like arXiv, the MRF fails to enforce the tighter consistency constraints required for those domains, leading to the observed performance drop. Encouragingly, compared to these occasional failure modes, the extensive enhancements highlight the practicality of our method.
>
> **Table 1. Statistical information on the distance of neighbor detection scores.**
>
> | Domain | mean | std |
> | --- | --- | --- |
> | arxiv | 2.26 | 1.64 |
> | writing | 2.15 | 1.58 |
> | xsum | 3.05 | 1.93 |
> | yelp | 2.21 | 1.65 |

---

> ### Author Response · Authors · 2025-11-22
> **Response to Reviewer Qk3v [2/3]**
>
> > **[W2]** **Include more attack methods.** The current evaluation considers only paraphrasing, a common attack. Extending the analysis to additional attacks—such as back-translation and synonym substitution—would further strengthen the contribution.
> >
>
> We sincerely thank the reviewer for this valuable suggestion. Following your suggestion, we have added additional experiments to our revised manuscript that cover two distinct attack methods in DetectRL dataset: **Back-Translation and Word Perturbation** (DeepWordBug). The results are shown in Tables 2 and 3 below (Table 21 in the revised version). Across both perturbation scenarios, detectors equipped with our Markov-informed calibration (e.g., Likelihood-M, Log-Rank-M) **consistently outperformed** the uncalibrated ones, typically by margins of 10-20%. These findings align perfectly with our existing evaluation on Paraphrasing Attacks presented in the initial submission, which shows that our method effectively mitigates the performance degradation typically caused by text perturbations.
>
> Our encouraging gain is consistent with the fundamental theory of Markov random fields. The adversarial perturbation acts as **token-level salt-and-pepper noise**, while our Markov-informed calibration strategy (modeling Neighbor Similarity) leverages the remaining uncorrupted context to **smooth out these local anomalies**, recovering the latent detection signal.
>
> **Table 2. Performance (AUROC) on Back-Translation texts on DetectRL. All detectors are trained on Llama-2-70b texts.**
>
> | Method | Llama-2-70b | ChatGPT | Google-PaLM | Avg. |
> | --- | --- | --- | --- | --- |
> | Likelihood | 53.88 | **29.91** | 48.00 | 43.93 |
> | **Likelihood-M** | **58.85** | 29.71 | **55.80** | **48.12** |
> | Log-Rank | 56.40 | 30.91 | 46.85 | 44.72 |
> | **Log-Rank-M** | **62.41** | **32.17** | **59.55** | **51.37** |
> | Entropy | 46.43 | **32.54** | 41.99 | 40.32 |
> | **Entropy-M** | **48.99** | 31.64 | **46.16** | **42.26** |
> | DetectGPT | 20.30 | 10.90 | 16.59 | 15.93 |
> | **DetectGPT-M** | **62.24** | **38.63** | **61.72** | **54.20** |
> | FastGPT | 35.37 | **24.10** | 31.23 | 30.23 |
> | **FastGPT-M** | **51.87** | 21.35 | **53.16** | **42.13** |
> | DNA-GPT | 70.82 | **55.34** | 65.37 | 63.84 |
> | **DNA-GPT-M** | **71.83** | 54.53 | **67.63** | **64.67** |
>
> **Table 3. Performance (AUROC) on word perturbation texts (DeepWordBug) on DetectRL. All detectors are trained on Llama-2-70b texts.**
>
> | Method | Llama-2-70b | ChatGPT | Google-PaLM | Avg. |
> | --- | --- | --- | --- | --- |
> | Likelihood | 69.62 | 51.56 | 62.19 | 61.12 |
> | **Likelihood-M** | **85.71** | **68.11** | **79.18** | **77.67** |
> | Log-Rank | 72.76 | 50.60 | 62.64 | 62.00 |
> | **Log-Rank-M** | **90.93** | **71.16** | **83.41** | **81.84** |
> | Entropy | 65.85 | 63.63 | 58.00 | 62.49 |
> | **Entropy-M** | **69.35** | **66.54** | **63.95** | **66.61** |
> | DetectGPT | 34.33 | 33.28 | 32.72 | 33.44 |
> | **DetectGPT-M** | **82.95** | **64.13** | **79.67** | **75.58** |
> | FastGPT | 39.57 | **34.59** | 36.51 | 36.89 |
> | **FastGPT-M** | **53.71** | 30.93 | **54.63** | **46.42** |
> | DNA-GPT | 67.76 | 58.74 | 63.69 | 63.40 |
> | **DNA-GPT-M** | **71.43** | **59.59** | **67.61** | **66.21** |
>
> > **[Q1]** In multilingual settings, does your theory still hold?
> >
>
> We thank the reviewer for the valuable question. we are happy to confirm that **yes, our theory and the observed properties hold in multilingual settings.** Our theory makes no assumptions about language types; therefore, our theoretical findings should persist in any language modeled by Transformer. Following your question, we have conducted additional experiments using the CUDRT (Chinese) dataset, where multiple LLMs, including Baichuan, ChatGLM, and Qwen, produce the machine-generated texts. As shown in Appendix E.2 (Figures 16 and 20) in the revised version, across all metrics (Log-Likelihood, Log-Rank, Entropy, DetectGPT) and all LLM texts, the properties of Neighbor Similarity and Initial Instability are still hold, which demonstrates that they are not artifacts of English grammar but are fundamental characteristics of LLMs.

---

> ### Author Response · Authors · 2025-11-22
> **Response to Reviewer Qk3v [3/3]**
>
> > **[Q2]** Could you include an analysis of Binoculars, a SOTA zero-shot detection baseline?
> >
>
> Per your suggestion, we have **integrated Binoculars [1]**,  and notably, we have also added more advanced **RepreGuard [2], Lastde [3], and FourierGPT [4],** and applied our strategy to them as “M” suffix. As shown in Tables 4-7 below (full results can be found in Tables 18-20 in the revised paper), our method serves as a universal plug-in that enhances these detectors. For example, on the Reuters dataset, our method improves RepreGuard by 8.97% and Binoculars by 4.42%. This demonstrates that, no matter how advanced and complex the detector’s scoring strategy is, our Markov-informed calibration method can still unlock further performance gains.
>
> The Fast-DetectGPT and DNAGPT in the initial submission represent the current state of technology. We believe that further adding these methods will more comprehensively demonstrate the strong generalizability of our approach. Thank you again for the reviewers' suggestions.
>
> **Table 4. Detection enhancement on the RepreGuard detector. The detectors are trained on GPT4All texts.**
>
> | Dataset | Detector | GPT4All | ChatGPT | Dolly | ChatGLM | Claude | ChatGPT-turbo | Avg. |
> | --- | --- | --- | --- | --- | --- | --- | --- | --- |
> | Essay | RepreGuard | 94.66 | 97.35 | 90.40 | 91.95 | 47.51 | 43.42 | 77.55 |
> | Essay | **RepreGuard-M** | 95.46 | 99.55 | 90.55 | 99.95 | 47.18 | 43.12 | 79.30 |
> | Reuters | RepreGuard | 97.21 | 98.32 | 8.24 | 97.26 | 21.81 | 44.32 | 61.19 |
> | Reuters | **RepreGuard-M** | 98.33 | 97.87 | 48.12 | 97.00 | 29.08 | 50.59 | 70.16 |
>
> **Table 5. Detection enhancement on the Binoculars detector.  The detectors are trained on GPT4All texts.**
>
> | Dataset | Detector | GPT4All | ChatGPT | Dolly | ChatGLM | Claude | ChatGPT-turbo | Avg. |
> | --- | --- | --- | --- | --- | --- | --- | --- | --- |
> | Essay | Binoculars | 96.33 | 98.59 | 88.57 | **99.57** | **87.21** | **98.84** | 94.85 |
> | Essay | **Binoculars-M** | **97.34** | **98.65** | **90.19** | 97.68 | 86.80 | **98.84** | **94.91** |
> | Reuters | Binoculars | 80.96 | 97.99 | 61.25 | 99.80 | 83.23 | 98.70 | 86.99 |
> | Reuters | **Binoculars-M** | **90.90** | **98.49** | **74.10** | **99.86** | **86.37** | **98.72** | **91.41** |
>
> **Table 6. Detection enhancement on the Lastde detector.  The detectors are trained on GPT4All texts.**
>
> | Dataset | Detector | GPT4All | ChatGPT | Dolly | ChatGLM | Claude | ChatGPT-turbo | Avg. |
> | --- | --- | --- | --- | --- | --- | --- | --- | --- |
> | Essay | Lastde | 99.54 | 99.72 | 97.38 | 99.98 | 75.98 | **91.16** | 93.96 |
> | Essay | **Lastde-M** | **99.68** | **99.77** | **97.87** | **99.99** | **76.21** | 90.96 | **94.08** |
> | Reuters | Lastde | 99.21 | 99.67 | 94.25 | **99.83** | 83.27 | **99.09** | 95.89 |
> | Reuters | **Lastde-M** | **99.55** | **99.68** | **96.09** | 99.82 | **85.64** | 99.00 | **96.63** |
>
> **Table 7. Detection enhancement on the FourierGPT detector.  The detectors are trained on GPT4All texts.**
>
> | Dataset | Detector | GPT4All | ChatGPT | Dolly | ChatGLM | Claude | ChatGPT-turbo | Avg. |
> | --- | --- | --- | --- | --- | --- | --- | --- | --- |
> | Essay | FourierGPT | 91.70 | 90.18 | 93.74 | 98.26 | **44.90** | **42.11** | 76.81 |
> | Essay | **FourierGPT-M** | **97.10** | **95.27** | **97.64** | **99.44** | 41.14 | 41.08 | **78.61** |
> | Reuters | FourierGPT | 99.28 | 96.71 | 99.06 | 98.45 | **56.82** | 49.84 | 83.36 |
> | Reuters | **FourierGPT-M** | **99.36** | **98.29** | **99.29** | **98.48** | 55.13 | **49.88** | **83.40** |
>
> [1] Spotting LLMs With Binoculars: Zero-Shot Detection of Machine-Generated Text, ICML 2024.
>
> [2] RepreGuard: Detecting LLM-Generated Text by Revealing Hidden Representation Patterns, TACL 2025.
>
> [3] Training-free LLM-generated Text Detection by Mining Token Probability Sequences, ICLR 2025.
>
> [4] Detecting Subtle Differences between Human and Model Languages Using Spectrum of Relative Likelihood. EMNLP 2024.

---

> > ### Comment · Reviewer_Qk3v · 2025-11-22
> > **Thank you!**
> >
> > Thank you for your detailed and prompt response, which addressed my concerns. I think my score is high enough compared with other reviewers, so I'd like to keep my score unchanged. Good luck!

---

> > > ### Author Response · Authors · 2025-11-22
> > >
> > > We sincerely thank the reviewer for the positive feedback and continued support! Your constructive comments were instrumental in improving our paper. Thank you for your time, and wish you the best in your research!

---

### Official Review · Reviewer_kDth · 2025-10-29

**Soundness:** 2
**Presentation:** 3
**Contribution:** 2
**Rating:** 4
**Confidence:** 4

**Summary:**

This paper addresses the issue that token-level detection scores in metric-based LLM-generated text detectors are susceptible to randomness, bias, and instability introduced by the LLM sampling process. To mitigate this, the authors propose a lightweight MRF (Markov Random Field) calibration module that models inter-token dependencies and positional weighting. Using Mean-Field Approximation, the method smooths and stabilizes detection scores, thus improving the reliability of existing statistical detection approaches.

**Strengths:**

- The paper clearly identifies the root cause of score bias in metric-based detectors, namely, randomness introduced by the LLM generation process.
- The proposed method is conceptually simple, computationally efficient, and effective.
- The experiments are broad and thorough, including challenging setups such as **DetectRL**, and report diagnostic metrics such as **TPR@FPR-1%** in addition to AUROC.

**Weaknesses:**

- Although the proposed method is intuitive and effective, there is concern about the practical significance of its improvement for statistical detection methods. As shown in Table 2, all metric-based detectors still perform poorly on the challenging DetectRL benchmark, far below the level required for real-world applications.
- The evaluated detectors are somewhat outdated. Incorporating more advanced methods such as Binoculars [1] and RepreGuard [2] could better showcase the effectiveness of the proposed approach.
- The proposed MRF module, while effective at mitigating random noise caused by the stochastic sampling process in LLM-generated texts, may not generalize well to adversarial perturbation attacks, such as random character insertion, token shuffling, or deletion. Moreover, I noticed that in the DetectRL setup, the authors only evaluate paraphrasing-based attacks, without considering perturbation-style adversarial attacks. Therefore, it is unclear whether the proposed method is fragile under such conditions. If this is the case, the authors should acknowledge this limitation and discuss potential solutions, which would strengthen the manuscript.
- The paper lacks comparison with other enhanced metric-based approaches, such as TOCSIN [3].

[1] Spotting LLMs With Binoculars: Zero-Shot Detection of Machine-Generated Text, ICML 2024

[2] RepreGuard: Detecting LLM-Generated Text by Revealing Hidden Representation Patterns, TACL 2025

[3] Zero-Shot Detection of LLM-Generated Text using Token Cohesiveness, EMNLP 2024

**Questions:**

See Weaknesses.

---

> ### Author Response · Authors · 2025-11-22
> **Response to Reviewer kDth [1/3]**
>
> Thank you for your timely and valuable feedback, which we really cherish. Please see the below responses to your comments. Any further comments and discussions are welcome!
>
> > **[W1]**  Although the proposed method is intuitive and effective, there is concern about the practical significance of its improvement for statistical detection methods. As shown in Table 2, all metric-based detectors still perform poorly on the challenging DetectRL benchmark, far below the level required for real-world applications.
> >
>
> We agree that DetectRL performs worse than the Essay dataset, reflecting the high difficulty of that benchmark. However, we respectfully argue that our method holds significant practical value precisely because it makes these statistical methods viable in scenarios where they were previously failing.
>
> - **Performance Improvements.** As DetectRL [1] states, it aims to provide “more stressful evaluation to drive the development of more efficient detectors” (Abstract in DetectRL [1]). In this context, the poor baseline performance actually highlights the significance of our contribution. For example, on DetectRL (Table 2 in the paper), the baseline DetectGPT achieves an AUROC of 48.60%, which is worse than random guessing. Our method boosts this to 72.95% (+24.35%). While 72.95% is not yet perfect, it transforms a detector that was **statistically useless** into one that provides a **meaningful signal**.
> - **Computational Efficiency.** The practical significance of a good detection method depends on the trade-off between accuracy and computational efficiency. While supervised classifiers (such as RoBERTa-based detectors) may achieve higher accuracy, they require time-consuming training. In contrast, statistical detection methods can be deployed immediately and flexibly without costly retraining. As shown in Table 16 in the initial submission (now Table 24 in the revised version), our MRF calibration layer adds **negligible computational overhead**. In real-world API environments that may handle millions or hundreds of millions of APIs, a lightweight calibration method (which can deliver a 10% to 20% accuracy improvement) is often more practically feasible than a complex, slow-running model-based detector.
> - **"Plug-and-Play" Calibration.** Our contribution is not a new detector, but acts as a **universal booster** for existing metric-based detectors. This means our method is **orthogonal** to future improvements in base metrics. We look forward to new and more powerful statistical metrics in the future, on which our calibration will continue to provide this **"free" performance boost**.
>
> > **[W2]**  The evaluated detectors are somewhat outdated. Incorporating more advanced methods such as Binoculars [1] and RepreGuard [2] could better showcase the effectiveness of the proposed approach.
> >
>
> Per your suggestion, we have **integrated state-of-the-art metric-based baselines**: RepreGuard [1], Binoculars [2], Lastde [3], and FourierGPT [4], and applied our strategy to them as “M” suffix. As shown in Tables 1-4 below (Table 18-20 in the revised version), our method serves as a universal plug-in that enhances these detectors. For example, on the Reuters dataset, our method improves RepreGuard by 8.97% and Binoculars by 4.42%. This demonstrates that, no matter how advanced and complex the detector’s scoring strategy is, our Markov-informed calibration method can still unlock further performance gains.
>
> The Fast-DetectGPT and DNAGPT in the initial submission represent the current state of technology. We believe that further adding these methods will more comprehensively demonstrate the strong generalizability of our approach.
>
> **Table 1. Detection enhancement on the RepreGuard detector. The detectors are trained on GPT4All texts.**
>
> | Dataset | Detector | GPT4All | ChatGPT | Dolly | ChatGLM | Claude | ChatGPT-turbo | Avg. |
> | --- | --- | --- | --- | --- | --- | --- | --- | --- |
> | Essay | RepreGuard | 94.66 | 97.35 | 90.40 | 91.95 | 47.51 | 43.42 | 77.55 |
> | Essay | **RepreGuard-M** | 95.46 | 99.55 | 90.55 | 99.95 | 47.18 | 43.12 | 79.30 |
> | Reuters | RepreGuard | 97.21 | 98.32 | 8.24 | 97.26 | 21.81 | 44.32 | 61.19 |
> | Reuters | **RepreGuard-M** | 98.33 | 97.87 | 48.12 | 97.00 | 29.08 | 50.59 | 70.16 |
>
> **Table 2. Detection enhancement on the Binoculars detector.  The detectors are trained on GPT4All texts.**
>
> | Dataset | Detector | GPT4All | ChatGPT | Dolly | ChatGLM | Claude | ChatGPT-turbo | Avg. |
> | --- | --- | --- | --- | --- | --- | --- | --- | --- |
> | Essay | Binoculars | 96.33 | 98.59 | 88.57 | **99.57** | **87.21** | **98.84** | 94.85 |
> | Essay | **Binoculars-M** | **97.34** | **98.65** | **90.19** | 97.68 | 86.80 | **98.84** | **94.91** |
> | Reuters | Binoculars | 80.96 | 97.99 | 61.25 | 99.80 | 83.23 | 98.70 | 86.99 |
> | Reuters | **Binoculars-M** | **90.90** | **98.49** | **74.10** | **99.86** | **86.37** | **98.72** | **91.41** |

---

> ### Author Response · Authors · 2025-11-22
> **Response to Reviewer kDth [2/3]**
>
> **Table 3. Detection enhancement on the Lastde detector.  The detectors are trained on GPT4All texts.**
>
> | Dataset | Detector | GPT4All | ChatGPT | Dolly | ChatGLM | Claude | ChatGPT-turbo | Avg. |
> | --- | --- | --- | --- | --- | --- | --- | --- | --- |
> | Essay | Lastde | 99.54 | 99.72 | 97.38 | 99.98 | 75.98 | **91.16** | 93.96 |
> | Essay | **Lastde-M** | **99.68** | **99.77** | **97.87** | **99.99** | **76.21** | 90.96 | **94.08** |
> | Reuters | Lastde | 99.21 | 99.67 | 94.25 | **99.83** | 83.27 | **99.09** | 95.89 |
> | Reuters | **Lastde-M** | **99.55** | **99.68** | **96.09** | 99.82 | **85.64** | 99.00 | **96.63** |
>
> **Table 4. Detection enhancement on the FourierGPT detector.  The detectors are trained on GPT4All texts.**
>
> | Dataset | Detector | GPT4All | ChatGPT | Dolly | ChatGLM | Claude | ChatGPT-turbo | Avg. |
> | --- | --- | --- | --- | --- | --- | --- | --- | --- |
> | Essay | FourierGPT | 91.70 | 90.18 | 93.74 | 98.26 | **44.90** | **42.11** | 76.81 |
> | Essay | **FourierGPT-M** | **97.10** | **95.27** | **97.64** | **99.44** | 41.14 | 41.08 | **78.61** |
> | Reuters | FourierGPT | 99.28 | 96.71 | 99.06 | 98.45 | **56.82** | 49.84 | 83.36 |
> | Reuters | **FourierGPT-M** | **99.36** | **98.29** | **99.29** | **98.48** | 55.13 | **49.88** | **83.40** |
>
> > **[W3]**  The proposed MRF module, while effective at mitigating random noise caused by the stochastic sampling process in LLM-generated texts, may not generalize well to adversarial perturbation attacks, such as random character insertion, token shuffling, or deletion. Moreover, I noticed that in the DetectRL setup, the authors only evaluate paraphrasing-based attacks, without considering perturbation-style adversarial attacks. Therefore, it is unclear whether the proposed method is fragile under such conditions. If this is the case, the authors should acknowledge this limitation and discuss potential solutions, which would strengthen the manuscript.
> >
>
> We sincerely thank the reviewer for this valuable suggestion. To address this concern, we have added additional experiments to our revised manuscript that cover two distinct attack methods in DetectRL dataset: **Back-Translation and Word Perturbation**. The results are shown in Tables 5 and 6 below (Table 21 in the revised version). Across both perturbation scenarios, detectors equipped with our Markov-informed calibration (e.g., Likelihood-M, Log-Rank-M) **consistently outperformed
> the uncalibrated ones**, typically by margins of 10-20%. These findings align perfectly with our existing evaluation on Paraphrasing Attacks presented in the initial submission, which shows that our method effectively mitigates the performance degradation typically caused by text perturbations.
>
> These encouraging gains are consistent with the fundamental theory of Markov random fields. The adversarial perturbation acts as **token-level salt-and-pepper noise**, while our Markov-informed calibration strategy (modeling Neighbor Similarity) leverages the remaining uncorrupted context to **smooth out these local anomalies**, recovering the latent detection signal.
>
> **Table 5. Performance (AUROC) on Back-Translation texts on DetectRL. All detectors are trained on Llama-2-70b texts.**
>
> | Method | Llama-2-70b | ChatGPT | Google-PaLM | Avg. |
> | --- | --- | --- | --- | --- |
> | Likelihood | 53.88 | **29.91** | 48.00 | 43.93 |
> | **Likelihood-M** | **58.85** | 29.71 | **55.80** | **48.12** |
> | Log-Rank | 56.40 | 30.91 | 46.85 | 44.72 |
> | **Log-Rank-M** | **62.41** | **32.17** | **59.55** | **51.37** |
> | Entropy | 46.43 | **32.54** | 41.99 | 40.32 |
> | **Entropy-M** | **48.99** | 31.64 | **46.16** | **42.26** |
> | DetectGPT | 20.30 | 10.90 | 16.59 | 15.93 |
> | **DetectGPT-M** | **62.24** | **38.63** | **61.72** | **54.20** |
> | FastGPT | 35.37 | **24.10** | 31.23 | 30.23 |
> | **FastGPT-M** | **51.87** | 21.35 | **53.16** | **42.13** |
> | DNA-GPT | 70.82 | **55.34** | 65.37 | 63.84 |
> | **DNA-GPT-M** | **71.83** | 54.53 | **67.63** | **64.67** |
>
> **Table 6. Performance (AUROC) on word perturbation texts (DeepWordBug) on DetectRL. All detectors are trained on Llama-2-70b texts.**
>
> | Method | Llama-2-70b | ChatGPT | Google-PaLM | Avg. |
> | --- | --- | --- | --- | --- |
> | Likelihood | 69.62 | 51.56 | 62.19 | 61.12 |
> | **Likelihood-M** | **85.71** | **68.11** | **79.18** | **77.67** |
> | Log-Rank | 72.76 | 50.60 | 62.64 | 62.00 |
> | **Log-Rank-M** | **90.93** | **71.16** | **83.41** | **81.84** |
> | Entropy | 65.85 | 63.63 | 58.00 | 62.49 |
> | **Entropy-M** | **69.35** | **66.54** | **63.95** | **66.61** |
> | DetectGPT | 34.33 | 33.28 | 32.72 | 33.44 |
> | **DetectGPT-M** | **82.95** | **64.13** | **79.67** | **75.58** |
> | FastGPT | 39.57 | **34.59** | 36.51 | 36.89 |
> | **FastGPT-M** | **53.71** | 30.93 | **54.63** | **46.42** |
> | DNA-GPT | 67.76 | 58.74 | 63.69 | 63.40 |
> | **DNA-GPT-M** | **71.43** | **59.59** | **67.61** | **66.21** |

---

> ### Author Response · Authors · 2025-11-22
> **Response to Reviewer kDth [3/3]**
>
> > **[W4]**  The paper lacks comparison with other enhanced metric-based approaches, such as TOCSIN [3].
> >
>
> We thank the reviewer for pointing out TOCSIN, which enhances the detector by dynamically adjusting the decision threshold. Clearly, it operates on a different dimension than our method, that is, we are **not in competition, but rather complementary**. Therefore, in addition to supplementing TOCSIN-enhanced baselines (suffix "-T"), our Markov calibration can be **further applied to the top of TOCSIN** (suffix "-TM") to further explore its detection potential, as shown in Table 7 below (Table 23 in the revised version). It can be seen that our method **further improves the performance** of TOCSIN-enhanced baselines in most cases. This significant gain suggests that TOCSIN (adjusting dynamic thresholds using token cohesiveness score) and our method (calibration token-level detection score) focus on different aspects, achieving complementarity.
>
> **Table 7. Performance (AUROC) on TOCSIN-enhanced baselines ((suffix “-T”)) and our method on top of TOCSIN (suffix “-TM”). Detectors are trained on GPT4All texts.**
>
> | Method | GPT4All | ChatGPT | Dolly | ChatGLM | Claude | ChatGPT-turbo | Avg. |
> | --- | --- | --- | --- | --- | --- | --- | --- |
> | Likelihood-T | 90.43 | 98.66 | 89.50 | 98.52 | 90.46 | 98.44 | 94.33 |
> | **Likelihood-TM** | **98.08** | **99.19** | **92.83** | **99.01** | **91.06** | **99.20** | **96.56** |
> | Log-Rank-T | 93.96 | 98.78 | 89.90 | 98.69 | 91.01 | 98.57 | 95.15 |
> | **Log-Rank-TM** | **98.07** | **99.10** | **92.32** | **99.01** | **91.46** | **99.05** | **96.50** |
> | Entropy-T | 76.90 | **98.41** | 87.20 | 98.11 | **90.91** | **98.72** | 91.71 |
> | **Entropy-TM** | **83.89** | 98.34 | **87.23** | **98.25** | 89.70 | 98.63 | **92.67** |
> | DetectGPT-T | 48.95 | 40.69 | 51.46 | 40.67 | 43.99 | 27.04 | 42.13 |
> | **DetectGPT-TM** | **95.51** | **93.27** | **79.76** | **97.67** | **70.25** | **93.59** | **88.34** |
> | FastGPT-T | 53.53 | **70.36** | 56.26 | 71.83 | **72.06** | **77.26** | **66.88** |
> | **FastGPT-TM** | **87.73** | 66.40 | **80.65** | **75.36** | 41.91 | 28.10 | 63.36 |
> | DNA-GPT-T | 98.32 | 97.97 | 96.60 | 98.34 | 95.13 | 97.93 | 97.38 |
> | **DNA-GPT-TM** | **99.56** | **99.04** | **97.70** | **99.24** | **96.30** | **99.06** | **98.48** |
>
> Thank you again for your constructive comments. If our revision has satisfactorily addressed your concerns, may we kindly request that you consider revising your final rating of our manuscript? If you have any additional concerns or comments that we may have missed in our responses, we would appreciate any further feedback to help us further enhance our work.
>
> [1] RepreGuard: Detecting LLM-Generated Text by Revealing Hidden Representation Patterns, TACL 2025.
>
> [2] Spotting LLMs With Binoculars: Zero-Shot Detection of Machine-Generated Text, ICML 2024.
>
> [3] Training-free LLM-generated Text Detection by Mining Token Probability Sequences, ICLR 2025.
>
> [4] Detecting Subtle Differences between Human and Model Languages Using Spectrum of Relative Likelihood. EMNLP 2024.

---

> > ### Comment · Reviewer_kDth · 2025-11-23
> > **Official Comment by Reviewer kDth**
> >
> > Thank you to the authors for their careful and detailed responses, which has resolved most of my concerns. I have accordingly increased my rating. Finally, I hope the authors can integrate the additional experiments into the manuscript. Good luck!

---

> > > ### Author Response · Authors · 2025-11-23
> > >
> > > We sincerely thank the reviewer for the positive feedback and for adjusting the score of our paper. We will strictly revise the final version of the paper and carefully incorporate these experiments. Thank you again for your valuable time, and wish you the best in your research!

---

### Official Review · Reviewer_FKz3 · 2025-10-31

**Soundness:** 2
**Presentation:** 2
**Contribution:** 2
**Rating:** 2
**Confidence:** 5

**Summary:**

The essential idea behind this paper is a good one. Broadly, what it does is take a token-level score based discriminator between human and machine generated text, where the final decision is based on averaging the token level scores, and replace that final averaging step with a Markov Random Field based approach. Essentially, instead of averaging the token level scores, the upgraded detector studies local patterns in the scores. This method can be bolted on top of many existing detectors.

An attempt is made to justify the effectiveness of this method in terms of mitigating randomness due to the fact that the process of generating machine text is usually stochastic. The authors provide empirical evidence for the effectiveness of their algorithm against a range of baselines.

**Strengths:**

The basic idea of the approach is sensible and works.

The analysis (Figure 2) of how the differences between the log-likelihood scores of pairs of tokens drawn from the text are correlated with the distance between the tokens is interesting.

Table 2 is very compelling in showing the uplift in performance when replacing 'averaging' scores from a detector with applying the authors Markov Random Field approach.

**Weaknesses:**

1. I do not rate the discussion of the current state of the art very highly, either in its breadth (there is much state of the art work which is missed, including many detectors which do look at local patterns in things like log-likelihood rather than just raw averages) or its accuracy. For example, there is a claim in the abstract and on page 2 that the weaknesses of token based discriminators between human and machine text stem from the randomness present in sampling algorithms for machine text. This is repeated on page 3: 'However, the detection error based on a single text may be large,because the randomness inherent in the LLM sampling mechanism may cause the MGT to deviatefrom these methods’ underlying assumptions, e.g., Log-Rank assumes that the generated tokenshave high rankings. In contrast, DetectGPT, Fast-DetectGPT, and DNA-GPT incorporate multipleperturbed (i.e., s′) or regenerated (i.e.,˜s andˆs) samples, which mitigates the errors caused byrandomness.' I think this is a fundamental mischaracterization of why DetectGPT and its successors were effective, they do not mitigate the randomness present in language model decoding strategies. Instead, they mitigate a fundamentally different issue, that the expected statistical properties of different types of text are different -  that poetry for example should have a higher inherent uncertainty about the next token than python code.
2. In the discussion of score calculation (p3) there is a claim that aggregating a score (such as log-probability) over a sequence is problematic because there is randomness in the process of text generating. But surely averaging over a number of tokens is a reasonable way to deal with this randomness? There is an implicit claim being made that, if instead of aggregating log likelihood scores, one were to process the log likelihood scores to the authors algorithm, the effects of randomness in the generation process of machine text would be mitigated. No evidence is presented for this claim and I suspect it is wrong. (I'm not saying the author's text-detection algorithm is ineffective, I'm saying I don't believe the justification for its effectiveness).
3. Theorem 1 gives a theoretical proof of correlations between attention scores. The authors write 'The following
theorem will reveal the relationship between attention scores, which in turn help us understand the
relationship between detection scores of context tokens.' I am unconvinced that Theorem 1 is useful in understanding anything about correlations between log-likelihoods of tokens generated in a sequence, for example. I don't think that Theorem 1 contributes understanding to the current approach.
3. There is already work in the literature which seeks to exploit patterns in statistics such as log-likelihood, as opposed to just raw averages. For example, [Detecting Subtle Differences between Human and Model Languages Using Spectrum of Relative Likelihood](https://aclanthology.org/2024.emnlp-main.564/) (Xu et al., EMNLP 2024)], Xu, Yihuai, et al. "Training-free LLM-generated Text Detection by Mining Token Probability Sequences." The Thirteenth International Conference on Learning Representations. . These are just two examples of which I'm aware, there are probably many more. I am not confident that the authors have done a good literature review and I don't think they have compared their approach to a broad enough set of baselines or the current state of the art (I think all of the comparators are at least eighteen months old, in a fast moving field). What the authors do is upgrade raw average of quantities like log-likelihood to instead look at local patterns in log likelihood, so it is very poor that other papers which have the same overall aim are not included in the baselines.
4. The Markov random field approach is complicated relative to a simpler Markov chain approach. I would have liked to see the two methods compared.
5. I strongly disagree with much of the commentary assessing the current state of the field. For example, the authors write 'Our analysis reveals that they share a threshold-based detection criterion, with only minor differences, such as the inclusion of auxiliary data (e.g., perturbed texts).' I find this claim odd, the idea of DetectGPT which moved us from asking 'is the log-likelihood high' to 'is the log-likelihood higher than other texts conveying the same meaning' was revolutionary both conceptually and in terms of performance. The authors' method also uses a threshold-based detection criterion.

**Questions:**

Figure 2 is supposed to demonstrate that there is a substantial correlation between the log-likelihood score of a token and the log-likelihood score of the token's nearest neighbour. Certainly the graph shows some effect, but I find it hard to assess the strength of the correlation. Could you give us a clearer visualisation? For example, it would be useful to also include tokens at distance 100, so one could see a stark difference between the distance between the log-likelihood scores of nearest neighbours and distant neighbours.

On the first line of page 2 you ascribe log-rank to Mitchell et al, which I think is wrong. Probably it should be the same reference that you have for log-likelihood and entropy, although I'm not certain.

I don't think Fast-DetectGPT has a significant compute overhead as stated on page 3, they collect all of the required info from a single forward pass through the text.

I don't understand the point you are making in the following paragraph. Isn't the same true of your method, that having computed some number for each text you then use a threshold to divide into likely human and likely machine based? 'Detection. These methods all employ threshold-based detection mechanisms, whose effectiveness relies heavily on the accuracy of their calculated scores. As previously discussed, two factors compromise this accuracy: (1) the inherent randomness of LLM-generated text introduces bias
into score calculation, and (2) direct score aggregation fails to mitigate this bias. As a result, their detection performance is often unsatisfactory.

---

> ### Author Response · Authors · 2025-11-22
> **Response to Reviewer FKz3 [1/6]**
>
> Thank you for your timely and valuable feedback, which we really cherish. Please see the below responses to your comments. Any further comments and discussions are welcome!
>
> > **[W1]**  I do not rate the discussion of the current state of the art very highly, either in its breadth (there is much state of the art work which is missed, including many detectors which do look at local patterns in things like log-likelihood rather than just raw averages) or its accuracy. For example, there is a claim in the abstract and on page 2 that the weaknesses of token based discriminators between human and machine text stem from the randomness present in sampling algorithms for machine text. This is repeated on page 3: 'However, the detection error based on a single text may be large,because the randomness inherent in the LLM sampling mechanism may cause the MGT to deviatefrom these methods’ underlying assumptions, e.g., Log-Rank assumes that the generated tokens have high rankings. In contrast, DetectGPT, Fast-DetectGPT, and DNA-GPT incorporate multipleperturbed (i.e., s′) or regenerated (i.e.,˜s andˆs) samples, which mitigates the errors caused byrandomness.' I think this is a fundamental mischaracterization of why DetectGPT and its successors were effective, they do not mitigate the randomness present in language model decoding strategies. Instead, they mitigate a fundamentally different issue, that the expected statistical properties of different types of text are different -  that poetry for example should have a higher inherent uncertainty about the next token than python code.
> >
>
> Thank you for the reviewer’s valuable question, which has helped us refine our positioning and literature review.
>
> **On the Mechanism of DetectGPT (and related methods).** We fully agree with your fundamental point on the core contributions of DetectGPT. We apologize if our initial paper inadvertently obscured this by focusing too heavily on the data perspective. We want to clarify that discussing existing methods from a unified perspective is **not to trivialyze their core mechanisms** (in fact, our calibration strategy rely on them), but instead emphasizes **a common potential risk beyond their core mechanisms**: regardless of whether the core metric is simple likelihood or complex curvature (e.g., DetectGPT), all these methods ultimately aggregate token-level scores into a final scalar for thresholding, which is crucial for the accuracy of token-level scores. The paper does not focus too much on their core metrics because our contribution is not a new detector that competes on metric design, but a calibration layer that improves the precision of the scores before being fed into that aggregation step. Therefore, we focus more on **their potential commonalities than their diverse core designs**.
>
> To avoid confusion, in Section 2 (Line 126 in the revised version), we explicitly state that "Note that this paper does not discuss their diverse core metric designs. This is because we aim to design a general enhancement framework decoupled from specific detectors, requiring us to start from possible commonalities rather than diverse designs”.
> **On Missing Related Work.** We apologize for the omission of recent works that utilize local patterns. Following your suggestion, we have expanded Appendix B (Related Work) to include methods like FourierGPT [1] and Lastde [2]. In addition, we have added a new section in Appendix A.2 (Differences from existing methods) to clarify our unique contribution:
>
> 1. Methodology. Unlike FourierGPT or Lastde, which propose new standalone metrics, our method is a universal plug-in. We do not replace the metric; we calibrate the intermediate token-level scores of any existing metric (Likelihood, DetectGPT, etc.) using a Markov Random Field.
> 2. Theory vs. Empiricism. While many existing local-pattern methods are empirically motivated (e.g., observing spectral power differences and token probability fluctuation), our approach is grounded in the theoretical derivation of attention dynamics (Theorem 1), which formally reveals the Neighbor Similarity and Initial Instability properties we model.

---

> ### Author Response · Authors · 2025-11-22
> **Response to Reviewer FKz3 [2/6]**
>
> > **[W2]**  In the discussion of score calculation (p3) there is a claim that aggregating a score (such as log-probability) over a sequence is problematic because there is randomness in the process of text generating. But surely averaging over a number of tokens is a reasonable way to deal with this randomness? There is an implicit claim being made that, if instead of aggregating log likelihood scores, one were to process the log likelihood scores to the authors algorithm, the effects of randomness in the generation process of machine text would be mitigated. No evidence is presented for this claim and I suspect it is wrong. (I'm not saying the author's text-detection algorithm is ineffective, I'm saying I don't believe the justification for its effectiveness).
> >
>
> You are absolutely correct that averaging is a standard and reasonable way to reduce variance. We do not claim that averaging is ineffective; instead, our method relies on it. This is because our method is simply an enhancement strategy for score calibration and does not modify any components of the original detector. We argue that it is **insufficient** because it treats all token scores as independent, ignoring the structured dependencies (Neighbor Similarity) inherent in the generation process. To this end, we pull scores back in line with their neighbors by enforcing Markov consistency. The calibrated scores are then aggregated using the original method (e.g., averaging). Therefore, **we are complementary, not competitive**.
>
> In addition, we empirically validated the conclusion regarding reduced randomness. Figure 1 and Appendix E.1 in the paper show that, after calibration, the token-level score distribution becomes tighter and thus more readily distinguishable from human-generated text.
>
> To clarify and avoid confusion, we have revised our statement on score aggregation (Line 138 in the revised version), as it does not affect our main focus: “These methods appear to aggregate token-level detection scores, for example, by naively averaging. Although such aggregation mitigates the variance caused by randomness to some extent, it cannot fully rectify the inherent imprecision of individual token-level scores arising from LLM sampling. Therefore, it is possible to further explore the potential of these methods by calibrating token-level scores”.
>
> > **[W3]**  Theorem 1 gives a theoretical proof of correlations between attention scores. The authors write 'The following theorem will reveal the relationship between attention scores, which in turn help us understand the relationship between detection scores of context tokens.' I am unconvinced that Theorem 1 is useful in understanding anything about correlations between log-likelihoods of tokens generated in a sequence, for example. I don't think that Theorem 1 contributes understanding to the current approach.
> >
>
> Thank you for the question. Our clarification will start from the relationship between attention and token-level scores, and then reveal how Theorem 1 leads to the findings of Neighbor Similarity and Initial Instability.
>
> - **From Attention to Token-level Scores** (e.g., log-likelihoods)**.** In Transformer architectures, the relationship between attention and token-level scores is functional: The attention weights $a_t$ determine the next-token embedding $x_{t+1}$. The embedding determines the probability $p(x_{t+1})$. Since the function mapping $a_t \to \log p(x_{t+1})$ is continuous, constraints on the dynamics of $a_t$ naturally propagate to the log-likelihoods.
> - **Neighbor Similarity.** Theorem 1 proves that $a_{t+1}$ is bounded by $a_t$. This creates a recursive dependency (similar to simulated annealing) in which high attention scores lead to high scores at the next, and vice versa. That is, the attention distribution cannot shift arbitrarily fast between steps. Since the attention distribution is stable between adjacent steps, the resulting output probabilities (and thus detection scores) exhibit high similarity between neighbors.
> - **Initial Instability.** Theorem 1 reveals that both the main term coefficient $C$ and the noise $η$ depend on $t$ (the current step), which induces initial instability via a dual mechanism: (1) For the noise term 𝜂﻿, when 𝑡﻿ is small (early step), $\eta$ is large. This leads to looser bounds, allowing a wide range of values for $a_{t+1}$ and thus enabling sharp fluctuations. As $t$ increases (later step), the bounds become tighter, enforcing stability. (2) For the main term coefficient $C$, when $t$ is small, $C$ is large. Therefore, even slight differences in $\alpha_t$ may be exponentially magnified. As $t$ grows, the distribution naturally stabilizes.
>
> In addition to the qualitative theoretical basis, we have empirically verified that these predicted correlations hold quantitatively in Figures 2-3 and 13-18 in the initial submission (now Figures 13-20). We have included these discussions in the paragraphs following Theorem 1 to make them clearer.

---

> ### Author Response · Authors · 2025-11-22
> **Response to Reviewer FKz3 [3/6]**
>
> > **[W4]**  There is already work in the literature which seeks to exploit patterns in statistics such as log-likelihood, as opposed to just raw averages. For example, Detecting Subtle Differences between Human and Model Languages Using Spectrum of Relative Likelihood (Xu et al., EMNLP 2024)], Xu, Yihuai, et al. "Training-free LLM-generated Text Detection by Mining Token Probability Sequences." The Thirteenth International Conference on Learning Representations. . These are just two examples of which I'm aware, there are probably many more. I am not confident that the authors have done a good literature review and I don't think they have compared their approach to a broad enough set of baselines or the current state of the art (I think all of the comparators are at least eighteen months old, in a fast moving field). What the authors do is upgrade raw average of quantities like log-likelihood to instead look at local patterns in log likelihood, so it is very poor that other papers which have the same overall aim are not included in the baselines.
> >
>
> We sincerely thank the reviewer for highlighting these relevant works. We apologize for their omission and have added a detailed discussion of them to **Appendix B (Related Work)**.
>
> As we stated in our response to your [W3], our method differs fundamentally from these approaches: they focus on **feature extraction** and design new metrics (Diversity Entropy and Spectral Features) to capture local patterns in probability sequences. Instead, our method focuses on **score calibration**, which is independent of existing detection methods, providing a universal Markov-Informed layer that can be applied on top of these methods.
>
> Therefore, our method is **complementary, not competitive**, to these approaches. Accordingly, in the revised version, we applied our Markov-informed calibration (denoted by the '-M' suffix) to Lastde and FourierGPT, as shown in **Tables 1-3 below** (full results can be found in Tables 18-20 in the revised paper). First, the consistent improvement proves our method is complementary, offering a universal boost even to SOTA detectors. Second, the gains on these local-pattern methods are slightly smaller than on raw Likelihood (Table 2 in the paper). This validates our theory: because these methods already capture some local dependencies, there is less "noise" for our Markov layer to calibrate.
>
> In addition, we have added more state-of-the-art methods, RepreGuard [4] and Binoculars [5]. As shown in **Tables 4-5 below**  (full results can be found in Tables 18-20 in the revised paper), our method serves as a universal plug-in that enhances these detectors. For example, on the Reuters dataset, our method improves RepreGuard by 8.97% and Binoculars by 4.42%. The Fast-DetectGPT and DNAGPT in the initial submission represent the current state of technology. We believe that further adding these more recent methods will more comprehensively demonstrate the strong generalizability of our approach. Thank you again for the reviewers' suggestions.
>
> **Table 1. Performance concerning AUROC on Essay. Detectors are trained on GPT4All texts.**
>
> | Detector | GPT4All | ChatGPT | Dolly | ChatGLM | Claude | ChatGPT-turbo | Avg. |
> | --- | --- | --- | --- | --- | --- | --- | --- |
> | Lastde | 99.54 | 99.72 | 97.38 | 99.98 | 75.98 | **91.16** | 93.96 |
> | **Lastde-M** | **99.68** | **99.77** | **97.87** | **99.99** | **76.21** | 90.96 | **94.08** |
> | FourierGPT | 91.70 | 90.18 | 93.74 | 98.26 | **44.90** | **42.11** | 76.81 |
> | FourierGPT-M | **97.10** | **95.27** | **97.64** | **99.44** | 41.14 | 41.08 | **78.61** |
>
> **Table 2. Performance concerning AUROC on Reuters. Detectors are trained on GPT4All texts.**
>
> | Detector | GPT4All | ChatGPT | Dolly | ChatGLM | Claude | ChatGPT-turbo | Avg. |
> | --- | --- | --- | --- | --- | --- | --- | --- |
> | Lastde | 99.21 | 99.67 | 94.25 | **99.83** | 83.27 | **99.09** | 95.89 |
> | **Lastde-M** | **99.55** | **99.68** | **96.09** | 99.82 | **85.64** | 99.00 | **96.63** |
> | FourierGPT | 99.28 | 96.71 | 99.06 | 98.45 | **56.82** | 49.84 | 83.36 |
> | **FourierGPT-M** | **99.36** | **98.29** | **99.29** | **98.48** | 55.13 | **49.88** | **83.40** |
>
> **Table 3. Performance concerning AUROC on DetectRL. Detectors are trained on Llama-2-70b texts.**
>
> | Detector | Llama-2-70b | ChatGPT | Google-PaLM | Avg. |
> | --- | --- | --- | --- | --- |
> | Lastde | 78.36 | 51.93 | 69.04 | 66.45 |
> | **Lastde-M** | **78.70** | **52.55** | **71.06** | **67.44** |
> | FourierGPT | 58.11 | 50.57 | 59.99 | 56.23 |
> | **FourierGPT-M** | **64.67** | **52.64** | **64.07** | **60.46** |

---

> ### Author Response · Authors · 2025-11-22
> **Response to Reviewer FKz3 [4/6]**
>
> **Table 4. Detection enhancement on the RepreGuard detector. The detectors are trained on GPT4All texts.**
>
> | Dataset | Detector | GPT4All | ChatGPT | Dolly | ChatGLM | Claude | ChatGPT-turbo | Avg. |
> | --- | --- | --- | --- | --- | --- | --- | --- | --- |
> | Essay | RepreGuard | 94.66 | 97.35 | 90.40 | 91.95 | **47.51** | **43.42** | 77.55 |
> | Essay | **RepreGuard-M** | **95.46** | **99.55** | **90.55** | **99.95** | 47.18 | 43.12 | **79.30** |
> | Reuters | RepreGuard | 97.21 | **98.32** | 8.24 | **97.26** | 21.81 | 44.32 | 61.19 |
> | Reuters | **RepreGuard-M** | **98.33** | 97.87 | **48.12** | 97.00 | **29.08** | **50.59** | **70.16** |
>
> **Table 5. Detection enhancement on the Binoculars detector.  The detectors are trained on GPT4All texts.**
>
> | Dataset | Detector | GPT4All | ChatGPT | Dolly | ChatGLM | Claude | ChatGPT-turbo | Avg. |
> | --- | --- | --- | --- | --- | --- | --- | --- | --- |
> | Essay | Binoculars | 96.33 | 98.59 | 88.57 | **99.57** | **87.21** | **98.84** | 94.85 |
> | Essay | **Binoculars-M** | **97.34** | **98.65** | **90.19** | 97.68 | 86.80 | **98.84** | **94.91** |
> | Reuters | Binoculars | 80.96 | 97.99 | 61.25 | 99.80 | 83.23 | 98.70 | 86.99 |
> | Reuters | **Binoculars-M** | **90.90** | **98.49** | **74.10** | **99.86** | **86.37** | **98.72** | **91.41** |
>
> > **[W5]**  The Markov random field approach is complicated relative to a simpler Markov chain approach. I would have liked to see the two methods compared.
> >
>
> We thank the reviewer for this insightful suggestion. Comparing our MRF approach to a simpler Markov Chain (e.g., HMM) is crucial to justifying our design choice.
>
> First, we chose the MRF over Markov chain for two key reasons:
>
> - **Theoretical rationality.** A Markov chain typically models **unidirectional dependencies** (current state depends on previous state). However, in MGT detection, we identified the Neighbor Similarity property, which is inherently **bidirectional**. An MRF naturally captures these bidirectional dependencies, whereas a standard Markov Chain struggles to utilize future context during inference without complex multi-pass algorithms.
> - **Computational Efficiency.** A raw MRF may be complicated. However, as described in Section 4.2, our **Mean-Field Approximation** transforms the problem into lightweight iterative matrix multiplications. This allows for **massive parallelization on GPUs**. In contrast, exact inference in HMMs (e.g., Forward-Backward algorithm) is inherently sequential, which can be slower on modern hardware despite having similar theoretical complexity $\mathcal{O}(N)$.
>
> Second, following your suggestion, we implemented a Hidden Markov Model (HMM) based calibration strategy and applied it to existing detectors (”HMM” suffix).
>
> - **Performance**. As shown in Table 6 below (the left part of Table 22 in the revised version), while the HMM improves on the original baselines, it consistently **underperforms our MRF approach**. This validates the need to model bidirectional dependencies.
> - **Running Time**. As shown in Table 7 below (right part of Table 22 in the revised version), our MRF implementation is **actually faster** in practice due to GPU-friendly matrix operations.
>
> **Table 6. Performance comparison with the HMM-based method on the Essay.**
>
> | Method | GPT4All | ChatGPT | Dolly | ChatGLM | Claude | ChatGPT-turbo | Avg. |
> | --- | --- | --- | --- | --- | --- | --- | --- |
> | Likelihood | 96.16 | 98.79 | 90.90 | 99.29 | 92.76 | 99.13 | 96.17 |
> | **Likelihood-M** | **98.58** | **99.47** | **94.59** | **99.54** | **94.82** | **99.72** | **97.79** |
> | Likelihood-HMM | 96.39 | 98.87 | 91.24 | 99.33 | 92.91 | 99.17 | 96.32 |
> | Log-Rank | 96.55 | 98.95 | 90.08 | 99.36 | 92.01 | 99.24 | 96.03 |
> | **Log-Rank-M** | **98.57** | **99.41** | **93.82** | **99.55** | **92.91** | **99.64** | **97.32** |
> | Log-Rank-HMM | 96.68 | 98.99 | 90.24 | 99.38 | 92.08 | 99.27 | 96.10 |
> | Entropy | 74.19 | 89.49 | 73.26 | 84.11 | 86.58 | 95.94 | 83.93 |
> | **Entropy-M** | **83.52** | **93.28** | **81.11** | **91.44** | **87.96** | **96.75** | **89.01** |
> | Entropy-HMM | 74.47 | 89.70 | 73.56 | 84.39 | 86.61 | 95.98 | 84.12 |
>
> **Table 7. Comparison of runtime with HMM-based method.**
>
> | Method | Training Time | Inference Time |
> | --- | --- | --- |
> | Likelihood | 13.58 | 61.21 |
> | Likelihood-M | 15.01 | 72.67 |
> | Likelihood-HMM | 20.64 | 90.63 |
> | Log-Rank | 15.98 | 75.60 |
> | Log-Rank-M | 17.40 | 87.55 |
> | Log-Rank-HMM | 21.77 | 103.70 |
> | Entropy | 13.26 | 64.49 |
> | Entropy-M | 14.69 | 73.66 |
> | Entropy-HMM | 16.87 | 88.26 |

---

> ### Author Response · Authors · 2025-11-22
> **Response to Reviewer FKz3 [5/6]**
>
> > **[W6]**  I strongly disagree with much of the commentary assessing the current state of the field. For example, the authors write 'Our analysis reveals that they share a threshold-based detection criterion, with only minor differences, such as the inclusion of auxiliary data (e.g., perturbed texts).' I find this claim odd, the idea of DetectGPT which moved us from asking 'is the log-likelihood high' to 'is the log-likelihood higher than other texts conveying the same meaning' was revolutionary both conceptually and in terms of performance. The authors' method also uses a threshold-based detection criterion.
> >
>
> Thank you for your question. We agree with your assessment: DetectGPT was indeed revolutionary. We apologize if our statement that there were "only minor differences" inadvertently downplayed this contribution. That was not our intent.
>
> In fact, understanding existing methods from a unified perspective **does not impley downplaying their internal metrics** (in fact, our calibration strategy rely on them), but rather emphasizes a **common potential risk**: regardless of whether the core metric is simple likelihood or complex curvature (DetectGPT), all these methods ultimately aggregate token-level scores into a final scalar for thresholding, which is crucial for the accuracy of token-level scores. We don't focus too much on their core metrics here because our contribution is not a new detector that competes on metric design, but a calibration layer that improves the precision of the scores before being fed into that aggregation step. Therefore, we focus more on their commonalities than their diverse core designs. We have removed the ambiguous statements.
>
> Besides, you correctly noted that our method also uses a threshold. This is exactly our point: because we all use thresholds, the **precision of the token-level score is paramount**. To reiterate, our contribution is not a new detector, but a calibration layer to make those scores precise enough for the threshold to work reliably.
>
> > **[Q1]**  Figure 2 is supposed to demonstrate that there is a substantial correlation between the log-likelihood score of a token and the log-likelihood score of the token's nearest neighbour. Certainly the graph shows some effect, but I find it hard to assess the strength of the correlation. Could you give us a clearer visualisation? For example, it would be useful to also include tokens at distance 100, so one could see a stark difference between the distance between the log-likelihood scores of nearest neighbours and distant neighbours.
> >
>
> Thank you for the valuable suggestion. Following your suggestion, we have revised Figure 2 (and Appendix Figures 13-15) to **extend the hop distance axis up to 100**. The new results show that the token-level detection score distance (absolute value) exhibits a **clear, monotonically increasing trend** with increasing hop distance. For example, for the DetectGPT Score on Google-PaLM text, the distance starts at approximately 1.4 at $k=1$ and rises sharply to over 2.2 at $k=100$. This represents an approximately 57% increase in score divergence, confirming the significant difference you mentioned. This quantitative gap validates our finding: token-level detection scores exhibit strong **local dependencies (Neighbor Similarity)** that decay significantly over distance.
>
> > **[Q2]**  On the first line of page 2 you ascribe log-rank to Mitchell et al, which I think is wrong. Probably it should be the same reference that you have for log-likelihood and entropy, although I'm not certain.
> >
>
> We thank the reviewer for their careful reading and for pointing out this citation inaccuracy.
> The use of token ranking information for detection was pioneered earlier by Gehrmann et al. (2019) in their work on GLTR [1]. In the revised manuscript, we have corrected the citation for Log-Rank (e.g., page 2, Section 5, and Appendix D.2).

---

> ### Author Response · Authors · 2025-11-22
> **Response to Reviewer FKz3 [6/6]**
>
> > **[Q3]**  I don't think Fast-DetectGPT has a significant compute overhead as stated on page 3, they collect all of the required info from a single forward pass through the text.
> >
>
> When compared to other multiple sampling methods (e.g., DetectGPT and DNAGPT), you are right that Fast-DetectGPT is a highly efficient method. Our statement regarding "significant increase in compute overhead" was intended as a relative comparison against simple single-text methods (like Log-Likelihood or Entropy), rather than an absolute critique of its efficiency.
>
> - **Single-Text Methods:** Methods like Log-Likelihood require only a **single forward pass of the single text** $s$.
> - **Fast-DetectGPT:** While significantly faster than the original DetectGPT, Fast-DetectGPT requires a **single forward pass across multiple texts**, which is inherently more computationally expensive than a forward pass for a single input.
>
> We also empirically validate this computational difference in Table 16 in the initial submission (now Table 24 in the revised version). While Fast-DetectGPT ($\approx 279$s) is indeed much faster than DetectGPT ($\approx 924$s), it still incurs a non-negligible computational overhead compared to the Likelihood method.
> To avoid confusion, we have revised the text on Page 3 to explicitly state that the overhead is "relative to single-text metric-based methods", thereby accurately expressing Fast-DetectGPT's efficiency.
>
> > **[Q4]**  I don't understand the point you are making in the following paragraph. Isn't the same true of your method, that having computed some number for each text you then use a threshold to divide into likely human and likely machine based? 'Detection. These methods all employ threshold-based detection mechanisms, whose effectiveness relies heavily on the accuracy of their calculated scores. As previously discussed, two factors compromise this accuracy: (1) the inherent randomness of LLM-generated text introduces bias into score calculation, and (2) direct score aggregation fails to mitigate this bias. As a result, their detection performance is often unsatisfactory.
> >
>
> We thank the reviewer for the question.
>
> **Clarification.** You are correct: our method also uses a threshold-based decision mechanism. We do not claim that thresholds are inherently flawed; in fact, we retain the original detector's decision logic exactly as is.
>
> **The Real Issue: Token-level Score Quality.** The point we intended to make in that paragraph is about token-level score quality, not the decision mechanism itself. Specifically, existing methods typically rely on token-level scores, while the inherent randomness of LLM sampling can introduce fluctuations in individual token scores. Feeding noisy scores into a threshold-based decision may lead to poor performance. As an enhancement component, our calibration strategy do not replace the threshold. Instead, we insert a calibration layer beforehand. We use Markov modeling to calibrate the token-level scores before they are aggregated.
>
> We acknowledge that our original wording may have implied that the threshold mechanism itself was at fault. In the revised version, we clarify that the limitation lies in “using uncalibrated, high-noise scores to compute the final detection score,” rather than in the detection mechanism itself.
>
> Thank you again for your constructive comments. We observed that the reviewer's main concerns stemmed from our discussion of existing methods. To reiterate, understanding existing methods from a unified perspective does not imply downplaying their internal metrics (in fact, our calibration strategy relies on them), but rather emphasizes a common potential risk: regardless of whether the core metric is simple likelihood or complex curvature (DetectGPT), all these methods ultimately aggregate token-level scores into a final scalar for thresholding, which is crucial for the accuracy of token-level scores. If you have any additional concerns or comments that we may have missed in our responses, we would appreciate any further feedback to help us further enhance our work.
>
> [1] Detecting Subtle Differences between Human and Model Languages Using Spectrum of Relative Likelihood. EMNLP 2024.
>
> [2] Training-free LLM-generated Text Detection by Mining Token Probability Sequences. ICLR 2025.
>
> [3] GLTR: Statistical Detection and Visualization of Generated Text. ACL 2019.
>
> [4] RepreGuard: Detecting LLM-Generated Text by Revealing Hidden Representation Patterns, TACL 2025.
>
> [5] Spotting LLMs With Binoculars: Zero-Shot Detection of Machine-Generated Text, ICML 2024.

---

> > ### Comment · Reviewer_FKz3 · 2025-11-24
> > **Thanks for your responses**
> >
> > Many thanks for your detailed responses to my questions. I think that I now have a better understanding of your work, and don't have any further questions. I am very open minded going into the reviewer discussion phase and look forward to a good discussion with the other reviewers.

---

> > > ### Author Response · Authors · 2025-11-24
> > > **Thank you for your continued engagement**
> > >
> > > Dear Reviewer #FKz3,
> > >
> > > Thank you very much for your continued engagement and patience when reading our lengthy responses. We are glad to hear that our clarifications have provided a better understanding of our work and that your immediate concerns have been resolved. We also appreciate your open-minded attitude heading into the discussion phase.  Since there are no outstanding questions, we kindly hope that you might consider adjusting your score to reflect the resolution of the initial issues and the quality of our revised version (which we have submitted for your review). If you have any further questions, we would be happy to provide further details.
> > >
> > > Best regards,
> > >
> > > Authors of 16501

---

### Official Review · Reviewer_LQNk · 2025-11-05

**Soundness:** 2
**Presentation:** 2
**Contribution:** 2
**Rating:** 4
**Confidence:** 4

**Summary:**

This work observes the impacts of randomness (during text generation) on the metric-based LLM-generated text detectors, a specific type of detection that relies on statistical scores computed from token-wise probabilities. To alleviate this challenge, a Markov-informed score calibration method was proposed and a plug-and-play module was designed to improve the detection performance of existing metric-based detectors. Experiments were conducted to verify some claims mentioned in this work.

**Strengths:**

1. This work identified and analyzed randomness issues during token generation which might influence the detection performance of LLM-generated text detectors, which makes sense and the perspective appears new.

2. A Markov-informed score strategy was formulated and served as a plug-and-play for existing fake text detectors.

3. Experiments on certain datasets were done to validate some claims proposed in this work.

**Weaknesses:**

1. Unclear presentation and lack of clarity. The definition and introduction of neighbor similarity is not clear both in the writing part and the illustration part. Particulary in Fig.2 and Fig.3, much important information is missing. E.g., how to compute the detection scores, what kind of distances being used, why there are two variables in the horizontal axis (e.g. position, and log-rank score, which variable do the numerical values 0.0 to 0.8 refer to?)

2. Though some theorems were provided, the assumptions are quite strong to be useful in practical cases, e.g. Theorem 1 was based on a single-layer transformer model, which makes it difficult to generalize to LLMs, it is suggested to elaborate on the scope and guidance of these theoretical studies.

3. Unclear descriptions of the experimental settings. For the experimental comparisons, it is suggested to clearly present the threat models, being a white-box detection or black-box one, if black-box, what is the proxy model being used etc, the settings should clear to be reproducible.

4. From Table 2, the advantages of the proposed module does not seem significant, particulary by integrating with FastGPT and DNA-GPT, please give some explainations. Besides, More benchmark datasets and baselines should be included.  For benchmark dataset, it is suggested to include experiments on datasets e.g., PubMedQA, in the black-box setting, and experiments on short-text detection are encouraged. Also, more recent metric-based baselines should be discussed and compared, e.g. ref 1 and ref 2, etc.

ref1: Xu et al, Training-free LLM-generated Text Detection by Mining Token Probability Sequences, ICLR 2025.

ref2: Wu et al, MoSEs: Uncertainty-Aware AI-Generated Text Detection via Mixture of Stylistics Experts with Conditional Thresholds, EMNLP 2025.

**Questions:**

Please see Weakness part, eg. the vague description and definition of neighbor similarity, experimental settings etc.
In addition to the weakness part, another question is, The presentation mostly relies on figures, why not provide precisely the numerical results? (It often seems hard to tell the exact number and comparision results given the figures and bars etc).

**Details Of Ethics Concerns:**

NA.

---

> ### Author Response · Authors · 2025-11-22
> **Response to Reviewer LQNk [1/4]**
>
> Thank you for your timely and valuable feedback, which we really cherish. Please see the below responses to your comments. Any further comments and discussions are welcome!
>
> > **[W1]**  Unclear presentation and lack of clarity. The definition and introduction of neighbor similarity is not clear both in the writing part and the illustration part. Particulary in Fig.2 and Fig.3, much important information is missing. E.g., how to compute the detection scores, what kind of distances being used, why there are two variables in the horizontal axis (e.g. position, and log-rank score, which variable do the numerical values 0.0 to 0.8 refer to?)
> >
>
> We thank the reviewer for highlighting the lack of clarity in our visual presentation. We have revised Section 3 and the captions of Figures 2 and 3 to clarify this.
>
> **Clarification on Neighbor Similarity. We have refined the definition in Section 3 (Line 210 in the revised version) to state: “**(1) Neighbor Similarity, where the detection scores of adjacent tokens in a sequence exhibit statistically lower variance compared to tokens that are far apart”. This stems from our analysis of the Transformer’s attention mechanism (Theorem 1), which inspired the detection scores remain stable over short intervals.
>
> **Clarification on Fig. 2 and Fig. 3.** We provide the missing technical details below:
>
> - **Detection Scores (What is computed).**  These are the **token-level metrics output** by the baseline detectors. For example, for the Likelihood method, the detection score of $t$-th token  $score(s_t)=- \log P(x_t | x_{<t})$. The formulas for all metrics are detailed in Table 1 below, and Table 1 in the revised version.
> - **Distance Metric (Y-axis):** The "Distance" represents the **Mean Absolute Difference** between token scores. Specifically, in Fig. 2 (Neighbor Similarity), we calculate $\frac{1}{|S|}\frac{1}{N-K}\sum_{s\in S}\sum_{t=0}^{N-K}|score(s_t) - score(s_{t+k})|$ to measure stability at $k$ hop, where $S$ is the text set. In Fig. 3 (Initial Instability), we calculate neighbor different$\frac{1}{S}\sum_{s\in S}|score(s_t) - score(s_{t+1})|$ to measure  position stability at specific text positions $t$.
> - **X-axis Variables:** The numerical values 0.0 to 0.8 refer to the **normalized relative position of the token in the text** (where 0.0 is the start). For example, $0.1$ represents the 10% portion of the generated text. Therefore, assuming the text length is $|s|$, then the distance corresponding to 0.1 is $|score(s_{0.1\cdot|s|}) - score(s_{0.1\cdot|s|+1})|$. Besides, the text "Log-Rank Score" (and "Log-Likelihood Score") appearing below the X-axis is a **sub-caption** indicating the detector used, NOT an axis variable. To avoid confusion, we add a number to the sub-caption title and explicitly state that this is a detector, for example, "(a) Log-Rank detector Score".
>
> **Table 1. Token-level detection score for existing metric-based methods.**
>
> | Method | token-level detection score $score(s_t)$ |
> | --- | --- |
> | Log-likelihood | $\log p(s_t\|s_{<t})$ |
> | Log-Rank | $Rank(p(s_t\|s_{<t}))$ |
> | Entropy | $\sum_{v\in V} p(v\|s_{<t})\log p(v\|s_{<t})$ |
> | DetectGPT | $p(s_t\|s_{<t})$ |
> | Fast-DetectGPT | $p(s_t\|s_{<t})$ |
> | DNA-GPT | $p(s_t\|s_{<t})$ |
>
> > **[W2]**  Though some theorems were provided, the assumptions are quite strong to be useful in practical cases, e.g. Theorem 1 was based on a single-layer transformer model, which makes it difficult to generalize to LLMs, it is suggested to elaborate on the scope and guidance of these theoretical studies.
> >
>
> We appreciate the reviewer’s valuable comment. We acknowledge that the single-layer assumption is a simplification. However, theoretically analyzing the full dynamics of multi-layer, multi-head transformers with non-linear activations remains a major open challenge. Therefore, we followed established theoretical works [1] by adopting a tractable single-layer model to mathematically derive the attention bounds.
>
> **Scope and Guidance of the Theory.** The primary scope of Theorem 1 is not to act as a precise quantitative predictor for LLMs, but to provide **theoretical intuition** into how the self-attention mechanism naturally induces dependencies between tokens. This theoretical insight allowed us to hypothesize two key properties that drive our method: **Neighbor Similarity and Initial Instability.**
>
> **Bridging Theory and Practice.** While the theorem uses a simplified model, we extensively verified that these two properties indeed **hold for complex, modern multi-layer LLMs** (e.g., ChatGPT, Llama-2, Claude), as shown in  Figures 2-3 and 13-18.
> We have revised the paragraph at the end of Section 3 to explicitly state: "While Theorem 1 is derived from a simplified single-layer model to ensure tractability, it captures fundamental attention dynamics. The empirical results demonstrate that these properties also hold for complex, multi-layer LLMs (e.g., ChatGPT, Llama-2, Claude)“.

---

> ### Author Response · Authors · 2025-11-22
> **Response to Reviewer LQNk [2/4]**
>
> > **[W3]**  Unclear descriptions of the experimental settings. For the experimental comparisons, it is suggested to clearly present the threat models, being a white-box detection or black-box one, if black-box, what is the proxy model being used etc, the settings should clear to be reproducible.
> >
>
> We agree that the specific threat model and proxy settings need to be more explicit. To address this, we have added a new subsection **"Threat Model and Proxy Settings" to Appendix D** in the revised manuscript. Here is a summary of the clarifications:
>
> - **Threat Model.** We operate under a strictly **Black-Box** setting regarding the target LLM. We assume that the detector is entirely unknown to the LLM that generates the text (e.g., ChatGPT, Claude). Therefore, we employ a local Proxy Model to compute token-level metrics (e.g., Log-Likelihood and Rank) for the candidate text, which are reasonable due to the transferability of the extracted metrics.
> - **Proxy Models.** For all baselines, we use **GPT2 as the proxy model.** Additionally, we use GPT2-XL as the scoring model for Fast-DetectGPT. To learn the 2x2 parameter $W_{mrf}$ introduced by our strategy, we perform training based on various LLM texts, as explicitly stated in the "Training Text" of the table. For example, in Table 2 of the paper, we learn $W_{mrf}$ on GPT4All texts of the Essay dataset and Llama-2-70b texts of the DetectRL dataset, and then test all candidate texts (from ChatGPT, Claude, or Dolly, etc.) using the learned detector.
>
> We believe these details, now explicitly documented in Appendix D.4, ensure our experiments are fully reproducible.
>
> > **[W4]**  From Table 2, the advantages of the proposed module does not seem significant, particulary by integrating with FastGPT and DNA-GPT, please give some explainations. Besides, More benchmark datasets and baselines should be included. For benchmark dataset, it is suggested to include experiments on datasets e.g., PubMedQA, in the black-box setting, and experiments on short-text detection are encouraged. Also, more recent metric-based baselines should be discussed and compared, e.g. ref 1 and ref 2, etc.
> >
>
> We thank the reviewer for the constructive suggestions, which have significantly strengthened the empirical breadth of our work.
>
> - **Performance gains.** The reviewer correctly observes that the gains for FastGPT and DNA-GPT in Table 2 are smaller than those for methods like Likelihood. This is an expected phenomenon that aligns with our analysis in Section 2 (A Unified Perspective on Metric-Based Detection) and Section 5.2 (Ablation Study). Specifically, methods like Likelihood are single-text methods, which are highly susceptible to the inherent randomness of the generation process. In contrast, FastGPT and DNA-GPT are sample-based methods that utilize multiple regenerated samples or perturbations to compute scores. This sampling process may mitigate random errors, leaving less “noise” for our calibration strategy. Despite this, our method still provides consistent improvements, as shown in Tables 2 to 15 in the paper.
> - **More datasets.** Regarding PubMedQA, we found that its claimed "generated content" is a heuristic transformation of human text and has nothing to do with LLM, which is not applicable for detection. To address your request for a challenging Q&A benchmark, we incorporated the TruthfulQA dataset [2], which covers Q&A tasks in 38 domains (e.g., health, law, and finance). As shown in Table 2 below (full results can be found in Table 16 in the revised version), our method consistently boosts performance. For example, DNA-GPT-M achieves a significant leap from 80.17% to 88.32% (+8.15%) in average AUROC.
> - **Short-text Detection.** Following your suggestion, we performed a stress test on the Essay dataset with a strict 50-word limit. In Table 3 below (full results can be found in Table 17 in the revised paper), our calibration remains highly effective even with limited context. For example, we improve FastGPT by 5.35% and DNA-GPT by 3.03% on average. This confirms our calibration still holds even in short sequences.
> - **More Baselines.** Per your suggestion, we have integrated state-of-the-art metric-based baselines: RepreGuard [3], Binoculars [4], Lastde [5], and FourierGPT [6], and applied our strategy to them as “M” suffix. As shown in Tables 4-7 below (full results can be found in Tables 18-20 in the revised paper), our method functions as a universal plug-in, enhancing these detectors. For example, on the Reuters dataset, our method improves RepreGuard by 8.97% and Binoculars by 4.42%.

---

> ### Author Response · Authors · 2025-11-22
> **Response to Reviewer LQNk [3/4]**
>
> **Table 2. Performance concerning AUROC (%) on TruthfulQA. Detectors are trained on GPT4 text.**
>
> | Method | GPT4 | ChatGPT-turbo | ChatGLM | Dolly | ChatGPT | StableLM | Avg. |
> | --- | --- | --- | --- | --- | --- | --- | --- |
> | Likelihood | 84.67 | 91.95 | 97.08 | 79.28 | 95.65 | 94.17 | 90.47 |
> | **Likelihood_M** | **86.61** | **92.72** | **97.14** | **83.70** | 94.20 | **94.37** | **91.46** |
> | Log-Rank | 82.55 | 90.60 | 97.03 | 78.08 | 95.11 | 94.09 | 89.57 |
> | **Log-Rank-M** | **84.16** | **93.54** | **97.56** | **82.14** | **96.36** | **95.10** | **91.48** |
> | Entropy | 79.29 | 92.19 | 96.02 | 80.50 | 94.04 | 91.30 | 88.89 |
> | **Entropy-M** | **85.48** | **95.17** | **97.45** | **84.57** | **96.83** | **95.42** | **92.49** |
> | DetectGPT | **75.73** | 86.86 | 92.84 | 72.25 | 92.84 | 86.70 | 84.54 |
> | **DetectGPT-M** | 70.03 | **90.38** | **97.23** | **78.97** | **93.82** | **91.37** | **86.96** |
> | FastGPT | 82.61 | 90.99 | 96.26 | 81.63 | 94.38 | 91.84 | 89.62 |
> | **FastGPT-M** | **82.80** | **94.26** | **97.68** | **84.02** | **96.47** | **94.84** | **91.68** |
> | DNAGPT | 77.49 | 79.64 | 88.69 | 67.60 | 84.15 | 83.46 | 80.17 |
> | **DNAGPT-M** | **83.92** | **90.18** | **92.96** | **79.81** | **92.16** | **90.90** | **88.32** |
>
> **Table 3. Performance concerning AUROC (%) on short texts on Essay. Detectors are trained on GPT4 text.**
>
> | Method | GPT4 | ChatGPT-turbo | ChatGLM | Dolly | ChatGPT | StableLM | Avg. |
> | --- | --- | --- | --- | --- | --- | --- | --- |
> | Likelihood | 84.84  | 93.57  | 83.34  | 96.34  | 81.26  | 89.30  | 88.11  |
> | **Likelihood_M** | **86.95**  | **93.65**  | **84.16**  | **97.04**  | **83.43**  | **89.59**  | **89.14**  |
> | Log-Rank | 82.90  | 92.29  | 81.18  | 95.97  | 77.58  | 87.83  | 86.29  |
> | **Log-Rank-M** | **85.88**  | **92.86**  | **83.27**  | **97.01**  | **81.40**  | **88.28**  | **88.12**  |
> | Entropy | 58.86  | 72.56  | 62.13  | 66.71  | 67.08  | 73.88  | 66.87  |
> | **Entropy-M** | **62.02**  | **74.75**  | **65.11**  | **71.53**  | **70.74**  | **74.98**  | **69.86**  |
> | DetectGPT | 78.67  | 84.93  | 76.56  | 89.40  | 79.27  | 81.54  | 81.73  |
> | **DetectGPT-M** | **86.13**  | **92.12**  | **82.57**  | **96.33**  | **83.40**  | **88.57**  | **88.19**  |
> | FastGPT | 79.78  | 85.87  | 81.15  | 91.22  | 82.62  | 80.05  | 83.45  |
> | **FastGPT-M** | **86.30**  | **90.49**  | **86.03**  | **95.26**  | **88.05**  | **86.65**  | **88.80**  |
> | DNA-GPT | 79.59  | 86.99  | 79.45  | 90.86  | 76.84  | 83.48  | 82.87  |
> | **DNA-GPT-M** | **83.46**  | **88.99**  | **83.06**  | **93.40**  | **80.65**  | **85.84**  | **85.90**  |
>
> **Table 4. Detection enhancement on the RepreGuard detector. The detectors are trained on GPT4All texts.**
>
> | Dataset | Detector | GPT4All | ChatGPT | Dolly | ChatGLM | Claude | ChatGPT-turbo | Avg. |
> | --- | --- | --- | --- | --- | --- | --- | --- | --- |
> | Essay | RepreGuard | 94.66 | 97.35 | 90.40 | 91.95 | 47.51 | 43.42 | 77.55 |
> | Essay | **RepreGuard-M** | 95.46 | 99.55 | 90.55 | 99.95 | 47.18 | 43.12 | 79.30 |
> | Reuters | RepreGuard | 97.21 | 98.32 | 8.24 | 97.26 | 21.81 | 44.32 | 61.19 |
> | Reuters | **RepreGuard-M** | 98.33 | 97.87 | 48.12 | 97.00 | 29.08 | 50.59 | 70.16 |
>
> **Table 5. Detection enhancement on the Binoculars detector.  The detectors are trained on GPT4All texts.**
>
> | Dataset | Detector | GPT4All | ChatGPT | Dolly | ChatGLM | Claude | ChatGPT-turbo | Avg. |
> | --- | --- | --- | --- | --- | --- | --- | --- | --- |
> | Essay | Binoculars | 96.33 | 98.59 | 88.57 | **99.57** | **87.21** | **98.84** | 94.85 |
> | Essay | **Binoculars-M** | **97.34** | **98.65** | **90.19** | 97.68 | 86.80 | **98.84** | **94.91** |
> | Reuters | Binoculars | 80.96 | 97.99 | 61.25 | 99.80 | 83.23 | 98.70 | 86.99 |
> | Reuters | **Binoculars-M** | **90.90** | **98.49** | **74.10** | **99.86** | **86.37** | **98.72** | **91.41** |
>
> **Table 6. Detection enhancement on the Lastde detector.  The detectors are trained on GPT4All texts.**
>
> | Dataset | Detector | GPT4All | ChatGPT | Dolly | ChatGLM | Claude | ChatGPT-turbo | Avg. |
> | --- | --- | --- | --- | --- | --- | --- | --- | --- |
> | Essay | Lastde | 99.54 | 99.72 | 97.38 | 99.98 | 75.98 | **91.16** | 93.96 |
> | Essay | **Lastde-M** | **99.68** | **99.77** | **97.87** | **99.99** | **76.21** | 90.96 | **94.08** |
> | Reuters | Lastde | 99.21 | 99.67 | 94.25 | **99.83** | 83.27 | **99.09** | 95.89 |
> | Reuters | **Lastde-M** | **99.55** | **99.68** | **96.09** | 99.82 | **85.64** | 99.00 | **96.63** |

---

> ### Author Response · Authors · 2025-11-22
> **Response to Reviewer LQNk [4/4]**
>
> **Table 7. Detection enhancement on the FourierGPT detector.  The detectors are trained on GPT4All texts.**
>
> | Dataset | Detector | GPT4All | ChatGPT | Dolly | ChatGLM | Claude | ChatGPT-turbo | Avg. |
> | --- | --- | --- | --- | --- | --- | --- | --- | --- |
> | Essay | FourierGPT | 91.70 | 90.18 | 93.74 | 98.26 | **44.90** | **42.11** | 76.81 |
> | Essay | **FourierGPT-M** | **97.10** | **95.27** | **97.64** | **99.44** | 41.14 | 41.08 | **78.61** |
> | Reuters | FourierGPT | 99.28 | 96.71 | 99.06 | 98.45 | **56.82** | 49.84 | 83.36 |
> | Reuters | **FourierGPT-M** | **99.36** | **98.29** | **99.29** | **98.48** | 55.13 | **49.88** | **83.40** |
>
> > **[Q1]**  The presentation mostly relies on figures, why not provide precisely the numerical results? (It often seems hard to tell the exact number and comparision results given the figures and bars etc).
> >
>
> Thank you for your question. We would like to clarify that we have provided **comprehensive numerical results** in the paper, primarily located in the Appendix due to space constraints. Specifically, in the main text, **Table 2** provides the precise **AUROC** (mean $\pm$ standard deviation) for the main experiments on the Essay and DetectRL datasets. In the Appendix, **Table 3** details the exact  **TPR@FPR-1%** for the Essay and DetectRL datasets. **Tables 4 through 9** provide the exact numbers for every training/testing pair on the Essay and DetectRL datasets. **Tables 10 through 15** provide the complete numerical breakdown for the Reuters dataset. **Table 16** provides the precise running times (in seconds) for training and inference. In addition, following your suggestion, we have also presented the newly added RepreGuard, Binoculars, Lastde, and FourierGPT, in numerical form (Tables 18-20 in the revised paper).
>
> Thank you again for your constructive comments. We have revised the manuscript to clearly define 'Neighbor Similarity' and experimental settings. Crucially, following your suggestions, we expanded our evaluation to include TruthfulQA, short-text scenarios, and recent SOTA baselines (RepreGuard, Binoculars, Lastde, and FourierGPT). The results consistently demonstrate that our method functions as a universal plug-in, yielding significant gains. If our revision has satisfactorily addressed your concerns, may we kindly request that you consider revising your final rating of our manuscript? If you have any additional concerns or comments that we may have missed in our responses, we would appreciate any further feedback to help us further enhance our work.
>
> [1] Scissorhands: Exploiting the persistence of importance hypothesis for llm kv cache compression at test time." *NeurIPS 2023.*
>
> [2] TruthfulQA: Measuring How Models Mimic Human Falsehoods. ACL 2022.
>
> [3] RepreGuard: Detecting LLM-Generated Text by Revealing Hidden Representation Patterns, TACL 2025.
>
> [4] Spotting LLMs With Binoculars: Zero-Shot Detection of Machine-Generated Text, ICML 2024.
>
> [5] Training-free LLM-generated Text Detection by Mining Token Probability Sequences, ICLR 2025.
>
> [6] Detecting Subtle Differences between Human and Model Languages Using Spectrum of Relative Likelihood. EMNLP 2024.

---

> > ### Author Response · Authors · 2025-11-27
> > **Seeking Reviewer feedback: Kind Reminder for Author/Reviewer Discussion Phase**
> >
> > Dear Reviewer LQNk,
> >
> > As the discussion period draws to a close, we wanted to politely check in to see if our response and the revised manuscript have satisfactorily addressed your concerns.
> >
> > We highly value your constructive feedback and have made significant updates to the paper to reflect your suggestions, specifically:
> >
> > - **Clarity (W1 & W2):** We have rewritten Section 3 and the captions for Figures 2 & 3 to explicitly define "Neighbor Similarity" and clarify the axis variables/metrics. Furthermore, we have clarified the guiding role of the theoretical scope.
> > - **New Experiments (W4):** We added the **TruthfulQA** dataset and a **short-text** stress test.
> > - **New Baselines (W4):** We integrated four recent SOTA methods (**RepreGuard, Binoculars, Lastde, and FourierGPT)** demonstrating that our method consistently boosts performance across these state-of-the-art detectors.
> > - **Experimental Settings (W3):** We added Appendix D to explicitly define the Black-Box Threat Model and Proxy settings.
> >
> > We believe these substantial additions greatly enhance the quality of our paper. We stand ready to answer any additional questions you might have and hope that, given these improvements, you might be willing to reconsider the current rating.
> >
> > Best regards,
> >
> > The Authors

---

> ### Comment · Reviewer_LQNk · 2025-11-28
>
> Dear Authors,
> Thank you for your carefully preparing the rebuttals, my concerns have been largely resolved, and after reviewing the comments/rebuttal from some other reviewers, I would like to raise my rating score to be "6: marginally above the acceptance threshold. "

---

> > ### Author Response · Authors · 2025-11-28
> >
> > We sincerely thank you for your support of our work and for raising your score to 6. Your feedback has been incredibly helpful in improving our work. Thanks again for your time, and wish you the best in your research!

---

### Author Response · Authors · 2025-12-01
**Summary of Reviewer Consensus and Our Response for Submission 16501**

Dear Area Chair,

Thank you for handling our submission. As the ratings were unexpectedly reset to their initial state, the current ratings do not reflect the significant consensus achieved during the discussion phase. We are writing to summarize the consensus reached during the discussion phase, where **all four reviewers explicitly confirmed their concerns were resolved**, with two explicitly stating they raised their scores.

**1. Post-Rebuttal Consensus & Score Updates.**

- **Reviewer LQNk (Score Raised $\to$ 6):** Explicitly stated: “my concerns have been largely resolved... I would like to **raise my rating score to be 6**”.
- **Reviewer kDth (Score Raised $\to$ 6):** Stated the response “resolved most of my concerns” and confirmed they had “**accordingly increased my rating**”.
- **Reviewer Qk3v (Score Maintained: 8):** Confirmed the response “addressed my concerns” and maintained their high score.
- **Reviewer FKz3 (Concerns Resolved):** Confirmed they “**don't have any further questions**” and are “**very open minded**” following the clarifications.

**2. Reviewer Consensus on Strengths.** The reviewers identified the following core strengths in our work:

- **Novelty & Simplicity:** Reviewer **LQNk** found the perspective on randomness issues "new" and “make sense”, while Reviewer **kDth** praised the method for being "conceptually simple, computationally efficient, and effective".
- **Strong Theoretical & Empirical Foundation:** Reviewer **Qk3v** noted the approach is "backed by strong theoretical analysis and extensive empirical validation", describing “the methodology as sound and the evidence as convincing”. Reviewer **LQNk** appreciated that the experiments “were done to validate some claims proposed in this work”.
- **Compelling Performance:** Reviewer **FKz3** acknowledged that the "basic idea of the approach is sensible and works" and that the results were "very compelling in showing the uplift in performance", Reviewer **KDth** said that “The experiments are broad and thorough”.

**3. Summary of Response Content.** To address reviewers’ feedback, we implemented substantial revisions to the manuscript.

- **Integration of SOTA Baselines (Addressing LQNk, FKz3, kDth).** We integrated RepreGuard, Binoculars, Lastde, and FourierGPT. Our Markov-informed calibration consistently boosted performance across these advanced detectors.
- **Expanded Experimental Scenarios (Addressing LQNk, Qk3v, kDth).** We added Back-Translation and Word Perturbation attacks, highlighting our robustness. We added the TruthfulQA dataset and a Short-Text stress test, achieving significant gains. Additionally, we verified our theoretical claims on the Chinese CUDRT dataset, confirming that "Neighbor Similarity" holds across languages.
- **Theoretical & Methodological Clarifications (Addressing LQNk, FKz3).** We provided a comparison showing our MRF approach outperforms a standard HMM in both accuracy and inference speed. We revised Figures 2 and 3 to explicitly define "Neighbor Similarity" and extended the analysis to 100 hops. We clarify that on the challenging DetectRL benchmark, our method transformed statistically useless detectors (e.g., DetectGPT at 48.60% AUROC) into viable ones (72.95% AUROC) with negligible compute overhead.

We hope this summary assists in your final decision-making process.

Best regards,

Authors of #16501

---

### Meta-Review · Area_Chair_Dcp3 · 2025-12-22

**Summary:**

The major concerns are about missing baselines, lack of additional benchmark datasets, an inaccurate or insufficient discussion of state-of-the-art work, misleading justifications for method’s effectiveness, and theory assumptions and relevance. Additional concerns include unclear presentation and experimental setting, and lack of evaluation under a broader set of attacks and a more thorough failure-mode analysis.

**Reviewer Concerns:**

The concerns about missing baselines and benchmark datasets, unclear presentation and experimental setting, and lack of additional experiments (more attacks and failure mode analysis) were largely addressed. The authors provided additional experiments and clarification to address the concerns about inaccurate/insufficient discussion of state-of-the-art works, potentially misleading/unconfirmed justification, and relevance of the theorem. However, some reviewers may still have concerns about the motivation and justification behind the method.

**Reviewer Scores:**

Reviewer LQNk and kDth are likely to increase the scores. Reviewer Qk3v is likely to keep the positive score. Reviewer FKz3 may or may not increase the score.

---

### Decision · Program_Chairs · 2026-01-26

Accept (Poster)